# HIERARCHIES OF REWARD MACHINES

## ABSTRACT

Reward machines (RMs) are a recent formalism for representing the reward function of a reinforcement learning task through a finite-state machine whose edges encode landmarks of the task using high-level events. The structure of RMs enables the decomposition of a task into simpler and independently solvable subtasks that help tackle long-horizon and/or sparse reward tasks. We propose a *formalism* for further abstracting the subtask structure by endowing an RM with the ability to call other RMs, thus composing a hierarchy of RMs (HRM). We *exploit* HRMs by treating each call to an RM as an independently solvable subtask using the options framework, and describe a curriculum-based method to *learn* HRMs from traces observed by the agent. Our experiments reveal that exploiting a handcrafted HRM leads to faster convergence than with a flat HRM, and that learning an HRM is feasible in cases where its equivalent flat representation is not.

## 1 INTRODUCTION

More than a decade ago, Dietterich et al. (2008) argued for the need to "learn at *multiple time scales* simultaneously, and with a rich *structure* of events and durations". Finite-state machines (FSMs) are a simple yet powerful formalism for abstractly representing temporal tasks in a structured manner. One of the most prominent types of FSMs used in reinforcement learning (RL; Sutton & Barto, 2018) are reward machines (RMs; Toro Icarte et al., 2018; 2022), where each edge is labeled with (i) a formula over a set of high-level events that capture a task's subgoal, and (ii) a reward for satisfying the formula. Hence, RMs fulfill the need for structuring events and durations. Hierarchical reinforcement learning (HRL; Barto & Mahadevan, 2003) frameworks, such as options (Sutton et al., 1999), have been applied over RMs to learn policies at two levels of abstraction: (i) select a formula (i.e., subgoal) from a given RM state to complete the overall task, and (ii) select an action to satisfy the chosen formula (Toro Icarte et al., 2018; Furelos-Blanco et al., 2021). Thus, RMs also allow learning at multiple scales simultaneously. The subtask decomposition powered by HRL eases the handling of long-horizon and sparse reward tasks. Besides, RMs can act as an external memory in partially observable tasks by keeping track of the subgoals achieved so far and those to be achieved.

In this work, we make the following contributions:

1. Enhance the abstraction power of RMs by defining *hierarchies of RMs (HRMs)*, where constituent RMs can call other RMs (Section 3). We prove that any HRM can be transformed into an *equivalent* flat HRM that behaves exactly like the original RMs. We show that under certain conditions, the equivalent flat HRM can have exponentially more states and edges.

2. Propose an HRL algorithm to *exploit* HRMs by treating each call as a subtask (Section 4). Learning policies in HRMs further fulfills the desiderata posed by Dietterich et al. since (i) there is an arbitrary number of time scales to learn across (not only two), and (ii) there is a richer range of increasingly abstract events and durations. Besides, hierarchies enable *modularity* and, hence, the *reusability* of the RMs and policies. Empirically, we show that leveraging a handcrafted HRM enables faster convergence than an equivalent flat HRM.

3. Introduce a curriculum-based method for *learning* HRMs from traces given a set of hierarchically composable tasks (Section 5). In line with the theory (Contribution 1), our experiments reveal that decomposing an RM into several is *crucial* to make its learning feasible (i.e., the flat HRM cannot be efficiently learned from scratch) since (i) the constituent RMs are simpler (i.e., they have fewer states and edges), and (ii) previously learned RMs can be used to efficiently *explore* the environment in the search for traces in more complex tasks.

## 2 BACKGROUND

Given a finite set $\mathcal{X}$, we use $\Delta(\mathcal{X})$ to denote the probability simplex over $\mathcal{X}$, $\mathcal{X}^*$ to denote (possibly empty) sequences of elements from $\mathcal{X}$, and $\mathcal{X}^+$ to denote non-empty sequences. We also use $\bot$ and $\top$ to denote the truth values false and true, respectively. $\mathbb{1}[A]$ is the indicator function of event $A$.

We represent RL tasks as episodic labeled Markov decision processes (MDPs, Xu et al., 2020), each consisting of a set of states $\mathcal{S}$, a set of actions $\mathcal{A}$, a transition function $p : \mathcal{S} \times \mathcal{A} \to \Delta(\mathcal{S})$, a reward function $r : (\mathcal{S} \times \mathcal{A})^+ \times \mathcal{S} \to \mathbb{R}$, a discount factor $\gamma \in [0, 1)$, a finite set of *propositions* $\mathcal{P}$ representing high-level events, a *labeling function* $l : \mathcal{S} \to 2^{\mathcal{P}}$ mapping states to proposition subsets called *labels*, and a *termination function* $\tau : (\mathcal{S} \times \mathcal{A})^* \times \mathcal{S} \to \{\bot, \top\} \times \{\bot, \top\}$. Hence the transition function $p$ is Markovian, but the reward function $r$ and termination function $\tau$ are not. Given a *history* $h_t = \langle s_0, a_0, \dots, s_t \rangle \in (\mathcal{S} \times \mathcal{A})^* \times \mathcal{S}$, a *label trace* (or trace, for short) $\lambda_t = \langle l(s_0), \dots, l(s_t) \rangle \in (2^{\mathcal{P}})^+$ assigns labels to all states in $h_t$. We assume $(\lambda_t, s_t)$ captures all relevant information about $h_t$; thus, the reward and transition information can be written $r(h_t, a_t, s_{t+1}) = r(h_{t+1}) = r(\lambda_{t+1}, s_{t+1})$ and $\tau(h_t) = \tau(\lambda_t, s_t)$, respectively. We aim to find a policy $\pi : (2^{\mathcal{P}})^+ \times \mathcal{S} \to \mathcal{A}$, a mapping from traces-states to actions, that maximizes the expected cumulative discounted reward (or *return*) $R_t = \mathbb{E}_\pi[\sum_{k=t}^n \gamma^{k-t} r(\lambda_{k+1}, s_{k+1})]$, where $n$ is the last episode's step.

At time $t$, the trace is $\lambda_t \in (2^{\mathcal{P}})^+$, and the agent observes a tuple $\boldsymbol{s}_t = \langle s_t, s_t^T, s_t^G \rangle$, where $s_t \in \mathcal{S}$ is the state and $(s_t^T, s_t^G) = \tau(\lambda_t, s_t)$ is the termination information, with $s_t^T$ and $s_t^G$ indicating whether or not the history $(\lambda_t, s_t)$ is terminal or a goal, respectively. If the history is non-terminal, the agent runs action $a_t \in \mathcal{A}$, and the environment transitions to state $s_{t+1} \sim p(\cdot | s_t, a_t)$. The agent then extends the trace as $\lambda_{t+1} = \lambda_t \oplus l(s_{t+1})$, receives reward $r_{t+1} = r(\lambda_{t+1}, s_{t+1})$, and observes a new tuple $\boldsymbol{s}_{t+1}$. A trace $\lambda_t$ is a *goal* trace if $(s_t^T, s_t^G) = (\top, \top)$, a *dead-end* trace if $(s_t^T, s_t^G) = (\top, \bot)$, and an *incomplete* trace if $s_t^T = \bot$. We assume that the reward is $r(\lambda_{t+1}, s_{t+1}) = \mathbb{1}[\tau(\lambda_{t+1}, s_{t+1}) = (\top, \top)]$, i.e. 1 for goal histories and 0 otherwise.

A *(simple) reward machine* (RM; Toro Icarte et al., 2018; 2022) is a tuple $\langle \mathcal{U}, \mathcal{P}, \varphi, r, u^0, \mathcal{U}^A, \mathcal{U}^R \rangle$, where $\mathcal{U}$ is a finite set of states; $\mathcal{P}$ is a finite set of propositions; $\varphi : \mathcal{U} \times \mathcal{U} \to \text{DNF}_{\mathcal{P}}$ is a state transition function such that $\varphi(u, u')$ denotes the disjunctive normal form (DNF) formula over $\mathcal{P}$ to be satisfied to transition from $u$ to $u'$; $r : \mathcal{U} \times \mathcal{U} \to \mathbb{R}$ is a reward transition function such that $r(u, u')$ is the reward for transitioning from $u$ to $u'$; $u^0 \in \mathcal{U}$ is an initial state; $\mathcal{U}^A \subseteq \mathcal{U}$ is a set of accepting states denoting the task's goal achievement; and $\mathcal{U}^R \subseteq \mathcal{U}$ is a set of rejecting states denoting the unfeasibility of achieving the goal. Ideally, RM states should capture traces, such that (i) pairs $(u, s)$ of an RM state and an MDP state are sufficient to predict the future, and (ii) the reward $r(u, u')$ matches the underlying MDP's reward. The state transition function is *deterministic*, i.e. at most one formula from each state is satisfied. To verify if a formula is satisfied by a label $\mathcal{L} \subseteq \mathcal{P}$, $\mathcal{L}$ is used as truth assignment where propositions in $\mathcal{L}$ are true, and false otherwise (e.g., $\{a\} \models a \wedge \neg b$).

Options (Sutton et al., 1999) address temporal abstraction in RL. Given an episodic labeled MDP, an option is a tuple $\omega = \langle \mathcal{I}_\omega, \pi_\omega, \beta_\omega \rangle$, where $\mathcal{I}_\omega \subseteq \mathcal{S}$ is the option's initiation set, $\pi_\omega : \mathcal{S} \to \mathcal{A}$ is the option's policy, and $\beta_\omega : \mathcal{S} \to [0, 1]$ is the option's termination condition. An option is available in $s \in \mathcal{S}$ if $s \in \mathcal{I}_\omega$, selects actions according to $\pi_\omega$, and terminates in $s \in \mathcal{S}$ with probability $\beta_\omega(s)$.

## 3 FORMALIZATION OF HIERARCHIES OF REWARD MACHINES

We here introduce our formalism for hierarchically composing reward machines, and propose the CRAFTWORLD domain (cf. Figure 1a) to illustrate it. In this domain, the agent (▲) can move forward or rotate 90°, staying put if it moves towards a wall. Locations are labeled with propositions from $\mathcal{P} = \{\text{🚪, ⛏, ♖, ✚, ✦, ⚒, ☁, ✂, ⚙, ⌂}\}$. The agent observes propositions that it steps on (e.g., $\{⚒\}$ in the top-left corner). Table 1 lists tasks that consist of observing a sequence of propositions, where the reward is 1 if the sequence is observed and 0 otherwise. These tasks are based on those by Andreas et al. (2017) and Toro Icarte et al. (2018), but they can be defined in terms of each other.

Reward machines (RMs) are the building blocks of our formalism. To constitute a hierarchy of RMs, we need to endow RMs with the ability to call each other. We redefine the *state transition function* as $\varphi : \mathcal{U} \times \mathcal{U} \times \mathcal{M} \to \text{DNF}_{\mathcal{P}}$, where $\mathcal{M}$ is a set of RMs. The expression $\varphi(u, u', M)$ denotes the disjunctive normal form (DNF) formula over $\mathcal{P}$ that must be satisfied to transition from $u \in \mathcal{U}$ to $u' \in \mathcal{U}$ by calling RM $M \in \mathcal{M}$. We refer to the formulas $\varphi(u, u', M)$ as *contexts*

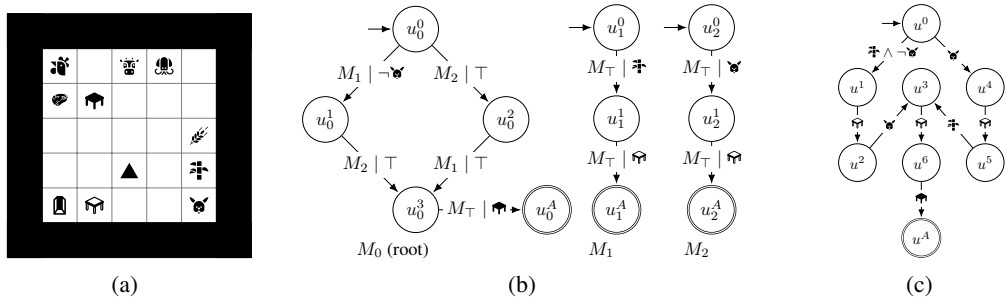

(a)                              (b)                              (c)

Figure 1: A CRAFTWORLD grid (a), an HRM for BOOK (b), and an equivalent flat HRM (c). An edge from state $u$ to $u'$ of an RM $M_i$ is of the form $M_j \mid \varphi_i(u, u', M_j)$, double circled states are accepting states, and loop transitions are omitted. Calls to the leaf RM $M_\top$ are omitted in (c).

Table 1: List of CRAFTWORLD tasks. Descriptions "$x \; ; \; y$" express sequential order (observe/do $x$ then $y$), and descriptions "$x \;\&\; y$" express that $x$ and $y$ can be observed/done in any order.

| Task | h | Description | Task | h | Description | Task | h | Description |
|---|---|---|---|---|---|---|---|---|
| BATTER | 1 | (🗡 & 🐄) ; 🏠 | QUILL | 1 | (🐙 & 🗡) ; 🏠 | BOOKQUILL | 3 | BOOK & QUILL |
| BUCKET | 1 | 🪣 ; 🏠 | SUGAR | 1 | ✥ ; 🏠 | MILKB.SUGAR | 3 | MILKBUCKET & SUGAR |
| COMPASS | 1 | (🪣 & 🍪) ; 🗺 | BOOK | 2 | (PAPER & LEATHER) ; 🏠 | CAKE | 4 | BATTER ; MILKB.SUGAR ; 🗺 |
| LEATHER | 1 | 🐄 ; 🗺 | MAP | 2 | (PAPER & COMPASS) ; 🏠 | | | |
| PAPER | 1 | ✥ ; 🗺 | MILKBUCKET | 2 | BUCKET ; 🐄 | | | |

since they represent conditions under which calls are made. As we will see later, contexts help preserve determinism and must be satisfied to start a call (a necessary but not sufficient condition). The hierarchies we consider contain an RM $M_\top$ called the *leaf* RM, which solely consists of an accepting state (i.e., $\mathcal{U}_\top = \mathcal{U}_\top^A = \{u_\top^0\}$), and immediately returns control to the RM that calls it.

**Definition 1.** A *hierarchy of reward machines (HRM)* is a tuple $H = \langle \mathcal{M}, M_r, \mathcal{P} \rangle$, where $\mathcal{M} = \{M_0, \ldots, M_{m-1}\} \cup \{M_\top\}$ is a set of $m$ RMs and the leaf RM $M_\top$, $M_r \in \mathcal{M} \setminus \{M_\top\}$ is the root RM, and $\mathcal{P}$ is a finite set of propositions used by all constituent RMs.

We make the following *assumptions*: (i) HRMs do not have circular dependencies (i.e., an RM cannot be called back from itself, including recursion), (ii) rejecting states are global (i.e., cause the root task to fail), (iii) accepting and rejecting states do not have transitions to other states, and (iv) the reward function of the root corresponds to the reward obtained in the underlying MDP. Given assumption (i), each RM $M_i$ has a *height* $h_i$, which corresponds to the maximum number of nested calls needed to reach the leaf. Formally, if $i = \top$, then $h_i = 0$; otherwise, $h_i = 1 + \max_j h_j$, where $j$ ranges over all RMs called by $M_i$ (i.e., there exists $(u, v) \in \mathcal{U}_i \times \mathcal{U}_i$ such that $\varphi_i(u, v, M_j) \neq \bot$).

Figure 1b shows BOOK's HRM, whose root has height 2. The PAPER and LEATHER RMs, which have height 1 and consist of observing a two-proposition sequence, can be run in any order followed by observing 🏠. The context ¬🐄 in the call to $M_1$ preserves determinism, as detailed later.

In the following paragraphs, we describe how an HRM processes a label trace. To indicate where the agent is in an HRM, we define the notion of hierarchy states.

**Definition 2.** Given an HRM $H = \langle \mathcal{M}, M_r, \mathcal{P} \rangle$, a *hierarchy state* is a tuple $\langle M_i, u, \Phi, \Gamma \rangle$, where $M_i \in \mathcal{M}$ is an RM, $u \in \mathcal{U}_i$ is a state, $\Phi \in \mathrm{DNF}_\mathcal{P}$ is an accumulated context, and $\Gamma$ is a call stack.

**Definition 3.** Given an HRM $H = \langle \mathcal{M}, M_r, \mathcal{P} \rangle$, a *call stack* $\Gamma$ contains tuples $\langle u, v, M_i, M_j, \phi, \Phi \rangle$, each denoting a call where $u \in \mathcal{U}_i$ is the state from which the call is made; $v \in \mathcal{U}_i$ is the next state in the calling RM $M_i \in \mathcal{M}$ after reaching an accepting state of the called RM $M_j \in \mathcal{M}$; $\phi \in \mathrm{DNF}_\mathcal{P}$ are the disjuncts of $\varphi_i(u, v, M_j)$ satisfied by a label; and $\Phi \in \mathrm{DNF}_\mathcal{P}$ is the accumulated context.

Call stacks determine where to resume the execution. Each RM appears in the stack at most once since, by assumption, HRMs have no circular dependencies. We use $\Gamma \oplus \langle u, v, M_i, M_j, \phi, \Phi \rangle$ to denote a stack recursively defined by a stack $\Gamma$ and a top element $\langle u, v, M_i, M_j, \phi, \Phi \rangle$, where the *accumulated context* $\Phi$ is the condition under which a call from a state $u$ is made. The initial hierarchy state of an HRM $H = \langle \mathcal{M}, M_r, \mathcal{P} \rangle$ is $\langle M_r, u_r^0, \top, [] \rangle$: we are in the initial state of the root, there is no accumulated context, and the stack is empty.

At the beginning of this section, we mention that satisfying the context of a call is a necessary but not sufficient condition to start the call. We now introduce a sufficient condition, called exit condition.

**Definition 4.** Given an HRM $H = \langle \mathcal{M}, M_r, \mathcal{P} \rangle$ and a hierarchy state $\langle M_i, u, \Phi, \Gamma \rangle$, the *exit condition* $\xi_{i,u,\Phi} \in \text{DNF}_{\mathcal{P}}$ is the formula that must be satisfied to leave that hierarchy state. Formally,

$$
\xi_{i,u,\Phi} = \begin{cases} \Phi & \text{if } i = \top, \\ \bigvee_{\substack{\phi = \varphi_i(u,v,M_j), \\ \phi \neq \bot, v \in \mathcal{U}_i, M_j \in \mathcal{M}}} \xi_{j,u_j^0,\text{DNF}(\Phi \wedge \phi)} & \text{otherwise,} \end{cases}
$$

where $\text{DNF}(\Phi \wedge \phi)$ is $\Phi \wedge \phi$ in DNF. The formula is $\Phi$ if $M_i = M_\top$ since it always returns control once called. Otherwise, the formula is recursively defined as the disjunction of the exit conditions from the initial state of the called RM. For instance, the exit condition for the initial hierarchy state in Figure 1b is $(\neg \text{❦} \wedge \text{⚐}) \vee \text{❦}$.

We now have everything needed to define the *hierarchical transition function* $\delta_H$, which maps a hierarchy state $\langle M_i, u, \Phi, \Gamma \rangle$ into another given a label $\mathcal{L}$. There are three cases:

1. If $u$ is an accepting state of $M_i$ and the stack $\Gamma$ is non-empty, pop the top element of $\Gamma$ and return control to the previous RM, recursively applying $\delta_H$ in case several accepting states are reached simultaneously. Formally, the next hierarchy state is $\delta_H(\langle M_j, u', \top, \Gamma' \rangle, \bot)$ if $u \in \mathcal{U}_i^A, |\Gamma| > 0$, where $\Gamma = \Gamma' \oplus \langle \cdot, u', M_j, M_i, \cdot, \cdot \rangle$, $\bot$ denotes a label that cannot satisfy any formula, and $\cdot$ denotes something unimportant for the case.

2. If $\mathcal{L}$ satisfies the context of a call and the exit condition from the initial state of the called RM, push the call onto the stack and recursively apply $\delta_H$ until $M_\top$ is reached. Formally, the next hierarchy state is $\delta_H(\langle M_j, u_j^0, \Phi', \Gamma \oplus \langle u, u', M_i, M_j, \phi, \Phi \rangle \rangle, \mathcal{L})$ if $\mathcal{L} \models \xi_{j,u_j^0,\Phi'}$, where $\phi = \varphi_i(u, u', M_j)(\mathcal{L})$ and $\Phi' = \text{DNF}(\Phi \wedge \phi)$. Here, $\varphi(\mathcal{L})$ denotes the disjuncts of a DNF formula $\varphi \in \text{DNF}_{\mathcal{P}}$ satisfied by $\mathcal{L}$.

3. If none of the previous conditions holds, the hierarchy state remains unchanged.

The state transition functions $\varphi$ of the RMs must be such that $\delta_H$ is *deterministic*, i.e. a label cannot simultaneously satisfy the contexts and exit conditions associated with two triplets $\langle u, v, M_i \rangle$ and $\langle u, v', M_j \rangle$ such that either (i) $v = v'$ and $i \neq j$, or (ii) $v \neq v'$. Contexts help enforce determinism by making formulas mutually exclusive. For instance, if the call to $M_1$ from the initial state of $M_0$ in Figure 1b had context $\top$ instead of $\neg \text{❦}$, then $M_1$ and $M_2$ could be both started if $\{\text{⚐}, \text{❦}\}$ was observed, thus making the HRM non-deterministic. Finally, we introduce hierarchy traversals, which determine how a label trace is processed by an HRM using $\delta_H$.

**Definition 5.** Given a label trace $\lambda = \langle \mathcal{L}_0, \ldots, \mathcal{L}_n \rangle$, a *hierarchy traversal* $H(\lambda) = \langle v_0, v_1, \ldots, v_{n+1} \rangle$ is a unique sequence of hierarchy states such that (i) $v_0 = \langle M_r, u_r^0, \top, [] \rangle$, and (ii) $\delta_H(v_i, \mathcal{L}_i) = v_{i+1}$ for $i = 0, \ldots, n$. An HRM $H$ *accepts* $\lambda$ if $v_{n+1} = \langle M_r, u, \top, [] \rangle$ and $u \in \mathcal{U}_r^A$ (i.e., an accepting state of the root is reached). Analogously, $H$ *rejects* $\lambda$ if $v_{n+1} = \langle M_k, u, \cdot, \cdot \rangle$ and $u \in \mathcal{U}_k^R$ for any $k \in [0, m-1]$ (i.e., a rejecting state in the HRM is reached).

**Example 1.** The HRM in Figure 1b accepts label trace $\lambda = \langle \{\text{⚐}\}, \{\text{⌂}\}, \{\}, \{\text{❦}\}, \{\text{⌂}\}, \{\text{♠}\} \rangle$ since the traversal is $H(\lambda) = \langle \langle M_0, u_0^0, \top, [] \rangle, \langle M_1, u_1^1, \top, [\langle u_0^0, u_0^1, M_0, M_1, \neg \text{❦}, \top \rangle] \rangle, \langle M_0, u_0^1, \top, [] \rangle, \langle M_0, u_0^1, \top, [] \rangle, \langle M_2, u_2^1, \top, [\langle u_0^1, u_0^3, M_0, M_2, \top, \top \rangle] \rangle, \langle M_0, u_0^3, \top, [] \rangle, \langle M_0, u_0^A, \top, [] \rangle \rangle$. The step-by-step application of the hierarchical transition function $\delta_H$ is shown in Appendix A.

The behavior of an HRM $H$ can be reproduced by an *equivalent flat* HRM $\bar{H}$; that is, (i) the root of $\bar{H}$ has height 1 and, (ii) $\bar{H}$ accepts a trace iff $H$ accepts it, rejects a trace iff $H$ rejects it, and neither accepts nor rejects a trace iff $H$ does not accept it nor reject it. Flat HRMs thus capture the original RM definition. Figure 1c shows a flat HRM for the BOOK task. We formally define equivalence and prove the equivalence theorem below by construction in Appendix B.1.

**Theorem 1.** *Given an HRM $H$, there exists an equivalent flat HRM $\bar{H}$.*

Given the construction used in Theorem 1, we show that the number of states and edges of the resulting flat HRM can be *exponential* in the height of the root (see Theorem 2). We prove this result in Appendix B.2 through an instance of a general HRM parametrization where the constituent RMs are *highly reused*, hence illustrating the convenience of HRMs to succinctly compose existing knowledge. In line with the theory, learning a non-flat HRM can take a few seconds, whereas learning an equivalent flat HRM is often unfeasible (see Section 6).

**Theorem 2.** *Let $H = \langle \mathcal{M}, M_r, \mathcal{P} \rangle$ be an HRM and let $h_r$ be the height of its root $M_r$. The number of states and edges in an equivalent flat HRM $\bar{H}$ can be exponential in $h_r$.*

## 4  POLICY LEARNING IN HIERARCHIES OF REWARD MACHINES

In what follows, we explain how to *exploit* the temporal structure of an HRM $H = \langle \mathcal{M}, M_r, \mathcal{P} \rangle$ using two types of *options*. We describe (i) how to learn the policies of these options, (ii) when these options terminate, and (iii) an option selection algorithm that ensures the currently running options and the current hierarchy state are aligned. We discuss implementation details in Appendix C.

**Types.** Given an RM $M_i \in \mathcal{M}$, a state $u \in \mathcal{U}_i$ and a context $\Phi$, an option $\omega_{i,u,\Phi}^{j,\phi}$ is derived for each non-false disjunct $\phi$ of each transition $\varphi_i(u, v, M_j)$, where $v \in \mathcal{U}_i$ and $M_j \in \mathcal{M}$. An option is either (i) a *formula option* if $j = \top$ (i.e., $M_\top$ is called), or (ii) a *call option* otherwise. A formula option attempts to reach a label that satisfies $\phi \wedge \Phi$ through primitive actions, whereas a call option aims to reach an accepting state of the called RM $M_j$ under context $\phi \wedge \Phi$ by invoking other options.

**Policies.** Policies are $\epsilon$-greedy during training, and greedy during evaluation. A *formula option's policy* is derived from a Q-function $q_{\phi \wedge \Phi}(s, a; \boldsymbol{\theta}_{\phi \wedge \Phi})$ approximated by a deep Q-network (DQN; Mnih et al., 2015) with parameters $\boldsymbol{\theta}_{\phi \wedge \Phi}$, which outputs the Q-value of each action given an MDP state. We store all options' experiences $(\boldsymbol{s}_t, a, \boldsymbol{s}_{t+1})$ in a single replay buffer $\mathcal{D}$, thus performing intra-option learning (Sutton et al., 1998). The Q-learning update uses the following loss function:

$$\mathbb{E}_{(\boldsymbol{s}_t, a, \boldsymbol{s}_{t+1}) \sim \mathcal{D}} \left[ \left( r_{\phi \wedge \Phi}(\boldsymbol{s}_{t+1}) + \gamma \max_{a'} q_{\phi \wedge \Phi}(\boldsymbol{s}_{t+1}, a'; \boldsymbol{\theta}_{\phi \wedge \Phi}^-) - q_{\phi \wedge \Phi}(\boldsymbol{s}_t, a; \boldsymbol{\theta}_{\phi \wedge \Phi}) \right)^2 \right], \quad (1)$$

where $r_{\phi \wedge \Phi}(\boldsymbol{s}_{t+1}) = \mathbb{1}[l(\boldsymbol{s}_{t+1}) \models \phi \wedge \Phi]$, i.e. the reward is 1 if $\phi \wedge \Phi$ is satisfied and 0 otherwise; the term $q_{\phi \wedge \Phi}(s_{t+1}, a'; \boldsymbol{\theta}_{\phi \wedge \Phi}^-)$ is 0 when $\phi \wedge \Phi$ is satisfied or a dead-end is reached (i.e., $s_{t+1}^T = \top$ and $s_{t+1}^G = \bot$); and $\boldsymbol{\theta}_{\phi \wedge \Phi}^-$ are the parameters of a fixed target network.

A *call option's policy* is induced by a Q-function $q_i(s, u, \Phi, \langle M_j, \phi \rangle; \boldsymbol{\theta}_i)$ associated with the called RM $M_i$ and approximated by a DQN with parameters $\boldsymbol{\theta}_i$ that outputs the Q-value of each call in the RM given an MDP state, an RM state and a context. We store experiences $(\boldsymbol{s}_t, \omega_{i,u,\Phi}^{j,\phi}, \boldsymbol{s}_{t+k})$ in a replay buffer $\mathcal{D}_i$ associated with $M_i$, and perform SMDP Q-learning using the following loss:

$$\mathbb{E}_{(\boldsymbol{s}_t, \omega_{i,u,\Phi}^{j,\phi}, \boldsymbol{s}_{t+k}) \sim \mathcal{D}_i} \left[ \left( r + \gamma^k \max_{j',\phi'} q_i(s_{t+k}, u', \Phi', \langle M_{j'}, \phi' \rangle; \boldsymbol{\theta}_i^-) - q_i(s_t, u, \Phi, \langle M_j, \phi \rangle; \boldsymbol{\theta}_i) \right)^2 \right],$$

where $k$ is the number of steps between $\boldsymbol{s}_t$ and $\boldsymbol{s}_{t+k}$; $r$ is the sum of discounted rewards during this time; $u'$ and $\Phi'$ are the RM state and context after running the option; $M_{j'}$ and $\phi'$ correspond to an outgoing transition from $u'$, i.e. $\phi' \in \varphi_i(u', \cdot, M_{j'})$; and $\boldsymbol{\theta}_i^-$ are the parameters of a fixed target network. The term $q_i(s_{t+k}, \dots)$ is 0 if $u'$ is accepting or rejecting. Following the definition of $\delta_H$, $\Phi'$ is $\top$ if the hierarchy state changes; thus, $\Phi' = \top$ if $u' \neq u$, and $\Phi' = \Phi$ otherwise. Following our assumption on the MDP reward, we define reward transition functions as $r_i(u, u') = \mathbb{1}[u \notin \mathcal{U}_i^A \wedge u' \in \mathcal{U}_i^A]$. Learning a call option's policy and lower-level option policies simultaneously can be unstable due to *non-stationarity* (Levy et al., 2019), e.g. the same lower-level option may only sometimes achieve its goal. To relax this problem, experiences are added to the buffer only when options achieve their goal (i.e., call options assume low-level options to terminate successfully). Due to the hierarchical structure, the policies will be *recursively optimal* (Dietterich, 2000) at best.

**Termination.** An option terminates in two cases. First, if the episode ends in a goal state or in a dead-end state. Second, if the hierarchy state changes and either successfully completes the option or interrupts the option. Concretely, a formula option $\omega_{i,u,\Phi}^{j,\phi}$ is only applicable in a hierarchy state $\langle M_i, u, \Phi, \Gamma \rangle$, while a call option $\omega_{i,u,\Phi}^{j,\phi}$ always corresponds to a stack item $\langle u, \cdot, M_i, M_j, \phi, \Phi \rangle$. We can thus analyze the hierarchy state to see if an option is still executing or should terminate.

**Algorithm.** An *option stack* $\Omega_H$ stores the currently executing options. Initially, $\Omega_H$ is empty. At each step, $\Omega_H$ is filled by repeatedly choosing options starting from the current hierarchy state using call option policies until a formula option is selected. Since HRMs have, by assumption, no circular dependencies, a formula option will eventually be chosen. After an action is selected using the

formula option's policy and applied, the DQNs associated with formula options are updated. The new hierarchy state is then used to determine which options in $\Omega_H$ have terminated. Experiences for the terminated options that achieved their goal are pushed into the corresponding buffers, and the DQNs associated with the call options are updated. Finally, $\Omega_H$ is updated to *match the call stack* of the new hierarchy state (if needed) by mapping each call stack item into an option, and adding it to $\Omega_H$ if it is not already there. By aligning the option stack with the call stack, we can update DQNs for options that ended up being run in *hindsight* and which would have been otherwise ignored.

## 5   LEARNING HIERARCHIES OF REWARD MACHINES FROM TRACES

In the previous section, we explained how a *given* HRM can be exploited using options; however, engineering an HRM is impractical. We here describe LHRM, a method that *interleaves* policy learning with HRM learning from interaction. We consider a *multi-task* setting. Given $T$ tasks and $I$ instances (e.g., grids) of an environment, the agent learns (i) an HRM for each task using traces from several instances for better accuracy, and (ii) general policies to reach the goal in each task-instance pair. Namely, the agent interacts with $T \times I$ MDPs $\mathbb{M}_{ij}$, where $i \in [1, T]$ and $j \in [1, I]$. The learning proceeds from simpler to harder tasks such that HRMs for the latter build on the former.

In what follows, we detail the components of LHRM. We *assume* that (i) all MDPs share propositions $\mathcal{P}$ and actions $\mathcal{A}$, while those defined on a given instance share states $\mathcal{S}$ and labeling function $l$; (ii) to stabilize policy learning, dead-end traces must be common across tasks;[1] (iii) the root's height of a task's HRM (or *task level*, for brevity) is known (see Table 1 for CRAFTWORLD); and (iv) without loss of generality, each RM has a single accepting state and a single rejecting state.

**Curriculum Learning (Bengio et al., 2009).** LHRM learns the tasks' HRMs from lower to higher levels akin to Pierrot et al. (2019). Before starting an episode, LHRM selects an MDP $\mathbb{M}_{ij}$, where $i \in [1, T]$ and $j \in [1, I]$. The probability of selecting an MDP $\mathbb{M}_{ij}$ is determined by an estimate of its average undiscounted return $R_{ij}$ such that lower returns are mapped into higher probabilities (see details in Appendix D). Initially, only level 1 MDPs can be chosen. When the minimum average return across MDPs up to the current level surpasses a given threshold, the current level increases by 1, hence ensuring the learned HRMs and their associated policies are reusable in higher level tasks.

**Learning an HRM.** The learning of an HRM is analogous to learning a flat RM (Toro Icarte et al., 2019; Xu et al., 2020; Furelos-Blanco et al., 2021; Hasanbeig et al., 2021). The objective is to learn the state transition function $\varphi_r$ of the root $M_r$ with height $h_r$ given (i) a set of states $\mathcal{U}_r$, (ii) a set of label traces $\Lambda = \Lambda^G \cup \Lambda^D \cup \Lambda^I$, (iii) a set of propositions $\mathcal{P}$, (iv) a set of RMs $\mathcal{M}$ with lower heights than $h_r$, (v) a set of callable RMs $\mathcal{M}_{\mathcal{C}} \subseteq \mathcal{M}$ (by default, $\mathcal{M}_{\mathcal{C}} = \mathcal{M}$), and (vi) the maximum number of disjuncts $\kappa$ in the DNF formulas labeling the edges. The learned state transition function $\varphi_r$ is such that the resulting HRM $H = \langle \mathcal{M} \cup \{M_r\}, M_r, \mathcal{P} \rangle$ accepts all goal traces $\Lambda^G$, rejects all dead-end traces $\Lambda^D$, and neither accepts or rejects incomplete traces $\Lambda^I$. The transition functions can be represented as sets of logic rules, which are learned using the ILASP (Law et al., 2015) inductive logic programming system (see Appendix E for details on the ILASP encoding).

**Interleaving Algorithm.** LHRM *interleaves* the induction of HRMs with policy learning akin to Furelos-Blanco et al. (2021). Initially, the HRM's root of each task $i \in [1, T]$ consists of 3 states (the initial, accepting, and rejecting states) and neither accepts nor rejects anything. A new HRM is learned when an episode's label trace is not correctly recognized by the current HRM (i.e., if a goal trace is not accepted, a dead-end trace is not rejected, or an incomplete trace is accepted or rejected). The number of states in $\mathcal{U}_r$ increases by 1 when an HRM that covers the examples cannot be learned, hence guaranteeing that the root has the smallest possible number of states (i.e., it is *minimal*) for a specific value of $\kappa$. When an HRM for task $i$ is learned, the returns $R_{ij}$ in the curriculum are set to 0 for all $j \in [1, I]$. Analogously to some RM learning methods (Toro Icarte et al., 2019; Xu et al., 2020; Hasanbeig et al., 2021), the first HRM for a task is learned using a set of traces; in our case, the $\rho_s$ shortest traces from a set of $\rho$ goal traces are used (empirically, short traces speed up learning). Finally, LHRM leverages learned options to *explore* the environment during the collection of the $\rho$ goal traces, speeding up the process when labels are sparse. Specifically, options from lower height RMs are sequentially selected uniformly at random, and their greedy policy is run until termination.

---

[1] The term $q_{\phi \wedge \Phi}(s_{t+1}, \ldots)$ in Equation 1 is 0 if $(s_{t+1}^T, s_{t+1}^G) = (\top, \bot)$. Since experiences $(s_t, a, s_{t+1})$ are shared through the replay buffer, evaluating the condition differently can produce instabilities.

## 6    EXPERIMENTAL RESULTS

We evaluate the policy and HRM learning components of our approach using two domains described below. We report the average performances across 5 runs, each consisting of a different set of 10 random instances. Learning curves show the average undiscounted return obtained by the greedy policy every 100 episodes across instances. For other metrics (e.g., learning times), we present the average and the standard error, with the latter in brackets. In HRM learning experiments, we set a 2-hour timeout to learn the HRMs. See Appendix F for experimental details and extended results.

**Domains.** We consider four grid types for the CRAFTWORLD domain introduced in Section 3: an open plan $7 \times 7$ grid (OP, Figure 1a), an open plan $7 \times 7$ grid with a lava location (OPL), a $13 \times 13$ four rooms grid (FR; Sutton et al., 1999), and a $13 \times 13$ four rooms grid with a lava location per room (FRL). The lava proposition must always be avoided. WATERWORLD (Karpathy, 2015; Sidor, 2016; Toro Icarte et al., 2018) consists of a 2D box containing 12 balls of 6 different colors (2 per color) each moving at a constant speed in a fixed direction. The agent ball can change its velocity in any cardinal direction. The propositions $\mathcal{P} = \{r, g, b, c, m, y\}$ are the balls' colors. Labels consist of the color of the balls the agent overlaps with and, unlike CRAFTWORLD, they may contain multiple propositions. The tasks consist in observing color sequences. We consider two settings: without dead-ends (WOD) and with dead-ends (WD). In WD, the agent must avoid 2 balls of an extra color.

**Policy Learning in Handcrafted HRMs.** We compare the performance of policy learning in handcrafted non-flat HRMs against in flat equivalents. Remember that an equivalent flat HRM always exists for any HRM (see Theorem 1). For fairness, the flat HRMs are minimal. Figure 2 shows the learning curves for some CRAFTWORLD tasks in the FRL setting. The convergence rate is similar in the simplest task (MILKBUCKET), but higher for non-flat HRMs in the hardest ones. As both approaches use the same set of formula option policies, differences arise from the lack of modularity in flat HRMs. Call options, which are not present in flat HRMs, constitute independent modules that help reduce reward sparsity. MILKBUCKET involves less high-level steps than BOOKQUILL and CAKE, thus reward is less sparse and non-flat HRMs are not as beneficial. The effectiveness of non-flat HRMs is also limited when (i) the task's goal is reachable regardless of the chosen options (e.g., if the are no edges to rejecting states, like in OP and FR), and (ii) the reward is not too sparse, like in OPL (the grid is small) or WATERWORLD (the balls can easily get near the agent).

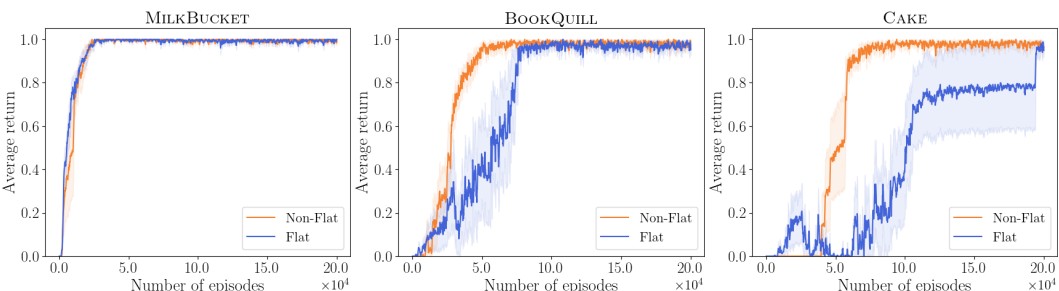

Figure 2: Learning curves for three CRAFTWORLD (FRL) tasks using handcrafted HRMs.

**Learning of Non-Flat HRMs.** Figure 3 shows the LHRM learning curves for CRAFTWORLD (FRL) and WATERWORLD (WD). These settings are the most challenging due to the inclusion of dead-ends since (i) they hinder the observation of goal examples in level 1 tasks using random walks, (ii) the RMs must include rejecting states, (iii) formula options must avoid dead-ends, and (iv) call options must avoid invoking options leading to rejecting states. In line with the curriculum method, LHRM does not start learning a task of a given level until tasks in previous levels are mastered. The convergence for high-level tasks is often fast due to the reuse of lower level HRMs and policies.

The average time (in seconds) exclusively spent on learning *all* HRMs is 1009.8 (122.3) for OP, 1622.6 (328.7) for OPL, 1031.6 (150.3) for FR, 1476.8 (175.3) for FRL, 35.4 (2.0) for WOD, and 67.0 (6.2) for WD. Including dead-ends (OPL, FRL, WD) incurs longer executions since (i) there is one more proposition, (ii) there are edges to the rejecting state(s), and (iii) there are dead-end traces to cover. We observe that the complexity of learning an HRM does not necessarily correspond with the task complexity (e.g., the times for OP and FRL are similar). Learning in WATERWORLD is faster than in CRAFTWORLD since the RMs have fewer states and there are fewer callable RMs.

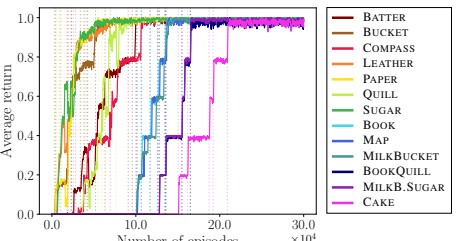 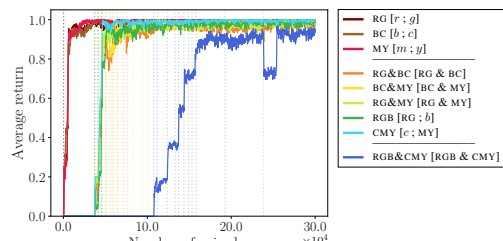

Figure 3: LHRM learning curves for CRAFTWORLD (FRL) and WATERWORLD (WD). The legend in WATERWORLD separates tasks by level, and the subtask order (in brackets) follows that introduced in Table 1. The dotted vertical lines correspond to episodes in which an HRM is learned.

By *restricting* the callable RMs to those required by the HRM (e.g., using just PAPER and LEATHER RMs to learn BOOK's HRM), there are fewer ways to label the edges of the induced root. Learning is 5-7× faster using 20% fewer calls to the learner (i.e., fewer examples) in CRAFTWORLD, and 1.5× faster in WATERWORLD; hence, HRM learning becomes less scalable as the number of tasks and levels grows. This is an instance of the *utility* problem (Minton, 1988). Refining the callable RMs set 'a priori' to speed up HRM learning is a direction for future work.

We evaluate the performance of *exploration with options* using the number of episodes needed to collect the $\rho$ goal traces for a given task since the activation of its level. Intuitively, the agent will rarely move far from a region of the state space using primitive actions only, thus taking more time to collect the traces; in contrast, options enable the agent to explore the state space more efficiently. In the FRL setting of CRAFTWORLD, we observe that using primitive actions requires 128.1× more episodes than options in MILKBUCKET, the only level 2 task for which $\rho$ traces are collected (although in just 2/5 runs). Likewise, primitive actions take 20.8× and 7.7× more episodes in OPL and WD respectively. In OP and WOD options are not beneficial since episodes are relatively long (1,000 steps), there are no dead-ends and it is easy to observe the different propositions.

Finally, we observe that using a *single goal trace* to learn the first HRMs ($\rho = \rho_s = 1$) incurs timeouts across all CRAFTWORLD settings, thus showing the value of using many short traces instead.

**Learning Flat HRMs.** Learning a flat HRM is often less scalable than learning a non-flat equivalent since (i) previously learned HRMs cannot be reused, and (ii) the flat HRM usually has more states and edges (as shown in Theorem 2, the growth can be exponential). We compare the performance of learning (from interaction) a non-flat HRM using LHRM with that of an equivalent flat HRM using LHRM, DeepSynth (Hasanbeig et al., 2021), LRM (Toro Icarte et al., 2019) and JIRP (Xu et al., 2020). Akin to LHRM, JIRP induces RMs with explicit accepting states, while DeepSynth and LRM do not. We use OP and WOD instances for CRAFTWORLD and WATERWORLD respectively.

The non-flat HRM for MILKBUCKET is learned in 1.5 (0.2) seconds, whereas the flat HRMs take longer to learn: 3.2 (0.6) w/LHRM, 325.6 (29.7) w/DeepSynth, 347.5 (64.5) w/LRM and 17.1 (5.5) w/JIRP. LHRM and JIRP learn minimal RMs, hence producing the same RM consisting of 4 states and 3 edges. DeepSynth and LRM do not learn a minimal RM but one that is good at predicting the next possible label given the current one. In domains like ours where propositions can be observed anytime (i.e., without temporal dependencies between them), these methods tend to 'overfit' the input traces and produce large outputs that barely reflect the task's structure, e.g. DeepSynth learns RMs with 13.4 (0.4) states and 93.2 (1.7) edges. In contrast, methods learning minimal machines exclusively from observable traces may suffer from *overgeneralization* (Angluin, 1980) in other domains (e.g., with temporally-dependent propositions). In more complex tasks such as BOOK, LHRM learns the non-flat HRM (see Figure 1b) in 191.2 (36.4) seconds, whereas methods learning the flat HRM (see Figure 1c) usually time out or, in the case of DeepSynth, learn bigger representations.

The performance of DeepSynth, LRM and JIRP is poor in WATERWORLD since they all learn RMs whose edges are labeled with proposition sets instead of formulas, unlike LHRM; thus, the RMs may require exponentially more edges, motivating the use of formulas for abstraction. For instance, the flat HRM for RG requires 64 edges instead of 2, and only LHRM and JIRP can learn it on time. All flat HRM learners time out in RG&BC, whereas the non-flat HRM is learned in 4.5 (0.3) seconds.

## 7    RELATED WORK

**RMs and Composability.** Our RMs differ from the original ones (Toro Icarte et al., 2018; 2022) in that (i) an RM can call other RMs, (ii) there are explicit accepting and rejecting states (Xu et al., 2020; Furelos-Blanco et al., 2021), and (iii) transitions are labeled with propositional logic formulas instead of proposition sets (Furelos-Blanco et al., 2021). Recent works *derive* RMs (and similar FSMs) from formal language specifications (Camacho et al., 2019; Araki et al., 2021) and expert demonstrations (Camacho et al., 2021), or *learn* them from experience using discrete optimization (Toro Icarte et al., 2019), SAT solving (Xu et al., 2020), active learning (Gaon & Brafman, 2020; Xu et al., 2021), state-merging (Xu et al., 2019; Gaon & Brafman, 2020), program synthesis (Hasanbeig et al., 2021) or inductive logic programming (Furelos-Blanco et al., 2021). Prior ways of composing RMs include (i) merging the state and reward transition functions (De Giacomo et al., 2020), and (ii) encoding a multi-agent task using an RM, decomposing it into one RM per agent and executing them in parallel (Neary et al., 2021). Task composability has also been modeled using subtask sequences called sketches (Andreas et al., 2017), context-free grammars defining a subset of English (Chevalier-Boisvert et al., 2019), formal languages (Jothimurugan et al., 2019; León et al., 2020; Wang et al., 2020) and logic-based algebras (Nangue Tasse et al., 2020).

**Hierarchical RL.** Our method for exploiting HRMs resembles a hierarchy of DQNs (Kulkarni et al., 2016). Akin to option discovery methods, LHRM induces a set of options from experience. LHRM requires a set of propositions and tasks, which bound the number of discoverable options; similarly, some of these methods impose an explicit bound (Bacon et al., 2017; Machado et al., 2017). LHRM requires each task to be solved at least once before learning an HRM (and, hence, options), just like other methods (McGovern & Barto, 2001; Stolle & Precup, 2002). The problem of discovering options for exploration has been considered before (Bellemare et al., 2016; Machado et al., 2017; Jinnai et al., 2019; Dabney et al., 2021). While our options are not explicitly discovered for exploration, we leverage them to find goal traces in new tasks. Levy et al. (2019) learn policies from multiple hierarchical levels in parallel by training each level as if the lower levels were optimal; likewise, we train call option policies from experiences where invoked options achieve their goal.

HRMs are close to hierarchical abstract machines (HAMs; Parr & Russell, 1997) since both are hierarchies of FSMs; however, there are two core differences. First, HAMs do not have reward transition functions. Second, (H)RMs decouple the traversal from the policies, i.e. independently of the agent's choices, the (H)RM is followed; thus, an agent exploiting an (H)RM must be able to interrupt its choices (see Section 4). While HAMs do not support interruption, Programmable HAMs (Andre & Russell, 2000) extend HAMs to support it along with other program-like features. Despite the resemblance, there are few works on learning HAMs (Leonetti et al., 2012) and there are many on learning RMs, hence showing (H)RMs are more amenable (yet expressive) to learning.

**Curriculum Learning.** Pierrot et al. (2019) learn hierarchies of neural programs given the level of each program, akin to our RMs' height; likewise, Andreas et al. (2017) prioritize tasks consisting of fewer high-level steps. The 'online' method by Matiisen et al. (2020) also keeps an estimate of each task's average return, but it is not applied in an HRL scenario. Wang et al. (2020) learn increasingly complex temporal logic formulas leveraging previously learned formulas using a set of templates.

## 8    CONCLUSIONS

We have here proposed (1) HRMs, a *formalism* that composes RMs in a hierarchy by enabling them to call each other, (2) an HRL method that *exploits* the structure of an HRM, and (3) a curriculum-based method for *learning* a collection of HRMs from traces. Non-flat HRMs have significant advantages over their flat equivalents. Theoretically, we have proved that the flat equivalent of a given HRM can have exponentially more states and edges. Empirically, we have shown that (i) our HRL method converges faster given a non-flat HRM instead of a flat equivalent one, and (ii) in line with the theory, learning an HRM is feasible in cases where its flat equivalent is not.

LHRM assumes that the proposition set is known, dead-end indicators are shared across tasks, there is a fixed set of tasks and the height for each HRM is provided. Relaxing these *assumptions* by forming the propositions from raw data, conditioning policies to dead-ends, and letting the agent propose its own composable tasks are promising directions for future work. Other interesting extensions include *non-episodic* settings and methods for learning *globally optimal* policies over HRMs.

## REPRODUCIBILITY

To make our work more understandable and reproducible, we provide pseudo-code, proofs and examples throughout the paper. We here outline the content covered in the appendices that help with reproducibility:

**Appendix A**  All the intermediate steps for the hierarchy traversal example in Section 3.

**Appendix B**  Proofs for Theorems 1 and 2.

**Appendix C**  The implementation details (including pseudo-code) for the policy learning algorithm outlined in Section 4. We include running examples of the algorithm to aid understanding.

**Appendix D**  The implementation details for the curriculum learning mechanism introduced in Section 5.

**Appendix E**  How the root of an HRM is learned using the ILASP inductive logic programming system, including proofs of correctness.

**Appendix F**  Details on the experiments described in Section 6, such as computational resources, implementation of the domains, training and evaluation details (e.g., hyperparameters) and extended results (e.g., tables on which the results in the main paper are based).

**Appendix G**  Handcrafted HRMs for the tasks used in the experiments.

We plan to release the code if the paper is accepted.

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

## A    Hierarchy Traversal Example

The HRM in Figure 1b accepts trace $\lambda = \langle\{⚒\}, \{🐀\}, \{\}, \{🐍\}, \{🐀\}, \{🐦\}\rangle$, whose traversal is $H(\lambda) = \langle v_0, v_1, v_2, v_3, v_4, v_5, v_6\rangle$, where:

$$v_0 = \langle M_0, u_0^0, \top, [] \rangle,$$

$$v_1 = \delta_H(v_0, \{⚒\})$$
$$= \delta_H(\langle M_0, u_0^0, \top, [] \rangle, \{⚒\})$$
$$= \delta_H(\langle M_1, u_1^0, \neg🐍, [\langle u_0^0, u_0^1, M_0, M_1, \neg🐍, \top\rangle] \rangle, \{⚒\})$$
$$= \delta_H(\langle M_\top, u_\top^0, \neg🐍 \wedge ⚒, [\langle u_0^0, u_0^1, M_0, M_1, \neg🐍, \top\rangle, \langle u_1^0, u_1^1, M_1, M_\top, ⚒, \neg🐍\rangle] \rangle, \{⚒\})$$
$$= \delta_H(\langle M_1, u_1^1, \top, [\langle u_0^0, u_0^1, M_0, M_1, \neg🐍, \top\rangle] \rangle, \bot)$$
$$= \langle M_1, u_1^1, \top, [\langle u_0^0, u_0^1, M_0, M_1, \neg🐍, \top\rangle] \rangle,$$

$$v_2 = \delta_H(v_1, \{🐀\})$$
$$= \delta_H(\langle M_1, u_1^1, \top, [\langle u_0^0, u_0^1, M_0, M_1, \neg🐍, \top\rangle] \rangle, \{🐀\})$$
$$= \delta_H(\langle M_\top, u_\top^0, 🐀, [\langle u_0^0, u_0^1, M_0, M_1, \neg🐍, \top\rangle, \langle u_1^1, u_1^A, M_1, M_\top, 🐀, \top\rangle] \rangle, \{🐀\})$$
$$= \delta_H(\langle M_1, u_1^A, \top, [\langle u_0^0, u_0^1, M_0, M_1, \neg🐍, \top\rangle] \rangle, \bot)$$
$$= \delta_H(\langle M_0, u_0^1, \top, [] \rangle, \bot)$$
$$= \langle M_0, u_0^1, \top, [] \rangle,$$

$$v_3 = \delta_H(v_2, \{\})$$
$$= \delta_H(\langle M_0, u_0^1, \top, [] \rangle, \{\})$$
$$= \langle M_0, u_0^1, \top, [] \rangle,$$

$$v_4 = \delta_H(v_3, \{🐍\})$$
$$= \delta_H(\langle M_0, u_0^1, \top, [] \rangle, \{🐍\})$$
$$= \delta_H(\langle M_2, u_2^0, \top, [\langle u_0^1, u_0^3, M_0, M_2, \top, \top\rangle] \rangle, \{🐍\})$$
$$= \delta_H(\langle M_\top, u_\top^0, 🐍, [\langle u_0^1, u_0^3, M_0, M_2, \top, \top\rangle, \langle u_2^0, u_2^1, M_2, M_\top, 🐍, \top\rangle] \rangle, \{🐍\})$$
$$= \delta_H(\langle M_2, u_2^1, \top, [\langle u_0^1, u_0^3, M_0, M_2, \top, \top\rangle] \rangle, \bot)$$
$$= \langle M_2, u_2^1, \top, [\langle u_0^1, u_0^3, M_0, M_2, \top, \top\rangle] \rangle,$$

$$v_5 = \delta_H(v_4, \{🐀\})$$
$$= \delta_H(\langle M_2, u_2^1, \top, [\langle u_0^1, u_0^3, M_0, M_2, \top, \top\rangle] \rangle, \{🐀\})$$
$$= \delta_H(\langle M_\top, u_\top^0, 🐀, [\langle u_0^1, u_0^3, M_0, M_2, \top, \top\rangle, \langle u_2^1, u_2^A, M_2, M_\top, 🐀, \top\rangle] \rangle, \{🐀\})$$
$$= \delta_H(\langle M_2, u_2^A, \top, [\langle u_0^1, u_0^3, M_0, M_2, \top, \top\rangle] \rangle, \bot)$$
$$= \delta_H(\langle M_0, u_0^3, \top, [] \rangle, \bot)$$
$$= \langle M_0, u_0^3, \top, [] \rangle,$$

$$v_6 = \delta_H(v_5, \{🐦\})$$
$$= \delta_H(\langle M_0, u_0^3, \top, [] \rangle, \{🐦\})$$
$$= \delta_H(\langle M_\top, u_\top^0, 🐦, [\langle u_0^3, u_0^A, M_0, M_\top, 🐦, \top\rangle] \rangle, \{🐦\})$$
$$= \delta_H(\langle M_0, u_0^A, \top, [] \rangle, \bot)$$
$$= \langle M_0, u_0^A, \top, [] \rangle.$$

Example 1 shows the traversal but omits the intermediate applications of the hierarchical transition function $\delta_H$.

## B    Equivalence to Flat Hierarchies of Reward Machines

In this section, we prove the theorems introduced in Section 3 regarding the equivalence of an arbitrary HRM to a flat HRM.

B.1   Proof of Theorem 1

We formally show that any HRM can be transformed into an equivalent one consisting of a single non-leaf RM. The latter HRM type is called *flat* since there is a single hierarchy level.

**Definition 6.** Given an HRM $H = \langle \mathcal{M}, M_r, \mathcal{P} \rangle$, a constituent RM $M_i \in \mathcal{M}$ is *flat* if its height $h_i$ is 1.

**Definition 7.** An HRM $H = \langle \mathcal{M}, M_r, \mathcal{P} \rangle$ is *flat* if the root RM $M_r$ is flat.

We now define what it means for two HRMs to be equivalent. This definition is based on that used in automaton theory (Sipser, 1997).

**Definition 8.** Given a set of propositions $\mathcal{P}$ and a labeling function $l$, two HRMs $H = \langle \mathcal{M}, M_r, \mathcal{P} \rangle$ and $H' = \langle \mathcal{M}', M_r', \mathcal{P} \rangle$ are *equivalent* if for any label trace $\lambda$ one of the following conditions holds: (i) both HRMs accept $\lambda$, (ii) both HRMs reject $\lambda$, or (iii) neither of the HRMs accepts or rejects $\lambda$.

We now have all the required definitions to prove Theorem 1, which is restated below.

**Theorem 1.** *Given an HRM $H$, there exists an equivalent flat HRM $\bar{H}$.*

To prove the theorem, we introduce an algorithm for flattening any HRM. Without loss of generality, we work on the case of an HRM with two hierarchy levels; that is, an HRM consisting of a root RM that calls flat RMs. Note that an HRM with an arbitrary number of levels can be flattened by considering the RMs in two levels at a time. We start flattening RMs in the second level (i.e., with height 2), which use RMs in the first level (by definition, these are already flat), and once the second level RMs are flat, we repeat the process with the levels above until the root is reached. This process is applicable since, by assumption, the hierarchies do not have cyclic dependencies nor recursion. For simplicity, we use the MDP reward assumption made in Section 2, i.e. the reward transition function of any RM $M_i$ is $r_i(u, u') = \mathbb{1}[u \notin \mathcal{U}_i^A \wedge u' \in \mathcal{U}_i^A]$ like in Section 4. However, the proof below could be adapted to arbitrary definitions of $r_i(u, u')$.

**Preliminary Transformation Algorithm.**   Before proving Theorem 1, we introduce an intermediate step that transforms a flat HRM into an equivalent one that takes contexts with which it may be called into account. Remember that a call to an RM is associated with a context. In the case of two-level HRMs such as the ones we are considering in this flattening process, the context and the exit condition from the called flat RM must be satisfied. Crucially, the context must only be satisfied at the time of the call; that is, it only lasts for a single transition. Therefore, if we revisit the initial state of the called RM by taking an edge to it, the context should not be checked anymore.

To make the need for this transformation clearer, we use the HRM illustrated in Figure 4a. The flattening algorithm described later embeds the called RM into the caller one; crucially, the context of the call is taken into account by putting it in conjunction with the outgoing edges from the initial state of the called RM.[2] Figure 4b is a flat HRM obtained using the flattening algorithm; however, it does not behave like the HRM in Figure 4a. Following the definition of the hierarchical transition function $\delta_H$, the context of a call only lasts for a single transition in the called RM in Figure 4a (i.e., $a \wedge \neg c$ is only checked when $M_1$ is started), but the context is kept permanently in Figure 4b, which is problematic if we go back to the initial state at some point. We later come back to this example after presenting the transformation algorithm.

To deal with the situation above, we need to transform an RM to ensure that contexts are only checked once from the initial state. We describe this transformation as follows. Given a flat HRM $H = \langle \mathcal{M}, M_r, \mathcal{P} \rangle$ with root $M_r = \langle \mathcal{U}_r, \mathcal{P}, \varphi_r, r_r, u_r^0, \mathcal{U}_r^A, \mathcal{U}_r^R \rangle$, we construct a new HRM $H' = \langle \mathcal{M}', M_r', \mathcal{P} \rangle$ with root $M_r' = \langle \mathcal{U}_r', \mathcal{P}, \varphi_r', r_r', u_r^0, \mathcal{U}_r^A, \mathcal{U}_r^R \rangle$ such that:

- $\mathcal{U}_r' = \mathcal{U}_r \cup \{\hat{u}_r^0\}$, where $\hat{u}_r^0$ plays the role of the initial state after the first transition is taken.

- The state transition function $\varphi_r'$ is built by copying $\varphi_r$ and applying the following changes:

  1. Remove the edges to the actual initial state from any state $v \in \mathcal{U}_r'$: $\varphi_r'(v, u_r^0, M_\top) = \bot$. Note that since the RM is flat, the only callable RM is the leaf $M_\top$.

  2. Add edges to the dummy initial state $\hat{u}_r^0$ from all states $v \in \mathcal{U}_r'$ that had an edge to the actual initial state: $\varphi_r'(v, \hat{u}_r^0, M_\top) = \varphi_r(v, u_r^0, M_\top)$.

_______________

[2]We refer the reader to the 'Flattening Algorithm' description introduced later for specific details.

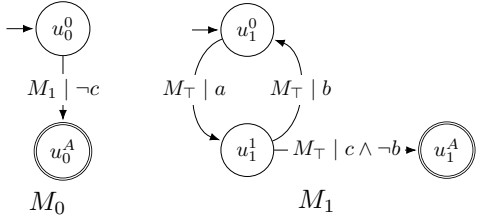
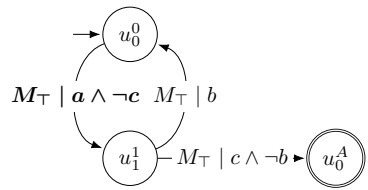

$M_0$      $M_1$

(a) Original HRM with root $M_0$.      (b) Flattened HRM without transforming $M_1$.

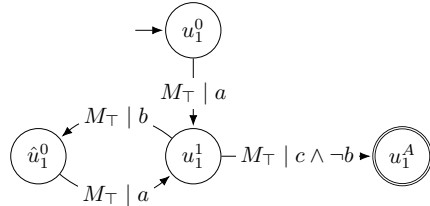
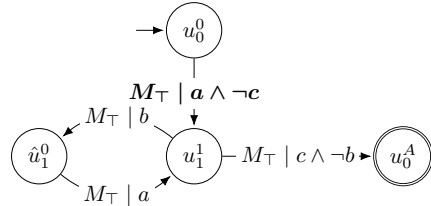

(c) Transformed $M_1$ from (a).      (d) Flattened HRM after transforming $M_1$.

Figure 4: Example to justify the need for the preliminary transformation algorithm.

3. Add edges from the dummy initial state $\hat{u}_r^0$ to all those states $v \in \mathcal{U}_r'$ that the actual initial state $u_r^0$ points to: $\varphi_r'(\hat{u}_r^0, v, M_\top) = \varphi_r'(u_r^0, v, M_\top)$.

- The reward transition function $r_r'(u, u') = \mathbb{1}[u \notin \mathcal{U}_r^A \wedge u' \in \mathcal{U}_r^A]$ is defined as stated at the beginning of the section.

The HRM $H'$ is such that $\mathcal{M}' = \{M_r', M_\top\}$. Note that this transformation is only required in HRMs where the RMs have initial states with incoming edges.

We now prove that this transformation is correct; that is, the HRMs are equivalent. There are two cases depending on whether the initial state has incoming edges or not. First, if the initial state $u_r^0$ does not have incoming edges, step 1 does not remove any edges going to $u_r^0$, and step 2 does not add any edges going to $\hat{u}_r^0$, making it unreachable. Even though edges from $\hat{u}_r^0$ to other states may be added, it is irrelevant since it is unreachable. Therefore, we can safely say that in this case, the transformed HRM is equivalent to the original one. Second, if the initial state has incoming edges, we prove equivalence by examining the traversals $H(\lambda)$ and $H'(\lambda)$ for the original HRM $H = \langle \mathcal{M}, M_r, \mathcal{P} \rangle$ and the transformed one $H' = \langle \mathcal{M}', M_r', \mathcal{P} \rangle$ given a generic label trace $\lambda = \langle \mathcal{L}_0, \ldots, \mathcal{L}_n \rangle$. By construction, both $H(\lambda)$ and $H'(\lambda)$ will be identical until reaching a state $w$ with an outgoing transition to $u_r^0$ in the case of $H$ and the dummy initial state $\hat{u}_r^0$ in the case of $H'$. More specifically, upon reaching $w$ and satisfying an outgoing formula to the aforementioned states, the traversals are:

$$H(\lambda) = \langle \langle M_r, u_r^0, \top, [] \rangle, \ldots, \langle M_r, w, \top, [] \rangle \rangle,$$
$$H'(\lambda) = \langle \langle M_r', u_r^0, \top, [] \rangle, \ldots, \langle M_r', w, \top, [] \rangle \rangle.$$

By construction, state $w$ is in both HRMs, and both of the aforementioned transitions from this state are associated with the same formula, i.e. $\varphi_r(w, u_r^0, M_\top) = \varphi_r'(w, \hat{u}_r^0, M_\top)$. Therefore, if one of them is satisfied, the other will be too, and the traversals will become:

$$H(\lambda) = \langle \langle M_r, u_r^0, \top, [] \rangle, \ldots, \langle M_r, w, \top, [] \rangle, \langle M_r, u_r^0, \top, [] \rangle \rangle,$$
$$H'(\lambda) = \langle \langle M_r', u_r^0, \top, [] \rangle, \ldots, \langle M_r', w, \top, [] \rangle, \langle M_r', \hat{u}_r^0, \top, [] \rangle \rangle.$$

We stay in $u_r^0$ and $\hat{u}_r^0$ until a transition to a state $w'$ is satisfied. By construction, $w'$ is in both HRMs and the same formula is satisfied, i.e., $\varphi_r(u_r^0, w', M_\top) = \varphi_r'(\hat{u}^0, w', M_\top)$. The hierarchy traversals then become:

$$H(\lambda) = \langle \langle M_r, u_r^0, \top, [] \rangle, \ldots, \langle M_r, w, \top, [] \rangle, \langle M_r, u_r^0, \top, [] \rangle, \ldots, \langle M_r, u_r^0, \top, [] \rangle, \langle M_r, w', \top, [] \rangle \rangle,$$
$$H'(\lambda) = \langle \langle M_r', u_r^0, \top, [] \rangle, \ldots, \langle M_r', w, \top, [] \rangle, \langle M_r', \hat{u}_r^0, \top, [] \rangle, \ldots, \langle M_r', \hat{u}_r^0, \top, [] \rangle, \langle M_r', w', \top, [] \rangle \rangle.$$

From here both traversals will be the same until transitions to $u_r^0$ and $\hat{u}_r^0$ are respectively satisfied again (if any) in $H$ and $H'$. Clearly, the only change in $H(\lambda)$ with respect to $H'(\lambda)$ (except for the different roots) is that the hierarchy states of the form $\langle M_r', \hat{u}_r^0, \top, [] \rangle$ in the latter appear as $\langle M_r, u_r^0, \top, [] \rangle$ in the former. We now check if the equivalence conditions from Definition 8 hold:

- If $H(\lambda)$ ends with state $u_r^0$, $H'(\lambda)$ ends with state $\hat{u}_r^0$ following the reasoning above. By construction, neither of these states is accepting or rejecting; therefore, neither of these HRMs accepts or rejects $\lambda$.

- If $H(\lambda)$ ends with state $w$, $H'(\lambda)$ will also end with this state following the reasoning above. Therefore, if $w$ is an accepting state, both HRMs accept $\lambda$; if $w$ is a rejecting state, both HRMs reject $\lambda$; and if $w$ is not an accepting or rejecting state, neither of the HRMs accepts or rejects $\lambda$.

Since all equivalence conditions are satisfied for any trace $\lambda$, $H$ and $H'$ are equivalent.

Figure 4c exemplifies the output of the transformation algorithm given $M_1$ in Figure 4a as input, whereas Figure 4d is the output of the flattening algorithm discussed next, which properly handles the context unlike the HRM in Figure 4b.

**Flattening Algorithm.** We describe the algorithm for flattening an HRM. As previously stated, we assume without loss of generality that the HRM to be flattened consists of two hierarchy levels (i.e., the root calls flat RMs). We also assume that the flat RMs have the form produced by the previously presented transformation algorithm.

Given an HRM $H = \langle \mathcal{M}, M_r, \mathcal{P} \rangle$ with root $M_r = \langle \mathcal{U}_r, \mathcal{P}, \varphi_r, r_r, u_r^0, \mathcal{U}_r^A, \mathcal{U}_r^R \rangle$, we build a flat RM $\bar{M}_r = \langle \bar{\mathcal{U}}_r, \mathcal{P}, \bar{\varphi}_r, \bar{r}_r, \bar{u}_r^0, \bar{\mathcal{U}}_r^A, \bar{\mathcal{U}}_r^R \rangle$ using the following steps:

1. Copy the sets of states and initial state from $M_r$ (i.e., $\bar{\mathcal{U}}_r = \mathcal{U}_r$, $\bar{u}_r^0 = u_r^0$, $\bar{\mathcal{U}}_r^A = \mathcal{U}_r^A$, $\bar{\mathcal{U}}_r^R = \mathcal{U}_r^R$).

2. Loop through the non-false entries of the transition function $\varphi_r$ and decide what to copy. That is, for each triplet $(u, u', M_j)$ where $u, u' \in \mathcal{U}_r$ and $M_j \in \mathcal{M}$ such that $\varphi_r(u, u', M_j) \neq \bot$:

   (a) If $M_j = M_\top$ (i.e., the called RM is the leaf), we copy the transition: $\bar{\varphi}_r(u, u', M_\top) = \varphi_r(u, u', M_\top)$.

   (b) If $M_j \neq M_\top$, we embed the transition function of $M_j = \langle \mathcal{U}_j, \mathcal{P}, \varphi_j, r_j, u_j^0, \mathcal{U}_j^A, \mathcal{U}_j^R \rangle$ into $\bar{M}_r$. Remember that $M_j$ is flat. To do so, we run the following steps:

      i. Update the set of states by adding all non-initial and non-accepting states from $M_j$. Similarly, the set of rejecting states is also updated by adding all rejecting states of the called RM. The initial and accepting states from $M_j$ are unimportant: their roles are played by $u$ and $u'$ respectively. In contrast, the rejecting states are important since, by assumption, they are global. Note that the added states $v$ are renamed to $v_{u,u',j}$ in order to take into account the edge being embedded: if the same state $v$ was reused for another edge, then we would not be able to distinguish them.

      $$\bar{\mathcal{U}}_r = \bar{\mathcal{U}}_r \cup \left\{ v_{u,u',j} \mid v \in \left( \mathcal{U}_j \setminus \left( \{u_j^0\} \cup \mathcal{U}_j^A \right) \right) \right\},$$
      $$\bar{\mathcal{U}}_r^R = \bar{\mathcal{U}}_r^R \cup \left\{ v_{u,u',j} \mid v \in \mathcal{U}_j^R \right\}.$$

      ii. Embed the transition function $\varphi_j$ of $M_j$ into $\bar{\varphi}_r$. Since $M_j$ is flat, we can make copies of the transitions straightaway: the only important thing is to check whether these transitions involve initial or accepting states which, as stated before, are going to be replaced by $u$ and $u'$ accordingly. Given a triplet $(v, w, M_\top)$ such that $v, w \in \mathcal{U}_j$ and for which $\varphi_j(v, w, M_\top) = \phi$ and $\phi \neq \bot$ we update $\bar{\varphi}_r$ as follows:[3]

---

[3]We do not to cover the case where $v$ is an accepting state since, by assumption, there are no outgoing transitions from it. In the case of rejecting states, we keep all of them as explained in the previous case and, therefore, there are no substitutions to be made. We also do not cover the case where $w = u_j^0$ since the input flat machines never have edges to their initial states, but to the dummy initial state.

A. If $v = u_j^0$ and $w \notin \mathcal{U}_j^A$, then $\bar{\varphi}_r(u, w_{u,u',j}, M_\top) = \text{DNF}(\phi \wedge \varphi_r(u, u', M_j))$. The initial state of $M_j$ has been substituted by $u$, we use the clone of $w$ associated with the call ($w_{u,u',j}$), and append the context of the call to $M_j$ to the formula $\phi$.

B. If $v = u_j^0$ and $w \in \mathcal{U}_j^A$, then $\bar{\varphi}_r(u, u', M_\top) = \text{DNF}(\phi \wedge \varphi_r(u, u', M_j))$. Like the previous case but performing two substitutions: $u$ replaces $v$ and $u'$ replaces $w$. The context is appended since it is a transition from the initial state of $M_j$.

C. If $v \neq u_j^0$ and $w \in \mathcal{U}_j^A$, then $\bar{\varphi}_r(v_{u,u',j}, u', M_\top) = \phi$. We substitute the accepting state $w$ by $u'$, and use the clone of $v$ associated with the call ($v_{u,u',j}$). This time the call's context is not added since $v$ is not the initial state of $M_j$.

D. If none of the previous cases holds, there are no substitutions to be made nor contexts to be taken into account. Hence, $\bar{\varphi}_r(v_{u,u',j}, w_{u,u',j}, M_\top) = \phi$. We just use the clones of $v$ and $w$ corresponding to the call ($v_{u,u',j}$ and $w_{u,u',j}$).

3. We apply the transformation algorithm we described before, and form a new flat HRM $\bar{H} = \langle \{\bar{M}_r, M_\top\}, \bar{M}_r, \mathcal{P} \rangle$ with the flattened (and transformed) root $\bar{M}_r$.

The reward transition function $r_r'(u, u') = \mathbb{1}[u \notin \bar{\mathcal{U}}_r^A \wedge u' \in \bar{\mathcal{U}}_r^A]$ is defined as stated at the beginning of the section. Note that $u$ might not necessarily be a state of the non-flat root, but derived from an RM with lower height.

We now have everything to prove the previous theorem. Without loss of generality and for simplicity, we assume that the transformation algorithm has not been applied over the flattened root (we have already shown that the transformation produces an equivalent flat machine).

**Theorem 1.** *Given an HRM $H$, there exists an equivalent flat HRM $\bar{H}$.*

*Proof.* Let us assume that an HRM $\bar{H} = \langle \bar{\mathcal{M}}, \bar{M}_r, \mathcal{P} \rangle$, where $\bar{M}_r = \langle \bar{\mathcal{U}}_r, \mathcal{P}, \bar{\varphi}_r, \bar{r}_r, \bar{u}_r^0, \bar{\mathcal{U}}_r^A, \bar{\mathcal{U}}_r^R \rangle$, is a flat HRM that results from applying the flattening algorithm on an HRM $H = \langle \mathcal{M}, M_r, \mathcal{P} \rangle$, where $M_r = \langle \mathcal{U}_r, \mathcal{P}, \varphi_r, r_r, u_r^0, \mathcal{U}_r^A, \mathcal{U}_r^R \rangle$. For these HRMs to be equivalent, any label trace $\lambda = \langle \mathcal{L}_0, \ldots, \mathcal{L}_n \rangle$ must satisfy one of the conditions in Definition 8. To prove the equivalence, we examine the hierarchy traversals $H(\lambda)$ and $\bar{H}(\lambda)$ given a generic label trace $\lambda$.

Let $u \in \mathcal{U}_r$ be a state in the root $M_r$ of $H$ and let $\varphi_r(u, u', M_\top)$ be a satisfied transition from that state. By construction, $u$ is also in the root $\bar{M}_r$ of the flat hierarchy $\bar{H}$, and $\bar{M}_r$ has an identical transition $\bar{\varphi}_r(u, u', M_\top)$, which must also be satisfied. If the hierarchy states are $\langle M_r, u, \top, [] \rangle$ and $\langle \bar{M}_r, u, \top, [] \rangle$ for $H$ and $\bar{H}$ respectively, then the next hierarchy states upon application of $\delta_H$ will be $\langle M_r, u', \top, [] \rangle$ and $\langle \bar{M}_r, u', \top, [] \rangle$. Therefore, both HRMs behave equivalently when calls to the leaf RM are made.

We now examine what occurs when a non-leaf RM is called in $H$. Let $\varphi_r(u, u', M_j)$ be a satisfied transition in $M_r$, and let $\varphi_j(u_j^0, w, M_\top)$ be a satisfied transition from $M_j$'s initial state. By construction, $\bar{M}_r$ contains a transition whose associated formula is the conjunction of the previous two, i.e. $\varphi_r(u, u', M_j) \wedge \varphi_j(u_j^0, w, M_\top)$. Now, the hierarchy traversals will be different depending on $w$:

- If $w \notin \mathcal{U}_j^A$ (i.e., $w$ is not an accepting state of $M_j$), by construction, $\bar{M}_r$ contains the transition $\bar{\varphi}_r(u, w_{u,u',j}, M_\top) = \varphi_r(u, u', M_j) \wedge \varphi_j(u_j^0, w, M_\top)$. If the current hierarchy states are (the equivalent) $\langle M_r, u, \top, [] \rangle$ and $\langle \bar{M}_r, u, \top, [] \rangle$ for $H$ and $\bar{H}$, then the next hierarchy states upon application of $\delta_H$ are $\langle M_j, w, \top, [\langle u, u', M_r, M_j, \varphi_r(u, u', M_j), \top \rangle] \rangle$ and $\langle \bar{M}_r, w_{u,u',j}, \top, [] \rangle$. These hierarchy states are equivalent since $w_{u,u',j}$ is a clone of $w$ that saves all the call information (i.e., a call to machine $M_j$ for transitioning from $u$ to $u'$).

- If $w \in \mathcal{U}_j^A$ (i.e., $w$ is an accepting state of $M_j$), by construction, $\bar{M}_r$ contains the transition $\bar{\varphi}_r(u, u', M_\top) = \varphi_r(u, u', M_j) \wedge \varphi_j(u_j^0, w, M_\top)$. If the current hierarchy states are (the equivalent) $\langle M_r, u, \top, [] \rangle$ and $\langle \bar{M}_r, u, \top, [] \rangle$ for $H$ and $\bar{H}$, then the next hierarchy states upon application of $\delta_H$ are $\langle M_r, u', \top, [] \rangle$ and $\langle \bar{M}_r, u', \top, [] \rangle$. These hierarchy states are clearly equivalent since the machine states are exactly the same.

We now check the case in which we are inside a called RM. Let $\varphi_r(u, u', M_j)$ be the transition that caused $H$ to start running $M_j$, and let $\varphi_j(v, w, M_\top)$ be a satisfied transition within $M_j$ such that

$v \neq u_j^0$. By construction, $\bar{M}_r$ contains a transition associated with the same formula $\varphi_j(v, w, M_\top)$. The hierarchy traversals vary depending on $w$:

- If $w \notin \mathcal{U}_j^A$ (i.e., $w$ is not an accepting state of $M_j$), by construction, $\bar{M}_r$ contains the transition $\bar{\varphi}_r(v_{u,u',j}, w_{u,u',j}, M_\top) = \varphi_j(v, w, M_\top)$. For the transition to be taken in $H$, the hierarchy state must be $\langle M_j, v, \top, [\langle u, u', M_r, M_j, \varphi_r(u, u', M_j), \top \rangle] \rangle$, whereas in $\bar{H}$ it will be $\langle \bar{M}_r, v_{u,u',j}, \top, [] \rangle$. These hierarchy states are clearly equivalent: $v_{u,u',j}$ is a clone of $v$ that saves all information related to the call being made (the called machine, and the starting and resulting states in the transition). Upon application of $\delta_H$, the hierarchy states will remain equivalent: $\langle M_j, w, \top, [\langle u, u', M_r, M_j, \varphi_r(u, u', M_j), \top \rangle] \rangle$ and $\langle \bar{M}_r, w_{u,u',j}, \top, [] \rangle$ (again $w_{u,u',j}$ saves all the call information, just like the stack).

- If $w \in \mathcal{U}_j^A$ (i.e., $w$ is an accepting state of $M_j$), by construction, $\bar{M}_r$ contains the transition $\bar{\varphi}_r(v_{u,u',j}, u', M_\top) = \varphi_j(v, w, M_\top)$. This case corresponds to that where control is returned to the calling RM. Like in the previous case, for the transition to be taken in $H$, the hierarchy state must be $\langle M_j, v, \top, [\langle u, u', M_r, M_j, \varphi_r(u, u', M_j), \top \rangle] \rangle$, whereas in $\bar{H}$ it will be $\langle \bar{M}_r, v_{u,u',j}, \top, [] \rangle$. The resulting hierarchy states then become $\langle M_r, u', \top, [] \rangle$ and $\langle \bar{M}_r, u', \top, [] \rangle$ respectively, which are clearly equivalent (the state is exactly the same and both come from equivalent hierarchy states).

We have shown both HRMs have equivalent traversals for any given trace, implying that both will accept, reject, or not accept nor reject a trace. Therefore, the HRMs are equivalent. $\qquad\square$

Figure 5a shows the result of applying the flattening algorithm on the BOOK HRM shown in Figure 1b. Note that the resulting HRM can be compressed: there are two states having an edge with the same label to a specific state. Indeed, the presented algorithm might not produce the smallest possible flat equivalent. Figure 5b shows the resulting compressed HRM, which is like Figure 1c but naming the states following the algorithm for clarity. Estimating how much a flat HRM (or any HRM) can be compressed and designing an algorithm to perform such compression are left as future work.

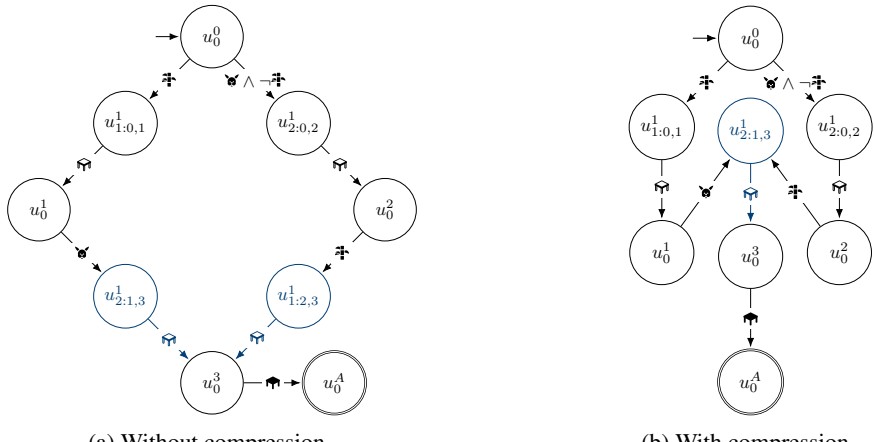

(a) Without compression.  (b) With compression.

Figure 5: Results of flattening the HRM in Figure 1b. The notation $u_{j:x,y}^i$ denotes the $i$-th state of RM $j$ in the call between states $x$ and $y$ in the parent RM. Note that $x$ and $y$ appear only if that state comes from a called RM. The blue states and edges in (a) can be compressed as shown in (b).

## B.2 PROOF OF THEOREM 2

We prove Theorem 2 by first characterizing an HRM $H$ using a set of abstract parameters. Then, we describe how the number of states and edges in an HRM and its corresponding flat equivalent are computed, and use these quantities to give an example for which the theorem holds. The parameters are the following:

- The height of the root $h_r$.
- The number of RMs with height $i$, $N^{(i)}$.
- The number of states in RMs with height $i$, $U^{(i)}$.
- The number of edges from each state in RMs with height $i$, $E^{(i)}$.

We assume that (i) RMs with height $i$ only call RMs with height $i - 1$; (ii) all RMs have a single accepting state and no rejecting states; (iii) all RMs except for the root are called; and (iv) the HRM is well-formed (i.e., it behaves deterministically and there are no cyclic dependencies). Note that $N^{(h_r)} = 1$ since there is a single root. Assumption (i) can be made since for the root to have height $h_r$ we need it to call at least one RM with height $h_r - 1$. Considering that all called RMs are have the same height simplifies the analysis since we can characterize the RMs at each height independently. Assumption (ii) is safe to make since a single accepting state is enough, and helps simplify the counting since only some RMs could have rejecting states. Assumption (iii) ensures that the flat HRM will comprise all RMs in the original HRM. This is also a fair assumption: if a given RM is not called by any RM in the hierarchy, we could remove it beforehand.

The *number of states $|H|$ in the HRM $H$* is obtained by summing the number of states of each RM:

$$|H| = \sum_{i=1}^{h_r} N^{(i)} U^{(i)}.$$

The *number of states $|\bar{H}|$ in the flat HRM $\bar{H}$* is given by the number of states in the flattened root RM

$$|\bar{H}| = \bar{U}^{(h_r)},$$

where $\bar{U}^{(i)}$ is the number of states in the flattened representation of an RM with height $i$, which is recursively defined as:

$$\bar{U}^{(i)} = \begin{cases} U^{(i)} & \text{if } i = 1, \\ U^{(i)} + \left(\bar{U}^{(i-1)} - 2\right)\left(U^{(i)} - 1\right) E^{(i)} & \text{if } i > 1. \end{cases}$$

That is, the number of states in a flattened RM with height $i$ has all states that the non-flat HRM had. In addition, for each of the $U^{(i)} - 1$ non-accepting states in the non-flat RM, there are $E^{(i)}$ edges, each of which calls an RM with height $i - 1$ whose number of states is $\bar{U}^{(i-1)}$. These edges are replaced by the called RM except for the initial and accepting states, whose role is now played by the states involved in the substituted edge (hence the $-2$). This construction process corresponds to the one used to prove Theorem 1.

The *total of number of edges* in an HRM is given by:

$$\sum_{i=1}^{h_r} N^{(i)} (U^{(i)} - 1) E^{(i)},$$

where $(U^{(i)} - 1) E^{(i)}$ is the total number of edges in an RM with height $i$ (the $-1$ is because the accepting state is discarded), so $N^{(i)} (U^{(i)} - 1) E^{(i)}$ determines how many edges there are across RMs with height $i$.

The *total number of edges in the flat HRM* is given by the total number of edges in the flattened root RM, $\bar{E}^{(h_r)}$, where $\bar{E}^{(i)}$ is the total number of edges in the flattened representation of an RM with height $i$, which is recursively defined as follows:

$$\bar{E}^{(i)} = \begin{cases} (U^{(i)} - 1) E^{(i)} & \text{if } i = 1, \\ (U^{(i)} - 1) E^{(i)} \bar{E}^{(i-1)} & \text{if } i > 1. \end{cases}$$

That is, each of the $(U^{(i)} - 1) E^{(i)}$ edges in an RM with height $i$ is replaced by $\bar{E}^{(i-1)}$ edges given by an RM with height $i - 1$ (if any).

The *key intuition* is that an HRM with root height $h_r > 1$ is beneficial representation-wise if the number of calls across RMs with height $i$ is higher than the number of RMs with height $i - 1$; in

other words, RMs with lower heights are being reused. Numerically, the total number of edges/calls in an RM with height $i$ is $(U^{(i)} - 1)E^{(i)}$ and, therefore, the total number of calls across RMs with height $i$ is $(U^{(i)} - 1)E^{(i)}N^{(i)}$. If this quantity is higher than $N^{(i-1)}$, then RMs with lower heights are reused, and therefore having RMs with different heights is beneficial.

**Theorem 2.** *Let $H = \langle \mathcal{M}, M_r, \mathcal{P} \rangle$ be an HRM and let $h_r$ be the height of its root $M_r$. The number of states and edges in an equivalent flat HRM $\bar{H}$ can be exponential in $h_r$.*

*Proof.* By example. Let $H = \langle \mathcal{M}, M_r, \mathcal{P} \rangle$ be an HRM whose root $M_r$ has height $h_r$ and is parameterized by $N^{(i)} = 1$, $U^{(i)} = 3$, $E^{(i)} = 1$ for $i = 1, \ldots, h_r$. Figure 6 shows an instance of this hierarchy. Let us write the number of states in the flat RMs of each level:

$$\bar{U}^{(1)} = U^{(1)} = 3,$$
$$\bar{U}^{(2)} = U^{(2)} + \left( \bar{U}^{(1)} - 2 \right) \left( U^{(2)} - 1 \right) E^{(2)} = 3 + (3 - 2)(3 - 1)1 = 5,$$
$$\bar{U}^{(3)} = U^{(3)} + \left( \bar{U}^{(2)} - 2 \right) \left( U^{(3)} - 1 \right) E^{(3)} = 3 + (5 - 2)(3 - 1)1 = 9,$$
$$\vdots$$
$$\bar{U}^{(i)} = 2\bar{U}^{(i-1)} - 1 = 2^i + 1.$$

Hence, the number of states in the flat HRM is $|\bar{H}| = \bar{U}^{(h_r)} = 2^{h_r} + 1$, showing that the number of states in the flat HRM grows exponentially with the height of the root. In contrast, the number of states in the HRM grows linearly with the height of the root, $|H| = \sum_{i=1}^{h_r} N^{(i)} U^{(i)} = \sum_{i=1}^{h_r} 1 \cdot 3 = 3h_r$.

Figure 6: Example of an HRM whose root has height $h_r$ used in the proof of Theorem 2.

In the case of the total number of edges, we again write some iterations to derive a general expression:

$$\bar{E}^{(1)} = (U^{(1)} - 1)E^{(1)} = (3 - 1)1 = 2,$$
$$\bar{E}^{(2)} = (U^{(2)} - 1)E^{(2)}\bar{E}^{(1)} = (3 - 1) \cdot 1 \cdot 2 = 4,$$
$$\bar{E}^{(3)} = (U^{(3)} - 1)E^{(3)}\bar{E}^{(2)} = (3 - 1) \cdot 1 \cdot 4 = 8,$$
$$\vdots$$
$$\bar{E}^{(i)} = 2\bar{E}^{(i-1)} = 2^i.$$

Therefore, the total number of edges in the flat HRM is $\bar{E}^{(h_r)} = 2^{h_r}$. In contrast, the total number of edges in the HRM grows linearly: $\sum_{i=1}^{h_r} N^{(i)}(U^{(i)} - 1)E^{(i)} = \sum_{i=1}^{h_r} 1(3 - 1)1 = 2h_r$.

Finally, we emphasize that the resulting flat HRM cannot be compressed, unlike the HRM in Figure 5: each state has at most one incoming edge, so there are not multiple paths that can be merged. We have thus shown that there are HRMs whose equivalent flat HRM has a number of states and edges that grows exponentially with the height of the root. □

Using the aforementioned intuition, we observe that the hierarchical structure is actually expected to be useful: the number of calls across RMs with height $i$ is $(U^{(i)} - 1)E^{(i)} = (3 - 1)1 = 2$, which is greater than the number of RMs with height $i - 1$ (only 1).

There are cases where having a multi-level hierarchy (i.e., with $h_r > 1$) is not beneficial. For instance, given an HRM whose root has height $h_r$ and parameterized by $N^{(i)} = 1$, $U^{(i)} = 2$ and $E^{(i)} = 1$, the number of states in the equivalent flat HRM is constant (2), whereas in the HRM itself it grows linearly with $h_r$. The same occurs with the number of edges. By checking the previously introduced intuition, we observe that $(U^{(i)} - 1)E^{(i)}N^{(i)} = (2 - 1) \cdot 1 \cdot 1 = 1 \not> N^{(i-1)} = 1$, which verifies that having non-reused RMs with multiple heights is not useful.

# C  POLICY LEARNING IMPLEMENTATION DETAILS

In this appendix, we describe some implementation details that were omitted in Section 4 for simplicity. First, we start by describing some methods used in policy learning. Second, we explain the option selection algorithm step-by-step and provide examples to ease its understanding.

## C.1  POLICIES

**Deep Q-networks (DQNs; Mnih et al., 2015).**  We use Double DQNs (van Hasselt et al., 2016) for both formula and call options. The DQNs associated with formula options simply take an MDP state and output a Q-value for each action. In contrast, the DQNs associated with call options also take an RM state and a context, which are encoded as follows:

- The RM state is encoded using a one-hot vector. The size of the vector is given by the number of states in the RM.
- The context, which is either $\top$ or a DNF formula with a single disjunct/conjunction, is encoded using a vector whose size is the number of propositions $|\mathcal{P}|$. Each vector position corresponds to a proposition $p \in \mathcal{P}$ whose value depends on how $p$ appears in the context: (i) +1 if $p$ appears positively, (ii) -1 if $p$ appears negatively, or (iii) 0 if $p$ does not appear. Note that if the context is $\top$, the vector solely consists of zeros.

These DQNs output a value for each possible call in the RM; however, some of these values must be masked if the corresponding calls are not available from the RM state-context used as input. For instance, the DQN for $M_0$ in Figure 1b outputs a value for $\langle M_1, \neg\text{⚫}\rangle$, $\langle M_2, \top\rangle$, $\langle M_1, \top\rangle$, and $\langle M_\top, \text{♠}\rangle$. If the RM state was $u_0^0$ and the context was $\top$, only the values for the first two calls are relevant. Just like unavailable calls, we also mask unsatisfiable calls (i.e., calls whose context cannot be satisfied in conjunction with the accumulated context used as input).

To speed up learning, a subset of the Q-functions associated with formula options is updated after each step. Updating all the Q-functions after each step is costly and we observed that similar performance could be obtained with this strategy. To determine the subset, we keep an update counter $c_\phi$ for each Q-function $q_\phi$, and a global counter $c$ (i.e., the total number of times Q-functions have been updated). The probability of updating $q_\phi$ is:

$$p_\phi = \frac{s_\phi}{\sum_{\phi'} s_{\phi'}}, \text{where } s_\phi = c - c_\phi - 1.$$

A subset of Q-functions is chosen using this probability distribution without replacement.

**Exploration.**  During training, the formula and call option policies are $\epsilon$-greedy. In the case of formula options, akin to Q-functions, each option $\omega_{i,u,\Phi}^{j,\phi}$ performs exploration with an exploration factor $\epsilon_{\phi\wedge\Phi}$, which linearly decreases with the number of steps performed using the policy induced by $q_{\phi\wedge\Phi}$. Likewise, Kulkarni et al. (2016) keep an exploration factor for each subgoal, but vary it depending on the option's success rather than on the number of performed steps. In the case of call options, each RM state-context pair is associated with its own exploration factor, which linearly decreases as options started from that pair terminate.

**The Formula Tree.**  As explained in Section 4, each formula option's policy is induced by a Q-function associated with a formula. In domains where certain proposition sets cannot occur, it is unnecessary to consider formulas that cover some of these sets. For instance, in a domain where two propositions $a$ and $b$ cannot be simultaneously observed (i.e., it is impossible to observe $\{a, b\}$), formulas such as $a \wedge \neg b$ or $b \wedge \neg a$ could instead be represented by the more abstract formulas $a$ or $b$; therefore, $a \wedge \neg b$ and $a$ could be both associated with a Q-function $q_a$, whereas $b \wedge \neg a$ and $b$ could be both associated with a Q-function $q_b$. By reducing the number of Q-functions, the learning naturally becomes more efficient.

We represent relationships between formulas using a *formula tree* which, as the name suggests, arranges a set of formulas in a tree structure. Formally, given a set of propositions $\mathcal{P}$, a formula tree is a tuple $\langle \mathcal{F}, F_r, \mathbb{L} \rangle$, where $\mathcal{F}$ is a set of nodes, each associated with a formula; $F_r \in \mathcal{F}$ is the root of the tree and it is associated with the formula $\top$; and $\mathbb{L} \subseteq (2^{\mathcal{P}})^*$ is a set of labels. All the nodes in the tree except for the root are associated with conjunctions. Let $\nu(X) \subseteq 2^{2^{\mathcal{P}}}$ denote the set of

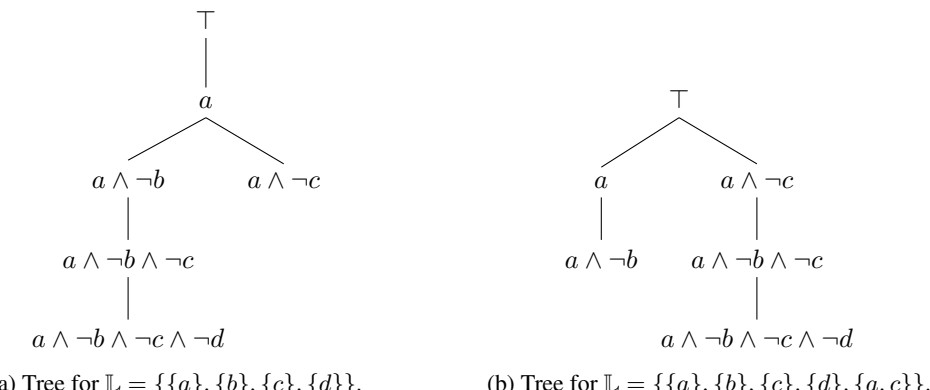

(a) Tree for $\mathbb{L} = \{\{a\}, \{b\}, \{c\}, \{d\}\}$.  (b) Tree for $\mathbb{L} = \{\{a\}, \{b\}, \{c\}, \{d\}, \{a, c\}\}$.

Figure 7: Examples of formula trees for different sets of literals. Note that the node $a \wedge \neg b \wedge \neg c$ in (a) could also be a child of $a \wedge \neg c$ (the parent depends on the insertion order).

literals of a formula $X$, e.g. if $X = a \wedge \neg b$, then $\nu(X) = \{a, \neg b\}$. A formula $X$ *subsumes* a formula $Y$ if (1) $X = \top$, or (2.i) $\nu(X) \subseteq \nu(Y)$ and (2.ii) for all labels $\mathcal{L} \in \mathbb{L}$, either $\mathcal{L} \models X$ and $\mathcal{L} \models Y$, or $\mathcal{L} \not\models X$ and $\mathcal{L} \not\models Y$. Case (2) indicates that $Y$ is a special case of $X$ (it adds literals but it is satisfied by exactly the same labels). The tree is organized such that the formula at a given node subsumes all its descendants. The set of Q-functions is determined by the children of the root.

During the agent-environment interaction, the formula tree is updated if (i) a new formula appears in the learned HRMs, or (ii) a new label is observed. Algorithm 1 contains the pseudo-code for updating the tree in these two cases. When a new formula is added (line 1), we create a node for the formula (line 2) and add it to the tree. The insertion place is determined by exploring the tree top-down from the root $F_r$ (lines 3-19). First, we check whether a child of the current node subsumes the new node (line 7). If such a node exists, then we go down this path (lines 8-9); otherwise, the new node is going to be a child of the current node (lines 16-17). In the latter case, in addition, all those children nodes from the current node that are subsumed by the new node need to become children of the new node (lines 11-15). The other core case in which the tree may need an update occurs when a new label is observed (lines 20-25) since we need to make sure that parenting relationships comply with the set of labels $\mathbb{L}$. First, we find nodes inconsistent with the new label: a parenting relationship is broken (line 39) when the formula of the parent non-root node is satisfied by the label but the formula of the child node is not (or vice versa). Once the inconsistent nodes are found, we remove their current parenting relationship (lines 45-46) and reinsert them in the tree (line 47). Figure 7 shows two simple examples of formula trees, where the Q-functions are $q_a$ in (a), and $q_a$ and $q_{a \wedge \neg c}$ in (b).

## C.2 Option Selection Algorithm

Algorithm 2 shows how options are selected, updated and interrupted during an episode. Lines 1-3 correspond to the algorithm's initialization. The initial state is that of the environment, while the initial hierarchy state is formed by the root RM $M_r$, its initial state $u_r^0$, an empty context (i.e., $\Phi = \top$), and an empty call stack. The option stack $\Omega_H$ contains the options we are currently running, where options at the front are the shallowest ones (e.g., the first option in the list is taken in the root RM). The steps taken during an episode are shown in lines 4-14, which are grouped as follows:

1. The agent fills the option stack $\Omega_H$ by selecting options in the HRM from the current hierarchy state until a formula option is chosen (lines 15-25). The context is propagated and augmented through the HRM (i.e., the context of the calls is conjuncted with the propagating context and converted into DNF form). Note that the context is initially $\top$ (true), and not that of the hierarchy state. It is possible that no new options are selected if the formula option chosen in a previous step has not terminated yet.

2. The agent chooses an action according to the last option in the option stack (line 6), which will always be a formula option whose policy maps states into actions. The action is ap-

---

**Algorithm 1** Formula tree operations

---

**Input:** A formula tree $\langle \mathcal{F}, F_r, \mathbb{L} \rangle$, where $\mathcal{F}$ is a set of nodes, $F_r \in \mathcal{F}$ is the root node (associated with the formula $\top$), and $\mathbb{L}$ is a set of labels.

 1: **procedure** ADDFORMULA($f$)
 2:     ADDNODE(CREATENODE($f$))
 3: **procedure** ADDNODE(new_node)
 4:     current_node $\leftarrow F_r$
 5:     added_node $\leftarrow \bot$
 6:     **while** added_node $= \bot$ **do**
 7:         child_node $\leftarrow$ FINDSUBSUMINGCHILD(current_node, new_node)
 8:         **if** child_node $\neq$ nil **then**                     ▷ Keep exploring down this path
 9:             current_node $\leftarrow$ child_node
10:         **else**                                              ▷ Insert the node
11:             subsumed_children $\leftarrow$ GETSUBSUMEDCHILDREN(current_node, new_node)
12:             new_node.children $\leftarrow$ new_node.children $\cup$ subsumed_children
13:             **for** child $\in$ subsumed_children **do**
14:                 current_node.children $\leftarrow$ current_node.children $\setminus \{$child$\}$
15:                 child.parent $\leftarrow$ new_node
16:             current_node.children $\leftarrow$ current_node.children $\cup \{$new_node$\}$
17:             new_node.parent $\leftarrow$ current_node
18:             added_node $\leftarrow \top$
19:     $\mathcal{F} \leftarrow \mathcal{F} \cup \{$new_node$\}$
20: **procedure** ONLABEL($\mathcal{L}$)
21:     $\mathbb{L} \leftarrow \mathbb{L} \cup \{\mathcal{L}\}$
22:     inconsistent_nodes $\leftarrow \{\}$
23:     **for** child $\in F_r$.children **do**
24:         FINDINCONSISTENTNODES(child, $\mathcal{L}$, inconsistent_nodes)
25:     REINSERTINCONSISTENTNODES(inconsistent_nodes)
26: **procedure** FINDSUBSUMINGCHILD(current_node, new_node)
27:     **for** child $\in$ current_node.children **do**
28:         **if** child.formula subsumes new_node.formula **then**
29:             **return** child
30:     **return** nil
31: **procedure** GETSUBSUMEDCHILDREN(current_node, new_node)
32:     subsumed_children $\leftarrow \{\}$
33:     **for** child $\in$ current_node.children **do**
34:         **if** new_node.formula subsumes child.formula **then**
35:             subsumed_children $\leftarrow$ subsumed_children $\cup \{$new_node$\}$
36:     **return** subsumed_children
37: **procedure** FINDINCONSISTENTNODES(current_node, $\mathcal{L}$, inconsistent_nodes)
38:     **for** child $\in$ current_node.children **do**
39:         **if** $\mathcal{L} \models$ current_node.formula $\oplus \mathcal{L} \models$ child.formula **then**
40:             inconsistent_nodes $\leftarrow$ inconsistent_nodes $\cup \{$child$\}$
41:         **else**
42:             FINDINCONSISTENTNODES(child, $\mathcal{L}$, inconsistent_nodes)
43: **procedure** REINSERTINCONSISTENTNODES(inconsistent_nodes)
44:     **for** node $\in$ inconsistent_nodes **do**
45:         node.parent.children $\leftarrow$ node.parent.children $\setminus \{$node$\}$
46:         node.parent $\leftarrow$ nil
47:         ADDNODE(node)

---

plied, and the agent gets the next state (line 7). The next hierarchy state is obtained by applying the hierarchical transition function $\delta_H$ using label $l(s_{t+1})$ (line 8). The Q-functions associated with formula options' policies are updated after this step (line 9).

3. The option stack $\Omega_H$ is updated by removing those options that have terminated (lines 10, 26-45). The terminated options are saved in a different list $\Omega_\beta$ to update the Q-functions of the RMs where they were initiated later on (line 11). The termination of the options is performed as described in Section 4. All options terminate if a terminal state is reached (lines 27-28). Otherwise, we check options in $\Omega_H$ from deeper to shallower levels. The first checked option is always a formula option, which terminates if the hierarchy state has changed (line 40). In contrast, a call option terminates if it does not appear in the stack (lines 33, 46-51).[4] When an option is found to terminate, it is added to $\Omega_\beta$ and removed from $\Omega_H$ (lines 35-36, 41-42). If a non-terminating option is found (lines 37, 43), we stop checking for termination (no higher level options can have terminated in this case).

4. If at least one option has terminated (line 12), the option stack is updated such that it contains all options appearing in the call stack (lines 13, 52-70). Options are derived for the full stack if $\Omega_H$ is empty (lines 53, 54), or for the part of the stack not appearing in $\Omega_H$ (lines 56-59). The new derived options (lines 61-70) from the call stack are assumed to start in the same state as the last terminated option (i.e., the shallowest terminated option, line 63) and to have been run for the same number of steps too. Crucially, the contexts should be propagated accordingly, starting from the context of the last terminated option (line 69).

As a result of the definition of the hierarchical transition function $\delta_H$, the contexts in the stack may be DNF formulas with more than one disjunct. In contrast, the contexts associated with options are either $\top$ or DNFs with a single disjunct (remember that an option is formed for each disjunct). For instance, this occurs if the context is $a \vee b$ and $\{a, b\}$ is observed: since both disjuncts are satisfied, the context shown in the call stack will be the full disjunction $a \vee b$. In the simplest case, the derived option (which as said before is associated with a DNF with a single disjunct or $\top$) can include one of these disjuncts chosen uniformly at random (line 67). Alternatively, we could memorize all the derived options and perform identical updates for both later on once terminated.

## C.3 Examples

We briefly describe some examples of how policy learning is performed in the HRM of Figure 1b. We first enumerate the options in the hierarchy. The formula options are $\omega_{1,0,\neg\blacktriangledown}^{\top,\maltese}$, $\omega_{2,0,\top}^{\top,\blacktriangledown}$, $\omega_{1,0,\top}^{\top,\maltese}$, $\omega_{1,1,\top}^{\top,\spadesuit}$, $\omega_{2,1,\top}^{\top,\spadesuit}$, and $\omega_{0,3,\top}^{\top,\clubsuit}$. The first option should lead the agent to observe the label $\{\maltese\}$ to satisfy $\maltese \wedge \neg\blacktriangledown$. The Q-functions associated with this set of options are $q_{\maltese \wedge \neg\blacktriangledown}$, $q_{\blacktriangledown}$, $q_{\maltese}$, $q_{\spadesuit}$ and $q_{\clubsuit}$. Note that $\omega_{1,1,\top}^{\top,\spadesuit}$ and $\omega_{2,1,\top}^{\top,\spadesuit}$ are both associated with $q_{\spadesuit}$. Conversely, the call options are $\omega_{0,0,\top}^{1,\neg\blacktriangledown}$, $\omega_{0,0,\top}^{2,\top}$, $\omega_{0,1,\top}^{2,\top}$, and $\omega_{0,2,\top}^{1,\top}$, where the first one achieves its local goal if formula options $\omega_{1,0,\neg\blacktriangledown}^{\top,\maltese}$ and $\omega_{1,1,\top}^{\top,\spadesuit}$ sequentially achieve theirs. The associated Q-functions are $q_0$, $q_1$ and $q_2$. Note that $\omega_{0,0,\top}^{2,\top}$ and $\omega_{0,1,\top}^{2,\top}$ are both associated with $q_2$.

We now describe a few steps of the aforementioned option selection algorithm in two scenarios. First, we consider the scenario where all chosen options are run to completion (i.e., until their local goals are achieved):

1. The initial hierarchy state is $\langle M_0, u_0^0, \top, [] \rangle$ and the option stack $\Omega_H$ is empty. We select options to fill $\Omega_H$. The first option is chosen from $u_0^0$ in $M_0$ using a policy induced by $q_0$. At this state, the available options are $\omega_{0,0,\top}^{1,\neg\blacktriangledown}$ and $\omega_{0,0,\top}^{2,\top}$. Let us assume that the former is chosen. Then an option from the initial state of $M_1$ under context $\neg\blacktriangledown$ is chosen, which can only be $\omega_{1,0,\neg\blacktriangledown}^{\top,\maltese}$. Since this option is a formula option (the call is made to $M_\top$), we do not select any more options and the option stack is $\Omega_H = \langle \omega_{0,0,\top}^{1,\neg\blacktriangledown}, \omega_{1,0,\neg\blacktriangledown}^{\top,\maltese} \rangle$.

---

[4]We denote by $\phi_1 \subseteq \phi_2$, where $\phi_1, \phi_2 \in \mathrm{DNF}_\mathcal{P}$, the fact that all the disjuncts of $\phi_1$ appear in $\phi_2$. This containment relationship also holds if both formulas are $\top$. For instance, $(a \wedge \neg c) \subseteq (a \wedge \neg c) \vee d$.

---

**Algorithm 2** Episode execution using an HRM (continues in p. 27)

---

**Input:** An HRM $H = \langle \mathcal{M}, M_r, \mathcal{P} \rangle$ and an environment $\text{ENV} = \langle \mathcal{S}, \mathcal{A}, p, r, \gamma, \mathcal{P}, l, \tau \rangle$.

1:  $s_0 \leftarrow \text{ENV.INIT}()$          ▷ Initial MDP tuple
2:  $\langle M_i, u, \Phi, \Gamma \rangle \leftarrow \langle M_r, u_r^0, \top, [] \rangle$          ▷ Initial hierarchy state
3:  $\Omega_H \leftarrow []$          ▷ Initial option stack
4:  **for** each step $t = 0, \ldots,$ **do**
5:      $\Omega_H \leftarrow \text{FILLOPTIONSTACK}(s_t, \langle M_i, u, \Phi, \Gamma \rangle, \Omega_H)$          ▷ Expand the option stack
6:      $a \leftarrow \text{SELECTACTION}(s_t, \Omega_H)$          ▷ Choose $a$ according to the last option in $\Omega_H$
7:      $s_{t+1} \leftarrow \text{ENV.APPLYACTION}(a)$
8:      $\langle M_j, u', \Phi', \Gamma' \rangle \leftarrow \delta_H(\langle M_i, u, \Phi, \Gamma \rangle, l(s_{t+1}))$          ▷ Apply transition function
9:      $\text{UPDATEFORMULAQFUNCTIONS}(s_t, a, s_{t+1})$
10:     $\Omega_\beta, \Omega_H \leftarrow \text{TERMINATEOPTIONS}(\Omega_H, s, \langle M_i, u, \Phi, \Gamma \rangle, \langle M_j, u', \Phi', \Gamma' \rangle)$
11:     $\text{UPDATECALLQFUNCTIONS}(\Omega_\beta, s_{t+1}, l(s_{t+1}))$
12:     **if** $|\Omega_\beta| > 0$ **then**
13:        $\Omega_H \leftarrow \text{ALIGNOPTIONSTACK}(\Omega_H, \Gamma', \Omega_\beta)$
14:     $\langle M_i, u, \Phi, \Gamma \rangle \leftarrow \langle M_j, u', \Phi', \Gamma' \rangle$
15: **procedure** $\text{FILLOPTIONSTACK}(s, \langle M_i, u, \cdot, \Gamma \rangle, \Omega_H)$
16:     $\Omega'_H \leftarrow \Omega_H$
17:     $\Phi \leftarrow \top$          ▷ The context is initially true
18:     $M_j \leftarrow M_i; v \leftarrow u$          ▷ The state-automaton pair in which an option is selected
19:     **while** the last option in $\Omega'_H$ is not a formula option **do**
20:        $\omega_{j,v,\Phi}^{x,\phi} \leftarrow \text{SELECTOPTION}(s, M_j, v, \Phi)$          ▷ Select an option (e.g., with $\epsilon$-greedy)
21:        **if** $x \neq \top$ **then**          ▷ If the option is a call option
22:           $M_j \leftarrow M_x; v \leftarrow u_x^0$          ▷ Next option is chosen on the called RM's initial state
23:           $\Phi \leftarrow \text{DNF}(\Phi \wedge \phi)$          ▷ Update the context
24:        $\Omega'_H \leftarrow \Omega_H \oplus \omega_{j,v,\Phi}^{x,\phi}$          ▷ Update the option stack (concatenate new option)
25:     **return** $\Omega'_H$
26: **procedure** $\text{TERMINATEOPTIONS}(\Omega_H, s, \langle M_i, u, \Phi, \Gamma \rangle, \langle M_j, u', \Phi', \Gamma' \rangle)$
27:     **if** $s^T = \top$ **then**
28:        **return** $\Omega_H, []$          ▷ All options terminate
29:     $\Omega_\beta \leftarrow []; \Omega'_H \leftarrow \Omega_H$          ▷ Initialize structures
30:     **while** $|\Omega'_H| > 0$ **do**          ▷ While the option stack is not empty
31:        $\omega_{k,v,\Psi}^{x,\phi} \leftarrow$ last option in $\Omega'_H$
32:        **if** $x \neq \top$ **then**          ▷ If the option is a call option
33:           in_stack, _ $\leftarrow \text{OPTIONINSTACK}(\omega_{k,v,\Psi}^{x,\phi}, \Gamma')$
34:           **if** $\neg$in_stack **then**
35:              $\Omega_\beta \leftarrow \Omega_\beta \oplus \omega_{k,v,\Psi}^{x,\phi}$          ▷ Update the list of terminated options
36:              $\Omega'_H \leftarrow \Omega'_H \ominus \omega_{k,v,\Psi}^{x,\phi}$          ▷ Remove the last option from the option stack
37:           **else**
38:              **break**          ▷ Stop unrolling
39:        **else**
40:           **if** $\langle M_i, u, \Phi, \Gamma \rangle \neq \langle M_j, u', \Phi', \Gamma' \rangle$ **then**          ▷ If the hierarchy state has changed...
41:              $\Omega_\beta \leftarrow \Omega_\beta \oplus \omega_{k,v,\Psi}^{x,\phi}$          ▷ Update the list of terminated options
42:              $\Omega'_H \leftarrow \Omega'_H \ominus \omega_{k,v,\Psi}^{x,\phi}$          ▷ Remove the last option from the option stack
43:           **else**
44:              **break**          ▷ Stop unrolling
45:     **return** $\Omega_\beta, \Omega'_H$
46: **procedure** $\text{OPTIONINSTACK}(\omega_{k,v,\Phi}^{x,\phi}, \Gamma)$
47:     **for** $l = 0 \ldots |\Gamma| - 1$ **do**
48:        $\langle u_f, \cdot, M_i, M_j, \phi', \Phi' \rangle \leftarrow \Gamma_l$
49:        **if** $u_f = v \wedge i = k \wedge j = x \wedge \phi \subseteq \phi' \wedge \Phi \subseteq \Phi'$ **then** ▷ The call option is in the call stack
50:           **return** $\top, l$          ▷ Return whether it appears in the stack and the index
51:     **return** $\bot, -1$

---

---

52: **procedure** ALIGNOPTIONSTACK($\Omega_H, \Gamma, \Omega_\beta$)
53:     **if** $|\Omega_H| = 0$ **then**
54:         **return** ALIGNOPTIONSTACKHELPER($\Omega_H, \Gamma, \Omega_\beta, 0$)
55:     **else**
56:         $\omega_{k,v,\Phi}^{x,\phi} \leftarrow$ last option in $\Omega_H$
57:         in_stack, stack_index $\leftarrow$ OPTIONINSTACK($\omega_{k,v,\Phi}^{x,\phi}, \Gamma$)
58:         **if** in_stack **then**
59:             **return** ALIGNOPTIONSTACKHELPER($\Omega_H, \Gamma, \Omega_\beta$, stack_index)
60:     **return** $\Omega_H$
61: **procedure** ALIGNOPTIONSTACKHELPER($\Omega_H, \Gamma, \Omega_\beta$, stack_index)
62:     $\Omega'_H \leftarrow \Omega_H$
63:     $\omega_{\cdot,\cdot,\Phi}^{\cdot,\cdot} \leftarrow$ last option in $\Omega_\beta$            ▷ Shallowest terminated option
64:     $\Phi' \leftarrow \Phi$            ▷ Context initialized from last option
65:     **for** $l = $ stack_index $\dots |\Gamma| - 1$ **do**
66:         $\langle u_f, \cdot, M_i, M_j, \phi, \cdot \rangle \leftarrow \Gamma_l$
67:         $\phi_{sel} \leftarrow$ Select disjunct from $\phi$ (e.g., randomly)
68:         $\Omega'_H \leftarrow \Omega'_H \oplus \omega_{i,u_f,\Phi'}^{j,\phi_{sel}}$         ▷ Append new option to the option stack
69:         $\Phi' \leftarrow$ DNF($\Phi' \wedge \phi$)
70:     **return** $\Omega'_H$

---

2. The agent selects options according to the formula option in $\Omega_H$, $\omega_{1,0,\neg\mathlargor}^{\top,\maltese}$, whose policy is induced by $q_{\maltese \wedge \neg\mathlarger}$. Let us assume that the policy tells the agent to turn right. Since the label at this location is empty, the hierarchy state remains the same; therefore, no options terminate, and the option stack does not change.

3. Let us assume that the agent moves forward twice, thus observing $\{\maltese\}$. The hierarchy state then becomes $\langle M_1, u_1^1, \top, [\langle u_0^0, u_0^1, M_0, M_1, \neg\mathlarger, \top \rangle] \rangle$ (see Appendix A for a step-by-step application of the hierarchical transition function). We check which options in $\Omega_H$ have terminated starting from the last chosen one. The formula option $\omega_{1,0,\neg\mathlarger}^{\top,\maltese}$ terminates because the hierarchy state has changed. In contrast, the call option $\omega_{0,0,\top}^{1,\neg\mathlarger}$ does not terminate since there is an item in the call stack, $\langle u_0^0, u_0^1, M_0, M_1, \neg\mathlarger, \top \rangle$ that can be mapped into it (meaning that the option is running).

4. An experience $(\boldsymbol{s}, \omega_{1,0,\neg\mathlarger}^{\top,\maltese}, \boldsymbol{s}')$ is formed for the terminated option, where $\boldsymbol{s}$ and $\boldsymbol{s}'$ are the observed tuples on initiation and termination respectively. This tuple is added to the replay buffer associated with the RM where the option appears, $\mathcal{D}_1$, since it achieved its goal (i.e., a label that satisfied $\maltese \wedge \neg\mathlarger$ was observed).

5. We align $\Omega_H$ with the new stack. In this case, $\Omega_H$ remains unchanged since its only option can be mapped into an item of the new stack.

6. We start a new step. Since the option stack does not contain a formula option, we select new options from the current hierarchy state according to a policy induced by $q_1$. In this case, there is a single eligible option: $\omega_{1,1,\top}^{\top,\mathlarger}$.

In the second scenario, we observe what occurs when the HRM traversal differs from the options chosen by the agent:

1. The initial step is like the one in the previous scenario, but we assume $\omega_{0,0,\top}^{2,\top}$ is selected instead. Then, since this is a call option, an option from the initial state of $M_2$ under context $\top$ is chosen, which can only be $\omega_{2,0,\top}^{\top,\mathlarger}$. The option stack thus becomes $\Omega_H = \langle \omega_{0,0,\top}^{2,\top}, \omega_{2,0,\top}^{\top,\mathlarger} \rangle$.

2. Let us assume that by taking actions according to $\omega_{2,0,\top}^{\top,\mathlarger}$ we end up observing $\{\maltese\}$. Like in the previous scenario, the hierarchy state becomes $\langle M_1, u_1^1, \top, [\langle u_0^0, u_0^1, M_0, M_1, \neg\mathlarger, \top \rangle] \rangle$. We check which options in $\Omega_H$ have terminated. The formula option $\omega_{2,0,\top}^{\top,\mathlarger}$ terminates since the hierarchy state has changed, and the call option $\omega_{0,0,\top}^{2,\top}$ also terminates since it

cannot be mapped into an item of the call stack. Note that these options should intuitively finish since the HRM is being traversed through a path different from that chosen by the agent.

3. The replay buffers are not updated for these options since they have not achieved their local goals.

4. We align $\Omega_H$ with the new stack. The only item of the stack $\langle u_0^0, u_0^1, M_0, M_1, \neg \mathbf{\text{✶}}, \top \rangle$ can be mapped into option $\omega_{0,0,\top}^{1,\neg \mathbf{\text{✶}}}$. We assume that this option starts on the same tuple $s$ and that it has run for the same number of steps as the last terminated option $\omega_{0,0,\top}^{2,\top}$.

## D  CURRICULUM LEARNING IMPLEMENTATION

We here describe the details of the curriculum learning method described in Section 5. When an episode is completed for $\mathbb{M}_{ij}$, $R_{ij}$ is updated using the episode's undiscounted return $r$ as $R_{ij} \leftarrow \beta R_{ij} + (1 - \beta)r$, where $\beta \in [0, 1]$ is a hyperparameter. A score $c_{ij} = 1 - R_{ij}$ is computed from the return and used to determine the probability of selecting tasks and instances. Note that this scoring function, also used in the curriculum method by Andreas et al. (2017), assumes that the undiscounted return ranges between 0 and 1 (see Section 2). The probability of choosing task $i$ is $\max_j c_{ij} / \sum_k \max_l c_{kl}$; that is, the task for which an instance is performing very poorly has a higher probability. Having selected task $i$, the probability of choosing instance $j$ is $c_{ij} / \sum_k c_{ik}$, i.e. instances where performance is worse have a higher probability of being chosen.

## E  LEARNING AN HRM FROM TRACES WITH ILASP

We formalize the task of learning an HRM using ILASP (Law et al., 2015), an inductive logic programming system that learns answer set programs (ASP) from examples. We address the reader to Gelfond & Kahl (2014) for an introduction to ASP, and to Law (2018) for ILASP. Our formalization is close to that by Furelos-Blanco et al. (2021) for flat finite-state machines. Without loss of generality, as stated in Section 5, we assume that each RM has exactly one accepting and one rejecting state. We first describe how HRMs are represented in ASP, and then explain the encoding of the HRM learning task in ILASP.

### E.1  REPRESENTATION OF AN HRM IN ANSWER SET PROGRAMMING

In this section, we explain how HRMs are represented using Answer Set Programming (ASP). First, we describe how traces are represented. Then, we present how HRMs themselves are represented and also introduce the general rules that describe the behavior of these hierarchies. Finally, we prove the correctness of the representation. We use $\mathbb{A}(X)$ to denote the ASP representation of $X$ (e.g., a trace).

**Definition 9** (ASP representation of a label trace). Given a label trace $\lambda = \langle \mathcal{L}_0, \ldots, \mathcal{L}_n \rangle$, $M(\lambda)$ denotes the set of ASP facts that describe it:

$$\mathbb{A}(\lambda) = \begin{cases} \{\texttt{label}(p, t). \mid 0 \leq t \leq n, p \in \mathcal{L}_t\} \cup \\ \{\texttt{step}(t). \mid 0 \leq t \leq n\} \cup \\ \{\texttt{last}(n).\}. \end{cases}$$

The $\texttt{label}(p, t)$ fact indicates that proposition $p \in \mathcal{P}$ is observed in step $t$, $\texttt{step}(t)$ states that $t$ is a step of the trace, and $\texttt{last}(n)$ indicates that the trace ends in step $n$.

**Example 2.** The set of ASP facts for the label trace $\lambda = \langle \{\mathbf{\text{▤}}\}, \{\}, \{\mathbf{\text{♠}}\} \rangle$ is $\mathbb{A}(\lambda) = \{\texttt{label}(\mathbf{\text{▤}}, 0).$, $\texttt{label}(\mathbf{\text{♠}}, 2).$, $\texttt{step}(0), \texttt{step}(1)., \texttt{step}(2)., \texttt{last}(2).\}$.

**Definition 10** (ASP representation of an HRM). Given an HRM $H = \langle \mathcal{M}, M_r, \mathcal{P} \rangle$, $\mathbb{A}(H) = \bigcup_{M_i \in \mathcal{M} \setminus \{M_\top\}} \mathbb{A}(M_i)$, where:

$$\mathbb{A}(M_i) = \mathbb{A}_{\mathcal{U}}(M_i) \cup \mathbb{A}_{\varphi}(M_i),$$
$$\mathbb{A}_{\mathcal{U}}(M_i) = \{\texttt{state}(u, M_i). \mid u \in \mathcal{U}_i\},$$

$$\mathbb{A}_\varphi(M_i) = \begin{cases} \texttt{call}(u, u', x+e, M_i, M_j). \\ \bar\varphi(u, u', x+e, M_i, \texttt{T})\!:\!\texttt{-not label}(p_1, \texttt{T}), \texttt{step}(\texttt{T}). \\ \quad\quad\quad\quad\quad\quad\quad\vdots \\ \bar\varphi(u, u', x+e, M_i, \texttt{T})\!:\!\texttt{-not label}(p_n, \texttt{T}), \texttt{step}(\texttt{T}). \\ \bar\varphi(u, u', x+e, M_i, \texttt{T})\!:\!\texttt{-label}(p_{n+1}, \texttt{T}), \texttt{step}(\texttt{T}). \\ \quad\quad\quad\quad\quad\quad\quad\vdots \\ \bar\varphi(u, u', x+e, M_i, \texttt{T})\!:\!\texttt{-label}(p_m, \texttt{T}), \texttt{step}(\texttt{T}). \end{cases} \left| \begin{array}{c} M_j \in \mathcal{M}, u, u' \in \mathcal{U}_i, \\[4pt] \varphi_i(u, u', M_j) \neq \bot, \\ x = \sum_{k=0}^{j-1} |\varphi_i(u, u', M_k)|, \\ e \in [1, |\varphi_i(u, u', M_j)|], \\[4pt] \phi_e \in \varphi_i(u, u', M_j), \\ \phi_e = p_1 \wedge \cdots \wedge p_n \\ \wedge \neg p_{n+1} \wedge \cdots \wedge \neg p_m \end{array} \right\}.$$

Note that each non-leaf RM $M_i$ in the hierarchy is associated with its own set of rules $\mathbb{A}(M_i)$, which are described as follows:

- Facts $\texttt{state}(u, M_i)$ indicate that $u$ is a state of RM $M_i$.
- Facts $\texttt{call}(u, u', e, M_i, M_j)$ indicate that edge $e$ between states $u$ and $u'$ in RM $M_i$ is labeled with a call to RM $M_j$.
- Normal rules whose *head* is of the form $\bar\varphi(u, u', e, M_i, \texttt{T})$ indicate that the transition from state $u$ to $u'$ with edge $e$ in RM $M_i$ does not hold at step $\texttt{T}$. The *body* of these rules consists of a single $\texttt{label}(p, \texttt{T})$ literal and a $\texttt{step}(\texttt{T})$ atom indicating that $\texttt{T}$ is a step. Commonly, variables are represented using upper case letters in ASP, which is the case of steps $\texttt{T}$ here.

There are some important things to take into account regarding the encoding:

- There is no leaf RM $M_\top$. We later introduce the ASP rules to emulate it.
- The edge identifiers $e$ between a given pair of states $(u, u')$ range from 1 to the total number of conjunctions/disjuncts between them. Note that in $\mathbb{A}_\varphi$ we assume that the leaf RM has an index, just like the other RMs in the HRM. The index could be $n$ since the rest are numbered from 0 to $n-1$.

**Example 3.** The following rules represent the HRM in Figure 1b:

$$\begin{cases} \texttt{state}(u_0^0, M_0). \texttt{state}(u_0^1, M_0). \texttt{state}(u_0^2, M_0). & \texttt{state}(u_0^3, M_0). \texttt{state}(u_0^A, M_0). \\ \texttt{call}(u_0^0, u_0^1, 1, M_0, M_1). \texttt{call}(u_0^0, u_0^2, 1, M_0, M_2). & \texttt{call}(u_0^1, u_0^3, 1, M_0, M_2). \texttt{call}(u_0^2, u_0^3, 1, M_0, M_1). \\ \texttt{call}(u_0^3, u_0^A, 1, M_0, M_\top). & \bar\varphi(u_0^0, u_0^1, 1, M_0, \texttt{T})\!:\!\texttt{-label}(\text{✇}, \texttt{T}), \texttt{step}(\texttt{T}). \\ \bar\varphi(u_0^3, u_0^A, 1, M_0, \texttt{T})\!:\!\texttt{-not label}(\text{♠}, \texttt{T}), \texttt{step}(\texttt{T}). \end{cases} \cup$$
$$\begin{cases} \texttt{state}(u_1^0, M_1). \texttt{state}(u_1^1, M_1). \texttt{state}(u_1^A, M_1). & \texttt{call}(u_1^0, u_1^1, 1, M_1, M_\top). \texttt{call}(u_1^1, u_1^A, 1, M_1, M_\top). \\ \bar\varphi(u_1^0, u_1^1, 1, M_1, \texttt{T})\!:\!\texttt{-not label}(\text{♣}, \texttt{T}), \texttt{step}(\texttt{T}). & \bar\varphi(u_1^1, u_1^A, 1, M_1, \texttt{T})\!:\!\texttt{-not label}(\text{♞}, \texttt{T}), \texttt{step}(\texttt{T}). \end{cases} \cup$$
$$\begin{cases} \texttt{state}(u_2^0, M_2). \texttt{state}(u_2^1, M_2). \texttt{state}(u_2^A, M_2). & \texttt{call}(u_2^0, u_2^1, 1, M_2, M_\top). \texttt{call}(u_2^1, u_2^A, 1, M_2, M_\top). \\ \bar\varphi(u_2^0, u_2^1, 1, M_2, \texttt{T})\!:\!\texttt{-not label}(\text{✇}, \texttt{T}), \texttt{step}(\texttt{T}). & \bar\varphi(u_2^1, u_2^A, 1, M_2, \texttt{T})\!:\!\texttt{-not label}(\text{♞}, \texttt{T}), \texttt{step}(\texttt{T}). \end{cases}.$$

**General Rules.** The following sets of rules, whose union is denoted by $\mathcal{R} = \cup_{i=0}^{5} \mathcal{R}_i$, represent how an HRM functions (e.g., how transitions are taken or the acceptance/rejection criteria). For simplicity, all initial, accepting and rejecting states are denoted by $u^0$, $u^A$ and $u^R$ respectively.

The rule below is the inversion of the negation of the state transition function $\bar\varphi$. Note that the predicate for $\varphi$ includes the called RM M2 as an argument.

$$\mathcal{R}_0 = \{\varphi(\texttt{X}, \texttt{Y}, \texttt{E}, \texttt{M}, \texttt{M2}, \texttt{T})\!:\!\texttt{-not } \bar\varphi(\texttt{X}, \texttt{Y}, \texttt{E}, \texttt{M}, \texttt{T}), \texttt{call}(\texttt{X}, \texttt{Y}, \texttt{E}, \texttt{M}, \texttt{M2}), \texttt{step}(\texttt{T}).\}.$$

The rule set $\mathcal{R}_1$ introduces the $\texttt{pre\_sat}(\texttt{X}, \texttt{M}, \texttt{T})$ predicate, which encodes the exit condition presented in Section 3 and indicates whether a call from state $\texttt{X}$ of RM $\texttt{M}$ can be started at time $\texttt{T}$. The first rule corresponds to the base case and indicates that if the leaf $M_\top$ is called then the condition is satisfied if the associated formula is satisfied. The second rule applies to calls to non-leaf RMs, where we need to satisfy the context of the call (like in the base case), and also check whether a call from the initial state of the potentially called RM can be started.

$$\mathcal{R}_1 = \begin{cases} \texttt{pre\_sat}(\texttt{X}, \texttt{M}, \texttt{T})\!:\!\texttt{-}\varphi(\texttt{X}, \_, \_, \texttt{M}, M_\top, \texttt{T}). \\ \texttt{pre\_sat}(\texttt{X}, \texttt{M}, \texttt{T})\!:\!\texttt{-}\varphi(\texttt{X}, \_, \_, \texttt{M}, \texttt{M2}, \texttt{T}), \texttt{pre\_sat}(u^0, \texttt{M2}, \texttt{T}), \texttt{M2}\!\texttt{!=}\!M_\top. \end{cases}.$$

The rule set $\mathcal{R}_2$ introduces the $\texttt{reachable}(\texttt{X}, \texttt{M}, \texttt{T0}, \texttt{T2})$ predicate, which indicates that state $\texttt{X}$ of RM $\texttt{M}$ is reached between steps $\texttt{T0}$ and $\texttt{T2}$. The latter step can also be seen as the step we are

currently at. The first fact indicates that the initial state of the root RM is reached from step 0 to step 0. The second rule indicates that the initial state of a non-root RM is reached from step T to step T (i.e., it is reached anytime). The third rule represents the loop transition in the initial state of the root $M_r$: we stay there if no call can be started at T (i.e., we are not moving in the HRM). The fourth rule is analogous to the third but for the accepting state of the root instead of the initial state. Remember this is the only accepting state in the HRM that does not return control to the calling RM. The fifth rule is also similar to the previous ones: it applies to states reached after T0 that are non-accepting, which excludes looping in initial states of non-root RMs at the time of starting them (i.e., loops are permitted in the initial state of a non-root RM if we can reach it afterwards by going back to it). The last rule indicates that Y is reached at step T2 in RM M started at T0 if there is an outgoing transition from the current state X to Y at time T that holds between T and T2, and state X has been reached between T0 and T. We will later see how $\delta$ is defined.

$$\mathcal{R}_2 = \begin{cases} \texttt{reachable}(u^0, M_r, 0, 0). \\ \texttt{reachable}(u^0, \texttt{M}, \texttt{T}, \texttt{T}) \texttt{:-} \texttt{state}(u^0, \texttt{M}), \texttt{M!=}M_r, \texttt{step}(\texttt{T}). \\ \texttt{reachable}(\texttt{X}, \texttt{M}, \texttt{T0}, \texttt{T+1}) \texttt{:-} \texttt{reachable}(\texttt{X}, \texttt{M}, \texttt{T0}, \texttt{T}), \texttt{not pre\_sat}(\texttt{X}, \texttt{M}, \texttt{T}), \\ \qquad\qquad\qquad \texttt{step}(\texttt{T}), \texttt{X=}u^0, \texttt{M=}M_r. \\ \texttt{reachable}(\texttt{X}, \texttt{M}, \texttt{T0}, \texttt{T+1}) \texttt{:-} \texttt{reachable}(\texttt{X}, \texttt{M}, \texttt{T0}, \texttt{T}), \texttt{not pre\_sat}(\texttt{X}, \texttt{M}, \texttt{T}), \\ \qquad\qquad\qquad \texttt{step}(\texttt{T}), \texttt{X=}u^A, \texttt{M=}M_r. \\ \texttt{reachable}(\texttt{X}, \texttt{M}, \texttt{T0}, \texttt{T+1}) \texttt{:-} \texttt{reachable}(\texttt{X}, \texttt{M}, \texttt{T0}, \texttt{T}), \texttt{not pre\_sat}(\texttt{X}, \texttt{M}, \texttt{T}), \\ \qquad\qquad\qquad \texttt{step}(\texttt{T}), \texttt{T0<T}, \texttt{X!=}u^A. \\ \texttt{reachable}(\texttt{Y}, \texttt{M}, \texttt{T0}, \texttt{T2}) \texttt{:-} \texttt{reachable}(\texttt{X}, \texttt{M}, \texttt{T0}, \texttt{T}), \delta(\texttt{X}, \texttt{Y}, \texttt{M}, \texttt{T}, \texttt{T2}). \end{cases}.$$

The rule set $\mathcal{R}_3$ introduces two predicates. The predicate $\texttt{satisfied}(\texttt{M}, \texttt{T0}, \texttt{TE})$ indicates that RM M is satisfied if its accepting state $u^A$ is reached between steps T0 and TE. Likewise, the predicate $\texttt{failed}(\texttt{M}, \texttt{T0}, \texttt{TE})$ indicates that RM M fails if its rejecting state $u^R$ is reached between steps T0 and TE. These two descriptions correspond to the first and third rules. The second rule applies to the leaf RM $M_\top$, which always returns control immediately; thus, it is always satisfied between any two consecutive steps.

$$\mathcal{R}_3 = \begin{cases} \texttt{satisfied}(\texttt{M}, \texttt{T0}, \texttt{TE}) \texttt{:-} \texttt{reachable}(u^A, \texttt{M}, \texttt{T0}, \texttt{TE}). \\ \texttt{satisfied}(M_\top, \texttt{T}, \texttt{T+1}) \texttt{:-} \texttt{step}(\texttt{T}). \\ \texttt{failed}(\texttt{M}, \texttt{T0}, \texttt{TE}) \texttt{:-} \texttt{reachable}(u^R, \texttt{M}, \texttt{T0}, \texttt{TE}). \end{cases}$$

The following set, $\mathcal{R}_4$, encodes multi-step transitions within an RM. The predicate $\delta(\texttt{X}, \texttt{Y}, \texttt{M}, \texttt{T}, \texttt{T2})$ expresses that the transition from state X to state Y in RM M is satisfied between steps T and T2. The first rule indicates that this occurs if the context labeling a call to an RM M2 is satisfied and that RM is also satisfied (i.e., its accepting state is reached) between these two steps. In contrast, the second rule is used for the case in which the rejecting state of the called RM is reached between those steps. In the latter case, we transition to the local rejecting state $u^R$ of M (i.e., the state we would have transitioned to does not matter). This follows from the assumption that rejecting states are global rejectors (see Section 3). The idea of this rule is that rejection is propagated bottom-up in the HRM.

$$\mathcal{R}_4 = \begin{cases} \delta(\texttt{X}, \texttt{Y}, \texttt{M}, \texttt{T}, \texttt{T2}) \texttt{:-} \varphi(\texttt{X}, \texttt{Y}, \_, \texttt{M}, \texttt{M2}, \texttt{T}), \texttt{satisfied}(\texttt{M2}, \texttt{T}, \texttt{T2}). \\ \delta(\texttt{X}, u^R, \texttt{M}, \texttt{T}, \texttt{T2}) \texttt{:-} \varphi(\texttt{X}, \_, \_, \texttt{M}, \texttt{M2}, \texttt{T}), \texttt{failed}(\texttt{M2}, \texttt{T}, \texttt{T2}). \end{cases}.$$

The last set, $\mathcal{R}_5$, encodes the accepting/rejecting criteria. Remember that the $\texttt{last}(\texttt{T})$ predicate indicates that T is the last step of a trace. Therefore, the trace is accepted if the root RM is satisfied from the initial step 0 to step $T + 1$ (the step after the last step of the trace, once the final label has been processed). In contrast, the trace is rejected if a rejecting state in the hierarchy is reached between these two same steps.

$$\mathcal{R}_5 = \begin{cases} \texttt{accept} \texttt{:-} \texttt{last}(\texttt{T}), \texttt{satisfied}(M_r, 0, \texttt{T+1}). \\ \texttt{reject} \texttt{:-} \texttt{last}(\texttt{T}), \texttt{failed}(M_r, 0, \texttt{T+1}). \end{cases}$$

Unlike the formalism introduced in Section 3, this encoding does not use stacks, which would be costly to do. Here we know the trace to be processed and, therefore, the RMs can be evaluated bottom-up; that is, we start evaluating the lowest level RMs first on different subtraces, and the result of this evaluation is used in higher level RMs.

We now prove the correctness of the ASP encoding. To do so, we first introduce what means for an HRM to be valid with respect to a trace, as well as a definition and a theorem due to Gelfond & Lifschitz (1988) that will help us derive the proof.

**Definition 11.** Given a label trace $\lambda^*$, where $* \in \{G, D, I\}$, an HRM $H$ is valid with respect to $\lambda^*$ if $H$ accepts $\lambda^*$ and $* = G$ (i.e., $\lambda^*$ is a goal trace), or $H$ rejects $\lambda^*$ and $* = D$ (i.e., $\lambda^*$ is a dead-end trace), or $H$ does not accept nor reject $\lambda^*$ and $* = I$ (i.e., $\lambda^*$ is an incomplete trace).

**Definition 12.** An ASP program $P$ is stratified when there is a partition

$$P = P_0 \cup P_1 \cup \cdots \cup P_n \qquad\qquad (P_i \text{ and } P_j \text{ disjoint for all } i \neq j)$$

such that, (1) for every predicate $p$, the definition of $p$ (all clauses with $p$ in the head) is contained in one of the partitions $P_i$ and, (2) for each $1 \leq i \leq n$, if a predicate occurs positively in a clause of $P_i$ then its definition is contained within $\bigcup_{j \leq i} P_j$, and if a predicate occurs negatively in a clause of $P_i$ then its definition is contained within $\bigcup_{j < i} P_j$.

**Theorem 3.** *If an ASP program $P$ is stratified, then it has a unique answer set.*

**Proposition 1** (Correctness of the ASP encoding). *Given a finite label trace $\lambda^*$, where $* \in \{G, D, I\}$, and an HRM $H = \langle \mathcal{M}, M_r, \mathcal{P} \rangle$ that is valid with respect to $\lambda^*$, the program $P = \mathbb{A}(H) \cup \mathcal{R} \cup \mathbb{A}(\lambda^*)$ has a unique answer set $AS$ and (1) `accept` $\in AS$ if and only if $* = G$, and (2) `reject` $\in AS$ if and only if $* = D$.*

*Proof.* First, we prove that the program $P = \mathbb{A}(H) \cup \mathcal{R} \cup \mathbb{A}(\lambda^*)$, where $\mathcal{R} = \bigcup_{i=0}^{5} \mathcal{R}_i$, has a unique answer set. By Theorem 3, if $P$ is stratified then it has a unique answer set. We show there is a way of partitioning $P$ following the constraints in Definition 12. A possible partition is $P = P_0 \cup P_1 \cup P_2 \cup P_3$, where $P_0 = \mathbb{A}(\lambda^*)$, $P_1 = \mathbb{A}(H)$, $P_2 = \mathcal{R}_0 \cup \mathcal{R}_1$, $P_3 = \mathcal{R}_2 \cup \mathcal{R}_3 \cup \mathcal{R}_4 \cup \mathcal{R}_5$. The unique answer set $AS = AS_0 \cup AS_1 \cup AS_2 \cup AS_3$, where $AS_i$ corresponds to partition $P_i$, is shown in Figure 8. For simplicity, $\lambda^*[t]$ denotes the $t$-th label in trace $\lambda^*$, $\lambda^*[t :]$ denotes the subtrace starting from the $t$-th label onwards, and $M_i(\lambda^*)$ denotes the hierarchy traversal using RM $M_i$ as the root.

We now prove that `accept` $\in AS$ if and only if $* = G$ (i.e., the trace achieves the goal). If $* = G$ then, since the hierarchy is valid with respect to $\lambda^*$ (see Definition 11), the hierarchy traversal $H(\lambda^*)$ finishes in the accepting state $u^A$ of the root; that is, $H(\lambda^*)[n + 1] = \langle M_r, u_r^A, \cdot, \cdot \rangle$. This holds if and only if `accept` $\in AS$.

The proof showing that `reject` $\in AS$ if and only if $* = D$ (i.e., the trace reaches a dead-end) is similar to the previous one. If $* = D$ then, since the hierarchy is valid with respect to $\lambda^*$, the hierarchy traversal $H(\lambda^*)$ finishes in a rejecting state $u^R$; that is, $H(\lambda^*)[n + 1] = \langle M_k, u^R, \cdot, \cdot \rangle$, where $M_k \in \mathcal{M}$. This holds if and only if `reject` $\in AS$. $\qquad\square$

### E.2 Representation of the HRM Learning Task in ILASP

We here formalize the learning of an HRM and its mapping to a general ILASP learning task. We start by defining the HRM learning task introduced in Section 5.

**Definition 13.** An *HRM learning task* is a tuple $T_H = \langle r, \mathcal{U}, \mathcal{P}, \mathcal{M}, \mathcal{M}_{\mathcal{C}}, u^0, u^A, u^R, \Lambda, \kappa \rangle$, where $r$ is the index of the root RM in the HRM; $\mathcal{U} \supseteq \{u^0, u^A, u^R\}$ is a set of states of the root RM always containing an initial state $u^0$, an accepting state $u^A$, and a rejecting state $u^R$; $\mathcal{P}$ is a set of propositions; $\mathcal{M} \supseteq \{M_\top\}$ is a set of RMs; $\mathcal{M}_{\mathcal{C}} \subseteq \mathcal{M}$ is a set of callable RMs; $\Lambda = \Lambda^G \cup \Lambda^D \cup \Lambda^I$ is a set of label traces; and $\kappa$ is the maximum number of conjunctions/disjuncts in each formula. An HRM $H = \langle \mathcal{M} \cup \{M_r\}, M_r, \mathcal{P} \rangle$ is a solution of $T_H$ if and only if it is valid with respect to all the traces in $\Lambda$.

We make some assumptions about the sets of RMs $\mathcal{M}$: (i) all RMs reachable from RMs in $\mathcal{M}_{\mathcal{C}}$ must be in $\mathcal{M}$, (ii) all RMs in $\mathcal{M}$ are deterministic, and (iii) all RMs in $\mathcal{M}$ are defined over the same set of propositions $\mathcal{P}$ (or a subset of it).

For completeness, we provide the definition of an ILASP task introduced by Law et al. (2016). The first definition corresponds to the form of the examples taken by ILASP, while the second corresponds to the ILASP tasks themselves.

$$AS_0 = \{\texttt{label}(p,t). \mid 0 \le t \le n, p \in \mathcal{L}_t\} \cup \{\texttt{step}(t). \mid 0 \le t \le n\} \cup \{\texttt{last}(n).\},$$

$$AS_1 = \begin{Bmatrix} \{\texttt{state}(u, M_i). \mid M_i \in \mathcal{M} \setminus \{M_\top\}, u \in \mathcal{U}_i\} \cup \\ \left\{\texttt{call}(u, u', x+e, M_i, M_j). \; \middle| \; \begin{array}{l} M_i \in \mathcal{M} \setminus \{M_\top\}, M_j \in \mathcal{M}, u, u' \in \mathcal{U}_i, \varphi_i(u, u', M_j) \ne \bot, \\ x = \sum_{k=0}^{j-1} |\varphi_i(u, u', M_k)|, e \in [1, |\varphi_i(u, u', M_j)|] \end{array}\right\} \cup \\ \left\{\bar\varphi(u, u', x+e, M_i, t). \; \middle| \; \begin{array}{l} 0 \le t \le n, M_i \in \mathcal{M} \setminus \{M_\top\}, M_j \in \mathcal{M}, u, u' \in \mathcal{U}_i, \\ \varphi_i(u, u', M_j) \ne \bot, x = \sum_{k=0}^{j-1} |\varphi_i(u, u', M_k)|, \\ e \in [1, |\varphi_i(u, u', M_j)|], \lambda^*[t] \not\models \varphi_i(u, u', M_j)[e] \end{array}\right\} \end{Bmatrix},$$

$$AS_2 = \begin{Bmatrix} \left\{\varphi(u, u', x+e, M_i, t). \; \middle| \; \begin{array}{l} 0 \le t \le n, M_i \in \mathcal{M} \setminus \{M_\top\}, M_j \in \mathcal{M}, u, u' \in \mathcal{U}_i, \\ \varphi_i(u, u', M_j) \ne \bot, x = \sum_{k=0}^{j-1} |\varphi_i(u, u', M_k)|, \\ e \in [1, |\varphi_i(u, u', M_j)|], \lambda^*[t] \models \varphi_i(u, u', M_j)[e] \end{array}\right\} \cup \\ \{\texttt{pre\_sat}(u, M_i, t). \mid 0 \le t \le n, M_i \in \mathcal{M} \setminus \{M_\top\}, u \in \mathcal{U}_i, \lambda^*[t] \models \xi_{i, u, \top}\} \end{Bmatrix},$$

$$AS_3 = \begin{Bmatrix} \{\texttt{reachable}(u^0, M_r, 0, 0).\} \cup \\ \{\texttt{reachable}(u^0, M_i, t, t). \mid 0 \le t \le n, M_i \in \mathcal{M} \setminus \{M_\top, M_r\}, u^0 \in \mathcal{U}_i\} \cup \\ \left\{\texttt{reachable}(u, M_r, t_1, t_2). \; \middle| \; \begin{array}{l} 0 \le t_1 < t_2 \le n+1, u \in \mathcal{U}_r, \\ H(\lambda^*[t_1:])[t_2 - t_1] = \langle M_r, u, \cdot, \cdot\rangle \end{array}\right\} \cup \\ \left\{\texttt{reachable}(u, M_i, t_1, t_2). \; \middle| \; \begin{array}{l} 0 \le t_1 < t_2 \le n+1, M_i \in \mathcal{M} \setminus \{M_r, M_\top\}, u \in \mathcal{U}_i, \\ \lambda^*[t_1] \models \xi_{i, u^0, \top}, \\ M_i(\lambda^*[t_1:])[t_2 - t_1] = \langle M_i, u, \cdot, \cdot\rangle, \\ M_i(\lambda^*[t_1:])[t_2 - t_1 - 1] \ne \langle M_i, u^A, \cdot, \cdot\rangle \end{array}\right\} \cup \\ \{\texttt{satisfied}(M_r, t_1, t_2) \mid 0 \le t_1 < t_2 \le n+1, H(\lambda^*[t_1:])[t_2 - t_1] = \langle M_r, u^A, \cdot, \cdot\rangle\} \\ \left\{\texttt{satisfied}(M_i, t_1, t_2). \; \middle| \; \begin{array}{l} 0 \le t_1 < t_2 \le n+1, M_i \in \mathcal{M} \setminus \{M_r, M_\top\}, \\ \lambda^*[t_1] \models \xi_{i, u^0, \top}, \\ M_i(\lambda^*[t_1:])[t_2 - t_1] = \langle M_i, u^A, \cdot, \cdot\rangle, \\ M_i(\lambda^*[t_1:])[t_2 - t_1 - 1] \ne \langle M_i, u^A, \cdot, \cdot\rangle \end{array}\right\} \cup \\ \{\texttt{satisfied}(M_\top, t, t+1) \mid 0 \le t \le n\} \cup \\ \{\texttt{failed}(M_r, t_1, t_2) \mid 0 \le t_1 < t_2 \le n+1, H(\lambda^*[t_1:])[t_2 - t_1] = \langle \cdot, u^R, \cdot, \cdot\rangle\} \cup \\ \left\{\texttt{failed}(M_i, t_1, t_2). \; \middle| \; \begin{array}{l} 0 \le t_1 < t_2 \le n+1, M_i \in \mathcal{M} \setminus \{M_r, M_\top\}, \\ \lambda^*[t_1] \models \xi_{i, u^0, \top}, \\ M_i(\lambda^*[t_1:])[t_2 - t_1] = \langle \cdot, u^R, \cdot, \cdot\rangle \end{array}\right\} \cup \\ \left\{\delta(u, u', M_i, t, t+1). \; \middle| \; \begin{array}{l} 0 \le t \le n, M_i \in \mathcal{M} \setminus \{M_\top\}, u, u' \in \mathcal{U}_i, \\ \lambda^*[t_1] \models \varphi_i(u, u', M_\top) \end{array}\right\} \cup \\ \left\{\delta(u, u', M_i, t_1, t_2). \; \middle| \; \begin{array}{l} 0 \le t_1 < t_2 \le n+1, M_i \in \mathcal{M} \setminus \{M_\top\}, u, u' \in \mathcal{U}_i, \\ \exists M_j \in \mathcal{M} \setminus \{M_\top\} \text{ s.t. } \phi = \varphi_i(u, u', M_j), \lambda^*[t_1] \models \xi_{j, u^0, \phi}, \\ M_j(\lambda^*[t_1:])[t_2 - t_1] = \langle M_j, u^A, \cdot, \cdot\rangle, \\ M_j(\lambda^*[t_1:])[t_2 - t_1 - 1] \ne \langle M_j, u^A, \cdot, \cdot\rangle \end{array}\right\} \cup \\ \left\{\delta(u, u^R, M_i, t_1, t_2). \; \middle| \; \begin{array}{l} M_i \in \mathcal{M} \setminus \{M_\top\}, u \in \mathcal{U}_i, 0 \le t_1 < t_2 \le n+1, \\ \exists M_j \in \mathcal{M} \setminus \{M_\top\} \text{ s.t. } \phi = \varphi_i(u, u', M_j), \lambda^*[t_1] \models \xi_{j, u^0, \phi}, \\ M_j(\lambda^*[t_1:])[t_2 - t_1] = \langle M_k, u^R, \cdot, \cdot\rangle, M_k \in \mathcal{M} \end{array}\right\} \cup \\ \{\texttt{accept} \mid H(\lambda^*)[n+1] = \langle M_r, u^A, \cdot, \cdot\rangle\} \cup \\ \{\texttt{reject} \mid H(\lambda^*)[n+1] = \langle M_k, u^R, \cdot, \cdot\rangle, M_k \in \mathcal{M} \setminus \{M_\top\}\} \end{Bmatrix}.$$

Figure 8: Answer sets for each of the partitions in the program $P = \mathbb{A}(H) \cup \mathcal{R} \cup \mathbb{A}(\lambda^*)$, where $H$ is an HRM, $\mathcal{R}$ is the set of general rules and $\lambda^*$ is a label trace.

**Definition 14.** A *context-dependent partial interpretation* (CDPI) is a pair $\langle\langle e^{inc}, e^{exc}\rangle, e^{ctx}\rangle$, where $\langle e^{inc}, e^{exc}\rangle$ is a pair of sets of atoms, called a *partial interpretation*, and $e^{ctx}$ is an ASP program called a *context*. A program $P$ accepts a CDPI $\langle\langle e^{inc}, e^{exc}\rangle, e^{ctx}\rangle$ if and only if there is an answer set $AS$ of $P \cup e^{ctx}$ such that $e^{inc} \subseteq AS$ and $e^{exc} \cap AS = \emptyset$.

**Definition 15.** An *ILASP task* is a tuple $T = \langle\mathcal{B}, \mathcal{S}_M, \langle\mathcal{E}^+, \mathcal{E}^-\rangle\rangle$ where $\mathcal{B}$ is the ASP background knowledge, which describes a set of known concepts before learning; $\mathcal{S}_M$ is the set of ASP rules allowed in the hypotheses; and $\mathcal{E}^+$ and $\mathcal{E}^-$ are sets of CDPIs called, respectively, the positive and negative examples. A hypothesis $\mathcal{H} \subseteq \mathcal{S}_M$ is an *inductive solution* of $T$ if and only if (i) $\forall e \in \mathcal{E}^+$, $\mathcal{B} \cup \mathcal{H}$ accepts $e$, and (ii) $\forall e \in \mathcal{E}^-$, $\mathcal{B} \cup \mathcal{H}$ does not accept $e$.

Given an HRM learning task $T_H$, we map it into an ILASP learning task $\mathbb{A}(T_H) = \langle\mathcal{B}, \mathcal{S}_M, \langle\mathcal{E}^+, \emptyset\rangle\rangle$ and use the ILASP system (Law et al., 2015) to find an inductive solution $\mathbb{A}_\varphi(H) \subseteq \mathcal{S}_M$ that covers the examples. Note that we do not use *negative examples* ($\mathcal{E}^- = \emptyset$). We define the components of $\mathbb{A}(T_H)$ below.

**Background Knowledge.** The background knowledge $\mathcal{B} = \mathcal{B}_\mathcal{U} \cup \mathcal{B}_\mathcal{M} \cup \mathcal{R}$ is a set of rules that describe the behavior of the HRM. The set $\mathcal{B}_\mathcal{U}$ consists of $\texttt{state}(u, M_r)$ facts for each state $u \in \mathcal{U}$ of the root RM with index $r$ we aim to induce, whereas $\mathcal{B}_\mathcal{M} = \bigcup_{M_i \in \mathcal{M}\setminus\{M_\top\}} \mathbb{A}(\mathcal{M}_i)$ contains the ASP representations of all RMs. Finally, $\mathcal{R}$ is the set of general rules introduced in Appendix E.1 that defines how HRMs process label traces. Importantly, the index of the root $r$ in these rules must correspond to the one used in $T_H$.

**Hypothesis Space.** The hypothesis space $\mathcal{S}_M$ contains all $\texttt{ed}$ and $\bar\varphi$ rules that characterize a transition from a non-terminal state $u \in \mathcal{U} \setminus \{u^A, u^R\}$ to a different state $u' \in \mathcal{U} \setminus \{u\}$ using edge $i \in [1, \kappa]$. Formally, it is defined as

$$\mathcal{S}_M = \left\{ \begin{array}{l} \texttt{call}(u, u', i, M). \\ \bar\varphi(u, u', i, M, \texttt{T}) \texttt{:- label}(p, \texttt{T}), \texttt{step}(\texttt{T}). \\ \bar\varphi(u, u', i, M, \texttt{T}) \texttt{:- not label}(p, \texttt{T}), \texttt{step}(\texttt{T}). \end{array} \left| \begin{array}{l} u \in \mathcal{U} \setminus \{u^A, u^R\}, \\ u' \in \mathcal{U} \setminus \{u\}, i \in [1, \kappa], \\ M \in \mathcal{M}_\mathcal{C}, p \in \mathcal{P} \end{array} \right. \right\}.$$

**Example Sets.** Given a set of traces $\Lambda = \Lambda^G \cup \Lambda^D \cup \Lambda^I$, the set of *positive examples* is defined as

$$\mathcal{E}^+ = \{\langle e^*, \mathbb{A}(\lambda)\rangle \mid * \in \{G, D, I\}, \lambda \in \Lambda^*\},$$

where

- $e^G = \langle\{\texttt{accept}\}, \{\texttt{reject}\}\rangle$,
- $e^D = \langle\{\texttt{reject}\}, \{\texttt{accept}\}\rangle$, and
- $e^I = \langle\{\}, \{\texttt{accept}, \texttt{reject}\}\rangle$

are the partial interpretations for goal, dead-end and incomplete traces. The $\texttt{accept}$ and $\texttt{reject}$ atoms express whether a trace is accepted or rejected by the HRM; hence, goal traces must only be accepted, dead-end traces must only be rejected, and incomplete traces cannot be accepted or rejected. Note that the context of each example is the set of ASP facts $\mathbb{A}(\lambda)$ that represents the corresponding trace (see Definition 9).

**Correctness of the Learning Task.** The following theorem captures the correctness of the HRM learning task.

**Theorem 4.** *Given an HRM learning task* $T_H = \langle r, \mathcal{U}, \mathcal{P}, \mathcal{M}, \mathcal{M}_\mathcal{C}, u^0, u^A, u^R, \Lambda, \kappa\rangle$, *an HRM* $H = \langle\mathcal{M} \cup \{M_r\}, M_r, \mathcal{P}\rangle$ *is a solution of* $T_H$ *if and only if* $\mathbb{A}_\varphi(M_r)$ *is an inductive solution of* $\mathbb{A}(T_H) = \langle\mathcal{B}, \mathcal{S}_M, \langle\mathcal{E}^+, \emptyset\rangle\rangle$.

*Proof.* Assume $H$ is a solution of $T_H$.

$\iff$ $H$ is valid with respect to all traces in $\Lambda$ (i.e., $H$ accepts all traces in $\Lambda^G$, rejects all traces in $\Lambda^D$ and does not accept nor reject any trace in $\Lambda^I$).

$\iff$ By Proposition 1, for each trace $\lambda^* \in \Lambda^*$ where $* \in \{G, D, I\}$, $\mathbb{A}(H) \cup \mathcal{R} \cup \mathbb{A}(\lambda^*)$ has a unique answer set $AS$ and (1) $\texttt{accept} \in AS$ if and only if $* = G$, and (2) $\texttt{reject} \in AS$ if and only if $* = D$.

$\Longleftrightarrow$ For each example $e \in \mathcal{E}^+$, $\mathcal{R} \cup \mathbb{A}(H)$ accepts $e$.

$\Longleftrightarrow$ For each example $e \in \mathcal{E}^+$, $\mathcal{B} \cup \mathbb{A}_\varphi(M_r)$ accepts $e$ (the two programs are identical).

$\Longleftrightarrow$ $\mathbb{A}_\varphi(M_r)$ is an inductive solution of $\mathbb{A}(T_H)$. $\qquad\qquad\qquad\qquad\qquad\square$

**Constraints.** We introduce several constraints encoding structural properties of the HRMs we want to learn. Some of these constraints are expressed in terms of facts $\mathtt{pos}(u, u', e, m, p)$ and $\mathtt{neg}(u, u', e, m, p)$, which indicate that proposition $p \in \mathcal{P}$ appears positively (resp. negatively) in edge $e$ from state $u$ to state $u'$ in RM $M_m$. These facts are derived from $\bar\varphi$ rules in $\mathbb{A}(H)$ and injected in the ILASP tasks using meta-program injection (Law et al., 2018).

The following set of constraints ensures that the learned root RM is *deterministic* using the *saturation* technique (Eiter & Gottlob, 1995). The idea is to check determinism top-down by selecting two edges from a given state in the root, each associated with a set of literals. Initially, the set of literals is formed by those in the formula labeling the edges. If a selected edge calls a non-leaf RM, we select an edge from the initial state of the called RM, augment the set of literals with the associated formula, and repeat the process until a call to the leaf RM is reached. We then check if the literal sets are mutually exclusive. If there is a pair of edges from the root that are not mutually exclusive, the solution is discarded. The set of rules is shown below. The first rule states that we keep two saturation IDs, one for each of the edges we select next and for which mutual exclusivity is checked. The second rule chooses a state X in the root, whereas the third rule selects two edges from this state and assigns a saturation ID to each of them. The fourth rule indicates that if one of the edges we have selected so far calls a non-leaf RM, we select one of the edges from the initial state of the called RM and create a new edge with the same saturation ID. The fifth and sixth rules take the propositions for each set of edges (one per saturation ID). The next three rules indicate that if the edges are mutually exclusive (i.e., a proposition appears positively in one set and negatively in the other) or they are the same, then the answer set is saturated. The saturation itself is encoded in the following three rules: an answer set is saturated by adding every possible $\mathtt{ed\_mtx}$ and $\mathtt{root\_point}$ atoms to the answer set. Due to the minimality of answer sets in disjunctive answer set programming, this "maximal" interpretation can only be an answer set if there is no smaller answer set. This will be the case if and only if every choice of edges satisfies the condition (i.e. every choice of $\mathtt{ed\_mtx}$ and $\mathtt{root\_point}$ atoms results in saturation). The constraint encoded in the final rule then discards answer sets in which saturation did not occur, meaning that the remaining solutions must satisfy the condition.

$$
\left\{
\begin{array}{l}
\mathtt{sat\_id}(1; 2).\\
\mathtt{root\_point}(X, M) : \mathtt{call}(X, \_, \_, M, \_), M{=}M_r.\\
\mathtt{ed\_mtx}((X, Y, E, M, M2), SatID) : \mathtt{call}(X, Y, E, M, M2) \mathtt{:-root\_point}(X, M), \mathtt{sat\_id}(SatID).\\
\mathtt{ed\_mtx}((u0, Y2, E2, M2, M3), SatID) : \mathtt{call}(u0, Y2, E2, M2, M3) \mathtt{:-}\, \mathtt{ed\_mtx}((\_, \_, \_, \_, M2), SatID),\\
\hspace{8cm} M2{!}{=}M_\top.\\
\mathtt{pos\_prop}(P, ID) \mathtt{:-} \mathtt{ed\_mtx}((X, Y, E, M, \_), ID), \mathtt{pos}(X, Y, E, M, P).\\
\mathtt{neg\_prop}(P, ID) \mathtt{:-} \mathtt{ed\_mtx}((X, Y, E, M, \_), ID), \mathtt{neg}(X, Y, E, M, P).\\
\mathtt{saturate} \mathtt{:-} \mathtt{pos\_prop}(P, 1), \mathtt{neg\_prop}(P, 2).\\
\mathtt{saturate} \mathtt{:-} \mathtt{pos\_prop}(P, 2), \mathtt{neg\_prop}(P, 1).\\
\mathtt{saturate} \mathtt{:-} \mathtt{ed\_mtx}((X, Y, \_, M, M2), 1), \mathtt{ed\_mtx}((X, Y, \_, M, M2), 2), \mathtt{root\_point}(X, M).\\
\mathtt{root\_point}(X, M) \mathtt{:-} \mathtt{call}(X, \_, \_, M, \_), \mathtt{saturate}, M{=}M_r.\\
\mathtt{ed\_mtx}((X, Y, E, M, M2), SatID) \mathtt{:-} \mathtt{call}(X, Y, E, M, M2), M{=}M_r, \mathtt{sat\_id}(SatID), \mathtt{saturate}.\\
\mathtt{ed\_mtx}((u0, Y, E, M, M2), SatID) \mathtt{:-} \mathtt{call}(u0, Y, E, M, M2), \mathtt{sat\_id}(SatID), \mathtt{saturate}.\\
\mathtt{:- not\ saturate}.
\end{array}
\right\}
$$

Other required constraints to learn sensible HRMs are shown below. The first rule prevents an edge from being labeled with calls to two different RMs. The second rule prevents edges from being labeled with the same literal both positively and negatively.

$$
\left\{
\begin{array}{l}
\mathtt{:- call}(X, Y, E, M, M2), \mathtt{call}(X, Y, E, M, M3), M2{!}{=}M3.\\
\mathtt{:- pos}(X, Y, E, M, P), \mathtt{neg}(X, Y, E, M, P).
\end{array}
\right\}
$$

The following constraints are used to speed up the learning of an HRM. First, we extend the *symmetry breaking* method by Furelos-Blanco et al. (2021), originally proposed for flat RMs, to our hierarchical setting. The main advantage of this method is that it accelerates learning without restricting the family of learnable HRMs. Other constraints analogous to those in previous work (Furelos-Blanco et al., 2021) that speed up the learning process further are enumerated below. For simplicity,

some of these constraints use the auxiliary rule below to define the $\mathtt{ed(X, Y, E, M)}$ predicate, which is equivalent to the $\mathtt{call(X, Y, E, M, M2)}$ predicate but omitting the called RM:

$$\mathtt{ed(X, Y, E, M)\,{:}\text{-}\,call(X, Y, E, M, \_)}.$$

The constraints are the following:

- Rule out inductive solutions where an edge calling the leaf $M_\top$ is labeled by a formula formed only by negated propositions. The rule below enforces a proposition to occur positively whenever a proposition appears negatively in an edge calling $M_\top$.

$$\mathtt{{:}\text{-}\,neg(X, Y, E, M, \_), not\ pos(X, Y, E, M, \_), call(X, Y, E, M,}M_\top).$$

- Rule out any inductive solution where an edge from $\mathtt{X}$ to $\mathtt{Y}$ with index $\mathtt{E}$ is not labeled by a positive or a negative literal. This rule only applies to calls to the leaf $M_\top$, thus avoiding unconditional transitions.

$$\mathtt{{:}\text{-}\,not\ pos(X, Y, E, M, \_), not\ neg(X, Y, E, M, \_), call(X, Y, E, M,}\ M_\top).$$

- Rule out inductive solutions containing states different from the accepting and rejecting states without outgoing edges. In general, these states are not interesting.

$$\left\{ \begin{array}{l} \mathtt{has\_outgoing\_edges(X, M)\,{:}\text{-}\,ed(X, \_, \_, M)}. \\ \mathtt{{:}\text{-}\,state(X, M), not\ has\_outgoing\_edges(X, M), X!{=}}u^A\mathtt{, X!{=}}u^R. \end{array} \right\}$$

- Rule out inductive solutions containing cycles; that is, solutions where two states can be reached from each other. The $\mathtt{path(X, Y, M)}$ predicate indicates there is a directed path (i.e., a sequence of directed edges) from $\mathtt{X}$ to $\mathtt{Y}$ in RM $\mathtt{M}$. The first rule states that there is a path from $\mathtt{X}$ to $\mathtt{Y}$ if there is an edge from $\mathtt{X}$ to $\mathtt{Y}$. The second rule indicates that there is a path from $\mathtt{X}$ to $\mathtt{Y}$ if there is an edge from $\mathtt{X}$ to an intermediate state $\mathtt{Z}$ from which there is a path to $\mathtt{Y}$. Finally, the third rule discards the solutions where $\mathtt{X}$ and $\mathtt{Y}$ can be reached from each other through directed edges.

$$\left\{ \begin{array}{l} \mathtt{path(X, Y, M)\,{:}\text{-}\,ed(X, Y, \_, M)}. \\ \mathtt{path(X, Y, M)\,{:}\text{-}\,ed(X, Z, \_, M), path(Z, Y, M)}. \\ \mathtt{{:}\text{-}\,path(X, Y, M), path(Y, X, M)}. \end{array} \right\}$$

# F  EXPERIMENTAL DETAILS

In this section, we describe the details of the experiments introduced in Section 6. We discuss how the domains are implemented, the hyperparameters used to run the algorithms, and provide all specific results through tables and plots. All experiments ran on 3.40GHz Intel® Core™ i7-6700 processors.

## F.1  DOMAINS

The CRAFTWORLD domain is based on MiniGrid (Chevalier-Boisvert et al., 2018), thus inheriting many of its features. At each step, the agent observes a $W \times H \times 3$ tensor, where $W$ and $H$ are the width and height of the grid. The three channels contain the object IDs, the color IDs, and object state IDs (including the orientation of the agent) respectively. Each of the objects we define (except for the lava ♦, which already existed in MiniGrid) has its own object and color IDs. Before providing the agent with the state, the content of all matrices is scaled between -1 and 1. Note that even though the agent gets a full view of the grid, it is still unaware of the completion degree of a task. Other works have previously used the full view of the grid (Igl et al., 2019; Jiang et al., 2021).

The grids are randomly generated. In all settings (OP, OPL, FR, FRL), the agent and the objects are randomly assigned an unoccupied position. In the case of FR and FRL, no object occupies a position between rooms nor its adjoining positions. There is a single object per object type (i.e., proposition) in OP and OPL, whereas there can be one or two per type in FR and FRL. Finally, there is a single lava location in OPL, which is randomly assigned (like the rest of the propositions), whereas in FRL there are four fixed lava locations placed in the intersections between doors as shown in Figure 9.

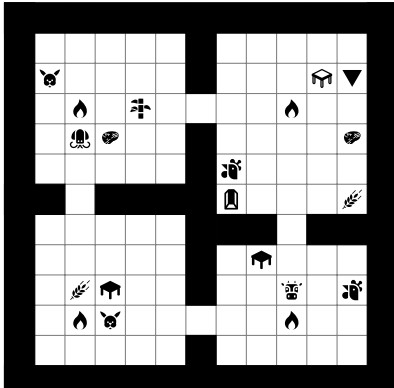

Figure 9: An instance of the CRAFTWORLD grid in the FRL setting.

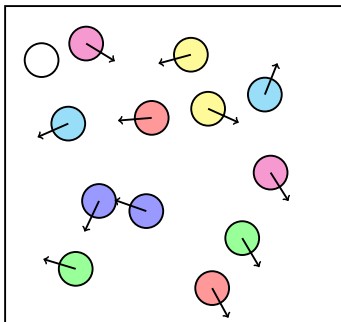

Figure 10: An instance of the WATERWORLD in the WOD setting (Toro Icarte et al., 2018).

The WATERWORLD domain (cf. Figure 10) has a continuous state space. The states are vectors containing the absolute position and velocity of the agent, and the relative positions and velocities of the other balls. The agent does not know the color of each ball. In all settings (WOD and WD), a WATERWORLD instance is created by assigning a random position and direction to each ball. Like in CRAFTWORLD, the agent does not know the degree of completion of a task.

### F.2 HYPERPARAMETERS

Table 2 lists some of the hyperparameters used in the experiments. The rest of the hyperparameters and other details (e.g., architectures, evaluation of other methods) are discussed in the following paragraphs.

**Architectures.** The DQNs for CRAFTWORLD consist of a 3-layer convolutional neural network (CNN) with 16, 32 and 32 filters respectively. All kernels are $2 \times 2$ and use a stride of 1. In the FR and FRL settings, there is a max pooling layer with kernel size $2 \times 2$ after the first convolutional layer. This part of the architecture is based on that by Igl et al. (2019) and Jiang et al. (2021), who also work on MiniGrid using the full view of the grid. In DQNs associated with formulas, the CNN's output is fed to a 3-layer multilayer perceptron (MLP) where the hidden layer has 256 rectifier units and the output layer has a single output for each action. In the case of DQNs for RMs, the output of the CNN is extended with the encoding of the RM state and the context (as discussed in Appendix C) before being fed to a 3-layer MLP where the hidden layer has 256 rectifier units and the output layer has a single output for each call in the RM.

The architecture for WATERWORLD is a simple modification of the one introduced by Toro Icarte et al. (2018). The formula DQNs consist of a 5-layer MLP, where each of the 3 hidden layers has 512 rectifier units. The DQN for the RMs share same architecture and, like in CRAFTWORLD, the state from the environment is extended with the state and context encodings.

Table 2: List of hyperparameters and their values.

| Parameter | CRAFTWORLD | WATERWORLD |
|---|---|---|
| *General* | | |
| Episodes | 300,000 | 300,000 |
| Maximum episode length | 1,000 | 1,000 |
| Num. of instances $I$ | 10 | 10 |
| *DQNs* | | |
| Learning rate $\alpha$ | $5 \times 10^{-4}$ | $1 \times 10^{-5}$ |
| Learning rate (SMDP) $\alpha$ | $5 \times 10^{-4}$ | $1 \times 10^{-3}$ |
| Optimizer | RMSprop (Hinton et al., 2012) | RMSprop (Hinton et al., 2012) |
| Discount $\gamma$ | 0.9 | 0.9 |
| Discount (SMDP) $\gamma$ | 0.99 | 0.99 |
| Updated formula Q-functions per step | 4 | 4 |
| Replay memory size | 500,000 | 500,000 |
| Replay start size | 100,000 | 100,000 |
| Target network update frequency | 1,500 | 1,500 |
| Replay memory size (SMDP) | 10,000 | 10,000 |
| Replay start size (SMDP) | 1,000 | 1,000 |
| Target network update frequency (SMDP) | 500 | 500 |
| Minibatch size | 32 | 32 |
| *Exploration* | | |
| Initial exploration | 1.0 | 1.0 |
| Final exploration | 0.1 | 0.1 |
| Annealing steps | 2,000,000 | 5,000,000 |
| Annealing steps (SMDP) | 10,000 | 10,000 |
| *HRM learning* | | |
| Curriculum weight $\beta$ | 0.99 | 0.99 |
| Curriculum threshold | 0.85 | 0.75 |
| ILASP time budget | 2 hours | 2 hours |
| Num. collected goal traces $\rho$ (height 1) | 25 | 25 |
| Num. collected goal traces $\rho$ (height $\geq 2$) | 150 | 150 |
| Num. goal traces $\rho_s$ to learn first HRM | 10 | 10 |

**Compression.** Akin to some methods for learning RMs (Furelos-Blanco et al., 2021; Toro Icarte et al., 2019), we compress label traces by merging consecutive equal labels into a single one. For instance, $\langle\{\}, \{◨\}, \{◨\}, \{\}, \{\}, \{♠\}, \{☃\}\rangle$ becomes $\langle\{\}, \{◨\}, \{\}, \{♠\}, \{☃\}\rangle$.

**Curriculum.** The average undiscounted returns (see Section 5 and Appendix D) are updated for each task-instance pair every 100 training episodes using the undiscounted return obtained by the greedy policies in a single evaluation episode.

**Flat HRM Learning Methods.** We briefly describe the methods used to learn flat HRMs. Each run consists of 150,000 episodes, and the set of instances is exactly the same across methods. The core difference with respect to learning non-flat HRMs is that there is a single task for which the HRM is learned. Our method, LHRM, is therefore not able to reuse previously learned HRMs for other tasks; however, it still uses the same hyperparameters. In the case of DeepSynth (Hasanbeig et al., 2021), LRM (Toro Icarte et al., 2019) and JIRP (Xu et al., 2020), we exclusively evaluate their RM learning components using traces collected through random walks. For a fair comparison against LHRM (both in the non-flat and flat learning cases), we (i) compress the traces using the aforementioned methodology, and (ii) use the OP and WOD settings of CRAFTWORLD and WATERWORLD respectively, where observing goal traces is very easy for simple tasks such as MILKBUCKET. In these approaches, a different instance is selected at each episode following a cyclic order (i.e., 1, 2,..., $I - 1$, $I$, 1, 2, ...). The set of propositions in these approaches includes a proposition covering the case where no other propositions are observed (if needed). In the case of LRM, one of the parameters is the maximum number of RM states, which we set to that of the minimal RM. Finally, we modify DeepSynth to avoid periodically calling the learner (i.e., only call it when a counterexample

trace is observed): this is not done in other approaches and usually causes timeouts unnecessarily (the same RM is repeatedly learned).[5]

**ILASP.** We use ILASP2 (Law et al., 2015) to learn the HRMs.[6] For efficiency, the default calls to the underlying ASP solver are modified to be made with the flag `---opt-mode=ignore`, meaning that non-minimal solutions might be obtained (i.e., solutions involving more rules than needed), so the learned root might contain some unnecessary edges. In practice, the solutions produced by ILASP rarely contain such edges and, if they do, these edges eventually disappear by observing an appropriate counterexample. We hypothesize that using this flag helps since no optimization is made every time ILASP is called. The design of the ILASP tasks is discussed in Appendix E.2. We highlight that this notion of minimality is not related to that of a minimal RM (i.e., an RM with the fewest number of states) described in Section 5.

### F.3 EXTENDED RESULTS

We organize the tables and figures following the structure in Section 6.

**Policy Learning.** Figure 11 shows the plots omitted in the main paper (the remaining CRAFT-WORLD setting, FRL, is shown in Figure 2). The number of plotted episodes varies across domains and tasks for clarity. Figure 12 shows the learning curves for a task called TENPAPERS, which consists of performing PAPER a total of 10 times.[7] The plot shows that exploiting the non-flat HRM leads to much faster convergence than the equivalent flat one. The difference arises from the fact that the non-flat HRM can reuse the policies for the called PAPER RM, whereas the flat one cannot.

**Learning Non-Flat HRMs.** We present tables containing the results for the HRM learning component of LHRM. The content of the columns is the following left-to-right: (1) task name; (2) number of runs in which at least one goal trace was observed; (3) number of runs in which at least one HRM was learned; (4) time spent to learn the HRMs; (5) number of calls to ILASP made to learn the HRMs; (6) number of states of the final HRM; (7) number of edges of the final HRM; (8) number of episodes between the learning of the first HRM and the activation of the task's level; (9) number of example traces of a given type (G = goal, D = dead-end, I = incomplete); and (10) length of the example traces of a given type. In addition, the bottom of the tables contains the number of completed runs (i.e., the number of runs that have not timed out), the total time spent on learning the HRMs, and the total number of calls made to ILASP. In the case of CRAFTWORLD, Table 3 shows the results for the default case (all lower level RMs are callable and options are used for exploration), Table 4 shows the results when the set of callable RMs contains only those actually needed, and Table 5 shows the results using primitive actions for exploration instead of options. Analogous results are shown for WATERWORLD in Tables 6, 7 and 8. The discussion of these results can be found in Section 6.

**Learning Flat HRMs.** Table 9 shows the results of learning a non-flat HRM using LHRM, and the results of learning a flat HRM using several approaches (LHRM, DeepSynth, LRM and JIRP). An extended discussion of these results can be found in Section 6.

## G EXAMPLES OF HIERARCHIES OF REWARD MACHINES

Figures 14 and 15 show the minimal root RMs for the CRAFTWORLD and WATERWORLD tasks, respectively. In the case of CRAFTWORLD, since two or more propositions can never occur simultaneously, the mutual exclusivity between formulas could be enforced differently. These RMs correspond to the settings without dead-ends; thus, they do not include rejecting states.

---

[5]The code of these methods is publicly available: DeepSynth (`https://github.com/grockious/deepsynth`, MIT license), LRM (`https://bitbucket.org/RToroIcarte/lrm`, no license), and JIRP (`https://github.com/corazza/stochastic-reward-machines`, no license). The first two links can be found in the papers, whereas the last one was provided by one of the authors through personal communication.

[6]ILASP is free to use for research and education (see `https://www.ilasp.com/terms`).

[7]Disclaimer: At the time of writing this revised version (November 10), this experiment consisting of 5 runs has not finished. However, we are confident the observations we have made still will hold when it is completed.

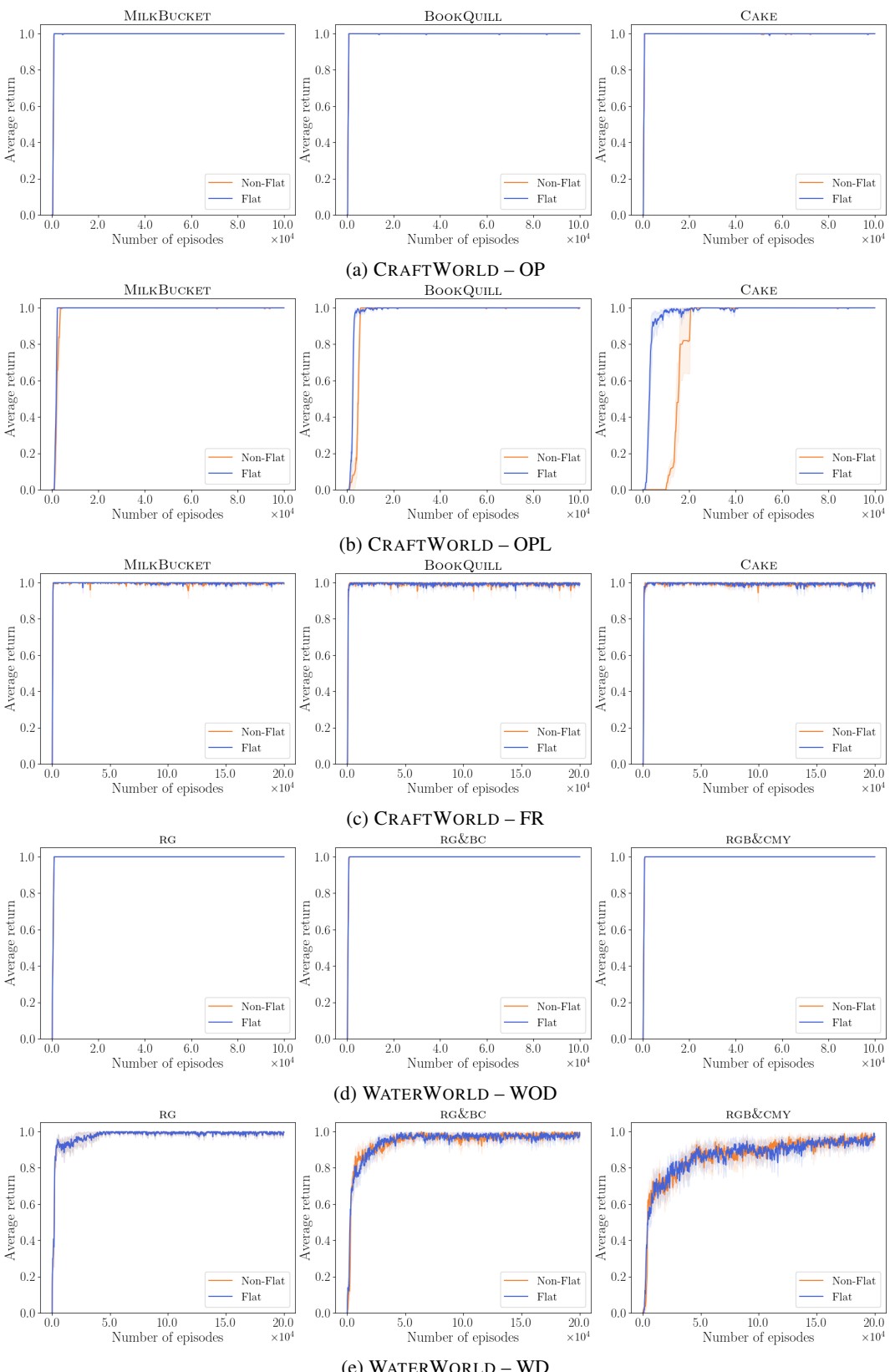

Figure 11: Learning curves comparing the performance of handcrafted non-flat and flat HRMs.

Table 3: Results of LHRM in CRAFTWORLD for the default case.

(a) OP

| Task | # G | # L | Time (s.) | Calls | States | Edges | Ep. First HRM ($\times 10^2$) | # Examples | | Example Length | |
|---|---|---|---|---|---|---|---|---|---|---|---|
| | | | | | | | | G | I | G | I |
| BATTER | 5 | 5 | 11.1 (1.7) | 17.8 (1.9) | 5.0 (0.0) | 5.2 (0.2) | 1.8 (0.1) | 12.2 (0.7) | 11.6 (1.4) | 26.5 (2.1) | 24.2 (3.2) |
| BUCKET | 5 | 5 | 0.9 (0.0) | 3.6 (0.2) | 3.0 (0.0) | 2.0 (0.0) | 1.7 (0.1) | 10.0 (0.0) | 1.6 (0.2) | 19.4 (1.1) | 19.3 (5.7) |
| COMPASS | 5 | 5 | 135.4 (73.3) | 18.6 (1.6) | 5.0 (0.0) | 5.2 (0.2) | 1.8 (0.2) | 11.8 (0.6) | 12.8 (1.4) | 28.7 (1.9) | 20.3 (2.8) |
| LEATHER | 5 | 5 | 0.9 (0.0) | 3.8 (0.2) | 3.0 (0.0) | 2.0 (0.0) | 1.8 (0.1) | 10.0 (0.0) | 1.8 (0.2) | 16.7 (1.7) | 17.9 (4.4) |
| PAPER | 5 | 5 | 0.8 (0.1) | 3.4 (0.2) | 3.0 (0.0) | 2.0 (0.0) | 1.6 (0.1) | 10.0 (0.0) | 1.4 (0.2) | 19.8 (2.0) | 40.6 (27.0) |
| QUILL | 5 | 5 | 18.0 (3.5) | 19.8 (1.2) | 5.0 (0.0) | 5.2 (0.2) | 2.1 (0.1) | 13.2 (0.4) | 12.6 (1.1) | 29.6 (2.5) | 24.4 (3.2) |
| SUGAR | 5 | 5 | 0.8 (0.1) | 3.2 (0.2) | 3.0 (0.0) | 2.0 (0.0) | 1.7 (0.2) | 10.0 (0.0) | 1.2 (0.2) | 17.7 (1.6) | 17.5 (3.2) |
| BOOK | 5 | 5 | 191.2 (36.4) | 22.8 (2.6) | 5.0 (0.0) | 5.8 (0.2) | 6.0 (0.2) | 11.4 (0.7) | 17.4 (2.2) | 20.5 (1.8) | 24.8 (1.5) |
| MAP | 5 | 5 | 549.4 (149.5) | 33.4 (3.2) | 5.0 (0.0) | 5.6 (0.2) | 6.0 (0.2) | 12.2 (0.6) | 27.2 (2.9) | 29.5 (3.2) | 28.7 (1.7) |
| MILKBUCKET | 5 | 5 | 1.5 (0.2) | 4.6 (0.4) | 3.0 (0.0) | 2.0 (0.0) | 6.8 (0.5) | 10.0 (0.0) | 2.6 (0.4) | 11.6 (0.7) | 15.3 (4.3) |
| BOOKQUILL | 5 | 5 | 17.9 (1.4) | 19.6 (1.1) | 4.0 (0.0) | 4.0 (0.0) | 3.8 (0.1) | 10.0 (0.0) | 16.6 (1.1) | 27.2 (1.3) | 20.8 (1.4) |
| MILKB.SUGAR | 5 | 5 | 7.3 (1.2) | 12.4 (1.2) | 4.0 (0.0) | 4.0 (0.0) | 3.8 (0.1) | 10.2 (0.2) | 9.2 (1.2) | 16.9 (0.8) | 14.3 (1.7) |
| CAKE | 5 | 5 | 74.5 (25.7) | 26.4 (3.7) | 4.0 (0.0) | 3.2 (0.2) | 2.1 (0.1) | 10.2 (0.2) | 23.2 (3.6) | 38.4 (0.9) | 22.7 (1.6) |
| Completed Runs | | 5 | | | | | | | | | |
| Total Time (s.) | | 1009.8 (122.3) | | | | | | | | | |
| Total Calls | | 189.4 (4.1) | | | | | | | | | |

(b) OPL

| Task | # G | # L | Time (s.) | Calls | States | Edges | Ep. First HRM ($\times 10^2$) | # Examples | | | Example Length | | |
|---|---|---|---|---|---|---|---|---|---|---|---|---|---|
| | | | | | | | | G | D | I | G | D | I |
| BATTER | 5 | 5 | 13.7 (2.9) | 23.0 (3.0) | 6.0 (0.0) | 9.2 (0.2) | 12.0 (1.0) | 11.4 (0.4) | 7.0 (1.2) | 10.6 (1.6) | 20.4 (1.1) | 18.7 (1.6) | 12.1 (1.7) |
| BUCKET | 5 | 5 | 1.8 (0.2) | 7.2 (0.6) | 4.0 (0.0) | 4.0 (0.0) | 8.0 (0.5) | 10.2 (0.2) | 2.2 (0.2) | 2.8 (0.4) | 10.2 (0.5) | 13.4 (1.9) | 6.8 (1.7) |
| COMPASS | 5 | 5 | 13.1 (1.7) | 22.0 (1.7) | 6.0 (0.0) | 9.2 (0.2) | 10.4 (1.4) | 11.0 (0.6) | 6.8 (1.0) | 10.2 (1.0) | 17.2 (1.6) | 20.9 (1.9) | 14.3 (0.8) |
| LEATHER | 5 | 5 | 1.9 (0.2) | 7.0 (0.5) | 4.0 (0.0) | 4.0 (0.0) | 6.9 (0.5) | 10.0 (0.0) | 2.4 (0.2) | 2.6 (0.4) | 11.1 (0.9) | 16.9 (5.6) | 8.9 (3.3) |
| PAPER | 5 | 5 | 2.0 (0.2) | 7.6 (0.6) | 4.0 (0.0) | 4.0 (0.0) | 7.7 (1.1) | 10.0 (0.0) | 3.0 (0.3) | 2.6 (0.4) | 10.1 (0.9) | 18.9 (3.3) | 5.6 (0.8) |
| QUILL | 5 | 5 | 11.3 (1.2) | 22.0 (1.2) | 6.0 (0.0) | 9.2 (0.2) | 12.8 (1.5) | 10.6 (0.2) | 6.4 (0.7) | 11.0 (0.9) | 15.3 (1.3) | 13.5 (1.0) | 12.1 (1.4) |
| SUGAR | 5 | 5 | 1.7 (0.1) | 6.4 (0.4) | 4.0 (0.0) | 4.0 (0.0) | 6.5 (0.7) | 10.0 (0.0) | 2.4 (0.2) | 2.0 (0.3) | 9.6 (0.6) | 15.3 (3.6) | 16.6 (9.2) |
| BOOK | 5 | 5 | 427.8 (201.6) | 32.6 (4.2) | 6.0 (0.0) | 6.6 (0.2) | 5.6 (0.2) | 12.0 (0.3) | 3.6 (0.7) | 23.0 (3.4) | 21.6 (1.5) | 25.9 (3.4) | 23.7 (1.3) |
| MAP | 5 | 5 | 647.9 (110.7) | 38.6 (3.6) | 6.0 (0.0) | 6.4 (0.2) | 5.6 (0.2) | 11.2 (0.4) | 3.8 (0.9) | 29.6 (3.5) | 23.1 (1.0) | 27.8 (4.6) | 26.1 (0.4) |
| MILKBUCKET | 5 | 5 | 2.1 (0.2) | 5.4 (0.4) | 4.0 (0.0) | 3.0 (0.0) | 7.6 (0.5) | 10.0 (0.0) | 1.4 (0.4) | 2.0 (0.0) | 11.1 (0.5) | 26.3 (6.5) | 15.2 (5.8) |
| BOOKQUILL | 5 | 5 | 18.7 (2.3) | 16.6 (1.3) | 4.0 (0.0) | 4.0 (0.0) | 3.7 (0.2) | 10.0 (0.0) | 0.4 (0.2) | 13.2 (1.4) | 29.0 (1.1) | 6.2 (5.5) | 27.8 (1.4) |
| MILKB.SUGAR | 5 | 5 | 7.7 (0.7) | 12.2 (0.9) | 4.0 (0.0) | 4.0 (0.0) | 3.8 (0.2) | 10.0 (0.0) | 0.2 (0.2) | 9.0 (0.9) | 16.0 (0.9) | 1.6 (1.6) | 16.3 (1.3) |
| CAKE | 5 | 5 | 472.9 (216.6) | 36.0 (6.0) | 5.0 (0.0) | 4.6 (0.2) | 2.1 (0.0) | 10.0 (0.0) | 1.6 (0.4) | 31.4 (5.7) | 39.5 (1.2) | 41.5 (8.6) | 26.9 (0.8) |
| Completed Runs | | 5 | | | | | | | | | | | |
| Total Time (s.) | | 1622.6 (328.7) | | | | | | | | | | | |
| Total Calls | | 236.6 (9.3) | | | | | | | | | | | |

(c) FR

| Task | # G | # L | Time (s.) | Calls | States | Edges | Ep. First HRM ($\times 10^2$) | # Examples | | Example Length | |
|---|---|---|---|---|---|---|---|---|---|---|---|
| | | | | | | | | G | I | G | I |
| BATTER | 5 | 5 | 12.3 (1.7) | 17.6 (1.3) | 5.0 (0.0) | 5.4 (0.2) | 9.2 (1.2) | 11.6 (0.4) | 12.0 (1.2) | 30.3 (2.3) | 27.8 (2.0) |
| BUCKET | 5 | 5 | 1.2 (0.1) | 3.8 (0.2) | 3.0 (0.0) | 2.0 (0.0) | 6.7 (0.9) | 10.0 (0.0) | 1.8 (0.2) | 16.6 (2.5) | 28.5 (4.1) |
| COMPASS | 5 | 5 | 14.1 (1.6) | 20.2 (1.7) | 5.0 (0.0) | 5.2 (0.2) | 9.8 (0.7) | 11.6 (0.6) | 14.6 (1.2) | 26.5 (0.8) | 26.5 (2.1) |
| LEATHER | 5 | 5 | 1.1 (0.1) | 3.6 (0.2) | 3.0 (0.0) | 2.0 (0.0) | 4.5 (0.7) | 10.0 (0.0) | 1.6 (0.2) | 13.4 (1.3) | 16.7 (3.6) |
| PAPER | 5 | 5 | 1.2 (0.0) | 4.0 (0.0) | 3.0 (0.0) | 2.0 (0.0) | 4.9 (0.9) | 10.0 (0.0) | 2.0 (0.0) | 12.4 (1.1) | 10.9 (2.5) |
| QUILL | 5 | 5 | 8.9 (0.9) | 16.0 (0.8) | 5.0 (0.0) | 5.2 (0.2) | 9.4 (1.7) | 10.6 (0.2) | 11.4 (0.6) | 25.4 (0.3) | 25.5 (2.7) |
| SUGAR | 5 | 5 | 1.1 (0.1) | 3.8 (0.2) | 3.0 (0.0) | 2.0 (0.0) | 5.2 (0.3) | 10.0 (0.0) | 1.8 (0.2) | 15.3 (1.7) | 21.0 (10.1) |
| BOOK | 5 | 5 | 220.2 (83.3) | 25.2 (3.4) | 5.0 (0.0) | 5.6 (0.2) | 6.1 (0.2) | 10.2 (0.2) | 21.0 (3.4) | 21.9 (1.0) | 18.4 (0.7) |
| MAP | 5 | 5 | 628.3 (85.4) | 37.8 (3.7) | 5.0 (0.0) | 5.6 (0.2) | 5.8 (0.1) | 10.0 (0.0) | 33.8 (3.7) | 26.4 (1.0) | 21.4 (0.7) |
| MILKBUCKET | 5 | 5 | 1.9 (0.2) | 5.0 (0.3) | 3.0 (0.0) | 2.0 (0.0) | 9.8 (0.7) | 10.0 (0.0) | 3.0 (0.3) | 13.2 (0.7) | 12.8 (3.2) |
| BOOKQUILL | 5 | 5 | 12.9 (2.2) | 15.6 (1.7) | 4.0 (0.0) | 4.0 (0.0) | 3.9 (0.1) | 10.0 (0.0) | 12.6 (1.7) | 29.0 (1.5) | 13.3 (0.8) |
| MILKB.SUGAR | 5 | 5 | 7.2 (0.6) | 12.0 (0.7) | 4.0 (0.0) | 4.0 (0.0) | 3.9 (0.2) | 10.0 (0.0) | 9.0 (0.7) | 18.9 (0.9) | 10.1 (1.0) |
| CAKE | 5 | 5 | 121.1 (41.1) | 34.0 (4.8) | 4.0 (0.0) | 3.0 (0.0) | 2.2 (0.0) | 10.0 (0.0) | 31.0 (4.8) | 42.2 (1.7) | 16.2 (1.1) |
| Completed Runs | | 5 | | | | | | | | | |
| Total Time (s.) | | 1031.6 (150.3) | | | | | | | | | |
| Total Calls | | 198.6 (11.3) | | | | | | | | | |

(d) FRL

| Task | # G | # L | Time (s.) | Calls | States | Edges | Ep. First HRM ($\times 10^2$) | # Examples | | | Example Length | | |
|---|---|---|---|---|---|---|---|---|---|---|---|---|---|
| | | | | | | | | G | D | I | G | D | I |
| BATTER | 5 | 5 | 11.3 (1.4) | 23.4 (2.5) | 6.0 (0.0) | 9.2 (0.2) | 468.4 (121.9) | 10.4 (0.2) | 7.6 (0.9) | 11.4 (1.9) | 11.9 (0.6) | 10.1 (1.3) | 9.9 (0.4) |
| BUCKET | 5 | 5 | 2.3 (0.2) | 7.0 (0.3) | 4.0 (0.0) | 4.0 (0.0) | 129.5 (69.4) | 10.2 (0.2) | 2.8 (0.2) | 2.0 (0.3) | 7.8 (0.5) | 9.9 (1.7) | 6.4 (2.1) |
| COMPASS | 5 | 5 | 13.0 (1.9) | 24.6 (2.2) | 6.0 (0.0) | 9.4 (0.2) | 550.8 (156.4) | 10.4 (0.2) | 7.8 (1.0) | 12.4 (1.2) | 12.5 (1.6) | 9.4 (1.0) | 8.4 (0.5) |
| LEATHER | 5 | 5 | 2.5 (0.3) | 7.8 (0.7) | 4.0 (0.0) | 4.0 (0.0) | 89.0 (18.0) | 10.0 (0.0) | 3.2 (0.4) | 2.6 (0.4) | 7.3 (0.4) | 9.3 (1.7) | 3.7 (0.4) |
| PAPER | 5 | 5 | 2.2 (0.1) | 7.0 (0.3) | 4.0 (0.0) | 4.0 (0.0) | 82.7 (18.8) | 10.0 (0.0) | 3.0 (0.3) | 2.0 (0.3) | 6.9 (0.7) | 10.2 (1.8) | 4.7 (2.7) |
| QUILL | 5 | 5 | 11.6 (1.1) | 23.8 (1.5) | 6.0 (0.0) | 9.6 (0.2) | 458.9 (61.0) | 10.6 (0.2) | 8.0 (0.9) | 11.2 (1.2) | 11.9 (0.6) | 13.1 (2.7) | 9.2 (0.8) |
| SUGAR | 5 | 5 | 2.7 (0.2) | 8.4 (0.7) | 4.0 (0.0) | 4.0 (0.0) | 103.5 (39.5) | 10.0 (0.0) | 3.6 (0.4) | 2.8 (0.5) | 8.2 (0.7) | 10.1 (1.9) | 5.0 (1.1) |
| BOOK | 5 | 5 | 301.7 (98.1) | 36.4 (1.9) | 6.0 (0.0) | 6.8 (0.2) | 5.3 (0.1) | 10.2 (0.2) | 5.0 (0.7) | 27.2 (1.9) | 21.7 (1.1) | 18.8 (2.2) | 16.1 (0.6) |
| MAP | 5 | 5 | 754.1 (158.2) | 44.6 (2.6) | 6.0 (0.0) | 7.0 (0.0) | 5.5 (0.2) | 10.4 (0.2) | 5.2 (0.4) | 35.2 (2.3) | 25.6 (0.5) | 20.4 (2.9) | 18.7 (0.6) |
| MILKBUCKET | 5 | 5 | 2.8 (0.1) | 6.6 (0.2) | 4.0 (0.0) | 3.0 (0.0) | 6.9 (0.4) | 10.0 (0.0) | 2.0 (0.0) | 2.6 (0.2) | 12.5 (0.8) | 13.1 (3.7) | 7.4 (2.2) |
| BOOKQUILL | 5 | 5 | 19.8 (2.9) | 19.6 (1.6) | 4.0 (0.0) | 4.0 (0.0) | 4.3 (0.1) | 10.0 (0.0) | 0.8 (0.4) | 15.8 (1.2) | 28.4 (1.1) | 2.7 (1.3) | 13.5 (0.9) |
| MILKB.SUGAR | 5 | 5 | 8.8 (0.9) | 12.6 (1.0) | 4.0 (0.0) | 4.0 (0.0) | 4.0 (0.1) | 10.0 (0.0) | 1.2 (0.5) | 8.4 (0.7) | 19.3 (1.3) | 3.7 (2.0) | 10.7 (2.0) |
| CAKE | 5 | 5 | 344.0 (87.7) | 46.2 (4.9) | 5.0 (0.0) | 4.8 (0.2) | 2.8 (0.1) | 10.0 (0.0) | 2.8 (0.7) | 40.4 (4.5) | 44.5 (2.3) | 21.8 (2.2) | 17.3 (1.0) |
| Completed Runs | | 5 | | | | | | | | | | | |
| Total Time (s.) | | 1476.8 (175.3) | | | | | | | | | | | |
| Total Calls | | 268.0 (6.5) | | | | | | | | | | | |

Table 4: Results of LHRM in CRAFTWORLD with a restricted set of callable RMs.

(a) OP

| Task | # G | # L | Time (s.) | Calls | States | Edges | Ep. First HRM ($\times 10^2$) | # Examples G | # Examples I | Example Length G | Example Length I |
|---|---|---|---|---|---|---|---|---|---|---|---|
| BATTER | 5 | 5 | 11.2 (1.6) | 17.8 (1.9) | 5.0 (0.0) | 5.2 (0.2) | 1.8 (0.1) | 12.2 (0.7) | 11.6 (1.4) | 26.5 (2.1) | 24.2 (3.2) |
| BUCKET | 5 | 5 | 0.9 (0.0) | 3.6 (0.2) | 3.0 (0.0) | 2.0 (0.0) | 1.7 (0.1) | 10.0 (0.0) | 1.6 (0.2) | 19.4 (1.1) | 19.3 (5.7) |
| COMPASS | 5 | 5 | 15.5 (4.2) | 18.6 (1.6) | 5.0 (0.0) | 5.2 (0.2) | 1.8 (0.2) | 11.8 (0.6) | 12.8 (1.4) | 28.7 (1.9) | 20.3 (2.8) |
| LEATHER | 5 | 5 | 0.9 (0.0) | 3.8 (0.2) | 3.0 (0.0) | 2.0 (0.0) | 1.8 (0.1) | 10.0 (0.0) | 1.8 (0.2) | 16.7 (1.7) | 17.9 (4.4) |
| PAPER | 5 | 5 | 0.9 (0.0) | 3.4 (0.2) | 3.0 (0.0) | 2.0 (0.0) | 1.6 (0.1) | 10.0 (0.0) | 1.4 (0.2) | 19.8 (2.0) | 40.6 (27.0) |
| QUILL | 5 | 5 | 18.2 (3.5) | 19.8 (1.2) | 5.0 (0.0) | 5.2 (0.2) | 2.1 (0.1) | 13.2 (0.4) | 12.6 (1.1) | 29.6 (2.5) | 24.4 (3.2) |
| SUGAR | 5 | 5 | 0.8 (0.0) | 3.2 (0.2) | 3.0 (0.0) | 2.0 (0.0) | 1.7 (0.2) | 10.0 (0.0) | 1.2 (0.2) | 17.7 (1.6) | 17.5 (3.2) |
| BOOK | 5 | 5 | 45.8 (4.5) | 19.6 (0.9) | 5.0 (0.0) | 5.6 (0.2) | 6.0 (0.2) | 11.2 (1.0) | 14.4 (0.9) | 21.6 (1.8) | 21.0 (1.7) |
| MAP | 5 | 5 | 64.1 (10.6) | 22.0 (2.6) | 5.0 (0.0) | 5.2 (0.2) | 6.1 (0.2) | 10.8 (0.4) | 17.2 (2.7) | 22.5 (1.6) | 23.0 (1.2) |
| MILKBUCKET | 5 | 5 | 1.2 (0.1) | 4.4 (0.4) | 3.0 (0.0) | 2.0 (0.0) | 6.8 (0.3) | 10.0 (0.0) | 2.4 (0.4) | 12.1 (0.7) | 15.3 (1.6) |
| BOOKQUILL | 5 | 5 | 4.5 (0.8) | 10.2 (1.4) | 4.0 (0.0) | 4.0 (0.0) | 3.9 (0.1) | 10.0 (0.0) | 7.2 (1.4) | 26.1 (0.8) | 22.4 (0.9) |
| MILKB.SUGAR | 5 | 5 | 3.5 (0.5) | 9.6 (1.3) | 4.0 (0.0) | 4.0 (0.0) | 3.9 (0.1) | 10.2 (0.2) | 6.4 (1.2) | 17.4 (0.5) | 12.5 (0.8) |
| CAKE | 5 | 5 | 9.1 (0.9) | 17.0 (0.9) | 4.0 (0.0) | 3.2 (0.2) | 2.1 (0.1) | 10.0 (0.0) | 14.0 (0.9) | 37.5 (1.9) | 18.0 (1.9) |

| Completed Runs | 5 |
|---|---|
| Total Time (s.) | 176.6 (13.1) |
| Total Calls | 153.0 (3.6) |

(b) OPL

| Task | # G | # L | Time (s.) | Calls | States | Edges | Ep. First HRM ($\times 10^2$) | # Examples G | # Examples D | # Examples I | Example Length G | Example Length D | Example Length I |
|---|---|---|---|---|---|---|---|---|---|---|---|---|---|
| BATTER | 5 | 5 | 13.9 (3.0) | 23.0 (3.0) | 6.0 (0.0) | 9.2 (0.2) | 12.0 (1.0) | 11.4 (0.4) | 7.0 (1.2) | 10.6 (1.6) | 20.4 (1.1) | 18.7 (1.6) | 12.1 (1.7) |
| BUCKET | 5 | 5 | 1.8 (0.1) | 7.2 (0.6) | 4.0 (0.0) | 4.0 (0.0) | 8.0 (0.5) | 10.2 (0.2) | 2.2 (0.2) | 2.8 (0.4) | 10.2 (0.5) | 13.4 (1.9) | 6.8 (1.7) |
| COMPASS | 5 | 5 | 13.2 (1.7) | 22.0 (1.7) | 6.0 (0.0) | 9.2 (0.2) | 10.4 (1.4) | 11.0 (0.0) | 6.8 (1.0) | 10.2 (1.0) | 17.2 (1.6) | 20.9 (1.9) | 14.3 (0.8) |
| LEATHER | 5 | 5 | 1.9 (0.1) | 7.0 (0.5) | 4.0 (0.0) | 4.0 (0.0) | 6.9 (0.5) | 10.0 (0.0) | 2.4 (0.2) | 2.6 (0.4) | 11.1 (0.9) | 16.9 (5.6) | 8.9 (3.3) |
| PAPER | 5 | 5 | 2.0 (0.2) | 7.6 (0.6) | 4.0 (0.0) | 4.0 (0.0) | 7.7 (1.1) | 10.0 (0.0) | 3.0 (0.3) | 2.6 (0.4) | 10.1 (0.9) | 18.9 (3.3) | 5.6 (0.8) |
| QUILL | 5 | 5 | 11.5 (1.3) | 22.0 (1.2) | 6.0 (0.0) | 9.2 (0.2) | 12.8 (1.5) | 10.6 (0.2) | 6.4 (0.7) | 11.0 (0.9) | 15.3 (1.3) | 13.5 (1.0) | 12.1 (1.4) |
| SUGAR | 5 | 5 | 1.6 (0.1) | 6.4 (0.4) | 4.0 (0.0) | 4.0 (0.0) | 6.5 (0.7) | 10.0 (0.0) | 2.4 (0.2) | 2.0 (0.3) | 9.6 (0.6) | 15.3 (3.6) | 16.6 (9.2) |
| BOOK | 5 | 5 | 69.0 (20.5) | 21.8 (2.2) | 6.0 (0.0) | 6.2 (0.2) | 5.5 (0.1) | 10.4 (0.2) | 5.2 (0.9) | 12.2 (1.9) | 20.4 (1.3) | 21.2 (2.0) | 20.8 (1.7) |
| MAP | 5 | 5 | 76.5 (6.0) | 24.2 (1.3) | 6.0 (0.0) | 6.4 (0.2) | 5.7 (0.3) | 11.6 (0.8) | 4.0 (0.3) | 14.6 (1.0) | 24.8 (3.0) | 21.4 (2.1) | 25.7 (0.8) |
| MILKBUCKET | 5 | 5 | 1.7 (0.2) | 6.0 (0.6) | 4.0 (0.0) | 3.0 (0.0) | 7.5 (0.7) | 10.2 (0.2) | 1.4 (0.2) | 2.4 (0.2) | 11.7 (0.7) | 25.4 (6.4) | 14.2 (3.4) |
| BOOKQUILL | 5 | 5 | 5.3 (0.9) | 10.8 (1.4) | 4.0 (0.0) | 4.0 (0.0) | 3.7 (0.1) | 10.0 (0.0) | 1.0 (0.5) | 6.8 (0.9) | 27.7 (1.0) | 11.2 (5.4) | 21.1 (1.7) |
| MILKB.SUGAR | 5 | 5 | 4.0 (0.9) | 9.8 (1.7) | 4.0 (0.0) | 4.0 (0.0) | 3.8 (0.1) | 10.0 (0.0) | 1.6 (0.7) | 5.2 (1.2) | 18.4 (0.7) | 8.3 (2.9) | 15.6 (1.7) |
| CAKE | 5 | 5 | 16.2 (0.4) | 20.8 (0.2) | 5.0 (0.0) | 4.0 (0.0) | 2.1 (0.1) | 10.0 (0.0) | 3.2 (0.2) | 14.6 (0.2) | 38.1 (0.9) | 22.5 (3.3) | 25.8 (1.7) |

| Completed Runs | 5 |
|---|---|
| Total Time (s.) | 218.6 (21.1) |
| Total Calls | 188.6 (5.4) |

(c) FR

| Task | # G | # L | Time (s.) | Calls | States | Edges | Ep. First HRM ($\times 10^2$) | # Examples G | # Examples I | Example Length G | Example Length I |
|---|---|---|---|---|---|---|---|---|---|---|---|
| BATTER | 5 | 5 | 12.6 (1.8) | 17.6 (1.3) | 5.0 (0.0) | 5.4 (0.2) | 9.2 (1.2) | 11.6 (0.4) | 12.0 (1.2) | 30.3 (2.3) | 27.8 (2.0) |
| BUCKET | 5 | 5 | 1.2 (0.1) | 3.8 (0.2) | 3.0 (0.0) | 2.0 (0.0) | 6.7 (0.9) | 10.0 (0.0) | 1.8 (0.2) | 16.6 (2.5) | 28.5 (4.1) |
| COMPASS | 5 | 5 | 14.1 (1.5) | 20.2 (1.7) | 5.0 (0.0) | 5.2 (0.2) | 9.8 (0.7) | 11.6 (0.6) | 14.6 (1.2) | 26.5 (0.8) | 26.5 (2.1) |
| LEATHER | 5 | 5 | 1.1 (0.1) | 3.6 (0.2) | 3.0 (0.0) | 2.0 (0.0) | 4.5 (0.7) | 10.0 (0.0) | 1.6 (0.2) | 13.4 (1.3) | 16.7 (3.6) |
| PAPER | 5 | 5 | 1.2 (0.1) | 4.0 (0.0) | 3.0 (0.0) | 2.0 (0.0) | 4.9 (0.9) | 10.0 (0.0) | 2.0 (0.0) | 12.4 (1.1) | 10.9 (2.5) |
| QUILL | 5 | 5 | 9.3 (0.8) | 16.0 (0.8) | 5.0 (0.0) | 5.2 (0.2) | 9.4 (1.7) | 10.6 (0.2) | 11.4 (0.6) | 25.4 (0.3) | 25.5 (2.7) |
| SUGAR | 5 | 5 | 1.4 (0.2) | 3.8 (0.2) | 3.0 (0.0) | 2.0 (0.0) | 5.2 (0.3) | 10.0 (0.0) | 1.8 (0.2) | 15.3 (1.7) | 21.0 (10.1) |
| BOOK | 5 | 5 | 43.8 (13.0) | 20.0 (1.9) | 5.0 (0.0) | 5.4 (0.2) | 6.0 (0.1) | 10.0 (0.0) | 16.0 (1.9) | 21.9 (1.0) | 14.7 (1.4) |
| MAP | 5 | 5 | 85.2 (13.4) | 22.2 (2.5) | 5.0 (0.0) | 5.2 (0.2) | 5.9 (0.1) | 10.2 (0.2) | 18.0 (2.6) | 26.5 (0.9) | 18.2 (1.2) |
| MILKBUCKET | 5 | 5 | 1.4 (0.1) | 4.4 (0.2) | 3.0 (0.0) | 2.0 (0.0) | 10.2 (0.9) | 10.0 (0.0) | 2.4 (0.2) | 13.0 (0.8) | 12.2 (2.8) |
| BOOKQUILL | 5 | 5 | 6.3 (0.9) | 13.2 (1.7) | 4.0 (0.0) | 4.0 (0.0) | 3.8 (0.1) | 10.0 (0.0) | 10.2 (1.7) | 30.6 (2.0) | 11.9 (1.2) |
| MILKB.SUGAR | 5 | 5 | 4.8 (0.6) | 11.8 (1.3) | 4.0 (0.0) | 4.0 (0.0) | 3.8 (0.1) | 10.0 (0.0) | 8.8 (1.3) | 19.8 (0.7) | 8.6 (1.0) |
| CAKE | 5 | 5 | 12.5 (1.8) | 20.8 (2.5) | 4.0 (0.0) | 3.0 (0.0) | 2.3 (0.1) | 10.0 (0.0) | 17.8 (2.5) | 44.2 (2.6) | 13.1 (1.2) |

| Completed Runs | 5 |
|---|---|
| Total Time (s.) | 194.9 (17.6) |
| Total Calls | 161.4 (7.0) |

(d) FRL

| Task | # G | # L | Time (s.) | Calls | States | Edges | Ep. First HRM ($\times 10^2$) | # Examples G | # Examples D | # Examples I | Example Length G | Example Length D | Example Length I |
|---|---|---|---|---|---|---|---|---|---|---|---|---|---|
| BATTER | 5 | 5 | 11.2 (1.4) | 23.4 (2.5) | 6.0 (0.0) | 9.2 (0.2) | 468.4 (121.9) | 10.4 (0.2) | 7.6 (0.9) | 11.4 (1.9) | 11.9 (0.6) | 10.1 (1.3) | 9.9 (0.4) |
| BUCKET | 5 | 5 | 2.4 (0.1) | 7.0 (0.3) | 4.0 (0.0) | 4.0 (0.0) | 129.5 (69.4) | 10.2 (0.2) | 2.8 (0.2) | 2.0 (0.3) | 7.8 (0.5) | 9.9 (1.7) | 6.4 (2.1) |
| COMPASS | 5 | 5 | 13.1 (1.9) | 24.6 (2.2) | 6.0 (0.0) | 9.4 (0.2) | 550.8 (156.4) | 10.4 (0.2) | 7.8 (1.0) | 12.4 (1.2) | 12.5 (1.6) | 9.4 (1.0) | 8.4 (0.5) |
| LEATHER | 5 | 5 | 2.5 (0.4) | 7.8 (0.7) | 4.0 (0.0) | 4.0 (0.0) | 89.0 (18.0) | 10.0 (0.0) | 3.2 (0.4) | 2.6 (0.4) | 7.3 (0.4) | 9.3 (1.7) | 3.7 (0.4) |
| PAPER | 5 | 5 | 2.1 (0.1) | 7.0 (0.3) | 4.0 (0.0) | 4.0 (0.0) | 82.7 (18.8) | 10.0 (0.0) | 3.0 (0.3) | 2.0 (0.3) | 6.9 (0.7) | 10.2 (1.8) | 4.7 (2.7) |
| QUILL | 5 | 5 | 11.6 (1.2) | 23.8 (1.5) | 6.0 (0.0) | 9.6 (0.2) | 458.9 (61.0) | 10.6 (0.2) | 8.0 (0.9) | 11.2 (1.2) | 11.9 (0.6) | 13.1 (2.7) | 9.2 (0.8) |
| SUGAR | 5 | 5 | 2.6 (0.2) | 8.4 (0.7) | 4.0 (0.0) | 4.0 (0.0) | 103.5 (39.5) | 10.0 (0.0) | 3.6 (0.4) | 2.8 (0.5) | 8.2 (0.7) | 10.1 (1.9) | 5.0 (1.1) |
| BOOK | 5 | 5 | 62.2 (13.2) | 27.4 (2.2) | 6.0 (0.0) | 6.6 (0.2) | 5.3 (0.1) | 10.2 (0.2) | 5.6 (0.6) | 17.6 (1.7) | 23.0 (1.0) | 16.5 (2.0) | 13.4 (1.0) |
| MAP | 5 | 5 | 131.3 (28.0) | 34.0 (3.0) | 6.0 (0.0) | 6.6 (0.2) | 5.5 (0.0) | 10.2 (0.2) | 6.8 (0.7) | 23.0 (2.4) | 26.2 (0.7) | 16.9 (1.6) | 14.5 (0.5) |
| MILKBUCKET | 5 | 5 | 2.7 (0.7) | 6.6 (0.6) | 4.0 (0.0) | 3.0 (0.0) | 6.8 (0.3) | 10.0 (0.0) | 2.2 (0.2) | 2.4 (0.4) | 12.0 (0.8) | 9.4 (1.0) | 9.9 (2.3) |
| BOOKQUILL | 5 | 5 | 6.8 (0.6) | 12.6 (0.7) | 4.0 (0.0) | 4.0 (0.0) | 4.4 (0.2) | 10.0 (0.0) | 1.6 (0.5) | 8.0 (0.3) | 32.3 (2.3) | 9.7 (4.5) | 11.6 (1.1) |
| MILKB.SUGAR | 5 | 5 | 5.4 (0.6) | 12.2 (1.0) | 4.0 (0.0) | 4.0 (0.0) | 4.0 (0.1) | 10.2 (0.2) | 1.0 (0.4) | 8.0 (0.4) | 20.4 (1.7) | 2.9 (1.4) | 10.3 (1.2) |
| CAKE | 5 | 5 | 16.3 (1.2) | 21.2 (1.0) | 5.0 (0.0) | 4.0 (0.0) | 2.8 (0.0) | 10.0 (0.0) | 2.6 (0.2) | 15.6 (1.2) | 47.7 (1.9) | 15.0 (0.7) | 16.0 (0.9) |

| Completed Runs | 5 |
|---|---|
| Total Time (s.) | 270.1 (34.6) |
| Total Calls | 216.0 (5.1) |

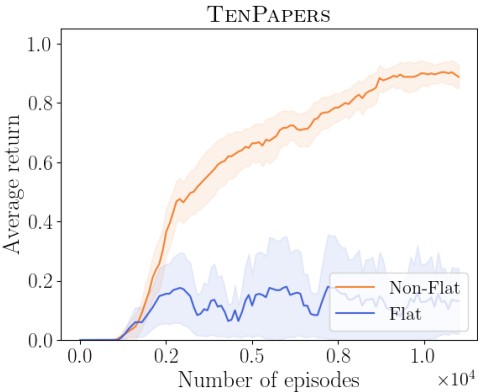

Figure 12: Learning curves comparing the performance of handcrafted non-flat and flat HRMs in the TENPAPERS task (FRL setting).

Table 5: Results of LHRM in CRAFTWORLD without exploration using options.

(a) OP

| Task | # G | # L | Time (s.) | Calls | States | Edges | Ep. First HRM ($\times 10^2$) | # Examples G | # Examples I | Example Length G | Example Length I |
|---|---|---|---|---|---|---|---|---|---|---|---|
| BATTER | 5 | 5 | 11.2 (1.7) | 17.8 (1.9) | 5.0 (0.0) | 5.2 (0.2) | 1.8 (0.1) | 12.2 (0.7) | 11.6 (1.4) | 26.5 (2.1) | 24.2 (3.2) |
| BUCKET | 5 | 5 | 0.9 (0.0) | 3.6 (0.2) | 3.0 (0.0) | 2.0 (0.0) | 1.7 (0.1) | 10.0 (0.0) | 1.6 (0.2) | 19.4 (1.1) | 19.3 (5.7) |
| COMPASS | 5 | 5 | 15.6 (4.1) | 18.6 (1.6) | 5.0 (0.0) | 5.2 (0.2) | 1.8 (0.2) | 11.8 (0.6) | 12.8 (1.4) | 28.7 (1.9) | 20.3 (2.8) |
| LEATHER | 5 | 5 | 0.9 (0.1) | 3.8 (0.2) | 3.0 (0.0) | 2.0 (0.0) | 1.8 (0.1) | 10.0 (0.0) | 1.8 (0.2) | 16.7 (1.7) | 17.9 (4.4) |
| PAPER | 5 | 5 | 0.9 (0.1) | 3.4 (0.2) | 3.0 (0.0) | 2.0 (0.0) | 1.6 (0.1) | 10.0 (0.0) | 1.4 (0.2) | 19.8 (2.0) | 40.6 (27.0) |
| QUILL | 5 | 5 | 18.3 (3.6) | 19.8 (1.2) | 5.0 (0.0) | 5.2 (0.2) | 2.1 (0.1) | 13.2 (0.4) | 12.6 (1.1) | 29.6 (2.5) | 24.4 (3.2) |
| SUGAR | 5 | 5 | 0.9 (0.0) | 3.2 (0.2) | 3.0 (0.0) | 2.0 (0.0) | 1.7 (0.2) | 10.0 (0.0) | 1.2 (0.2) | 17.7 (1.6) | 17.5 (3.2) |
| BOOK | 5 | 5 | 529.0 (164.2) | 21.2 (1.4) | 5.0 (0.0) | 5.8 (0.2) | 6.8 (0.2) | 10.2 (0.2) | 17.0 (1.5) | 33.0 (2.6) | 23.7 (1.3) |
| MAP | 5 | 5 | 1924.2 (443.5) | 28.0 (3.8) | 5.0 (0.0) | 5.4 (0.2) | 7.8 (0.4) | 10.4 (0.2) | 23.6 (3.7) | 40.1 (1.0) | 29.4 (1.3) |
| MILKBUCKET | 5 | 5 | 1.6 (0.2) | 4.4 (0.4) | 3.0 (0.0) | 2.0 (0.0) | 6.1 (0.3) | 10.0 (0.0) | 2.4 (0.4) | 16.0 (1.0) | 14.2 (1.3) |
| BOOKQUILL | 5 | 5 | 42.7 (10.1) | 24.6 (3.9) | 4.0 (0.0) | 4.0 (0.0) | 6.8 (0.2) | 10.0 (0.0) | 21.6 (3.9) | 55.8 (2.7) | 21.2 (1.1) |
| MILKB.SUGAR | 5 | 5 | 8.1 (0.8) | 11.8 (1.0) | 4.0 (0.0) | 4.0 (0.0) | 4.9 (0.1) | 10.2 (0.2) | 8.6 (1.2) | 31.1 (0.7) | 13.1 (0.8) |
| CAKE | 5 | 5 | 198.3 (47.5) | 43.0 (5.3) | 4.0 (0.0) | 3.8 (0.2) | 5.5 (0.2) | 10.0 (0.0) | 40.0 (5.3) | 65.0 (0.9) | 22.0 (0.9) |

| | |
|---|---|
| Completed Runs | 5 |
| Total Time (s.) | 2752.8 (503.2) |
| Total Calls | 203.2 (11.8) |

(b) OPL

| Task | # G | # L | Time (s.) | Calls | States | Edges | Ep. First HRM ($\times 10^2$) | # Examples G | # Examples D | # Examples I | Example Length G | Example Length D | Example Length I |
|---|---|---|---|---|---|---|---|---|---|---|---|---|---|
| BATTER | 5 | 5 | 14.1 (3.2) | 23.0 (3.0) | 6.0 (0.0) | 9.2 (0.2) | 12.0 (1.0) | 11.4 (0.4) | 7.0 (1.2) | 10.6 (1.6) | 20.4 (1.1) | 18.7 (1.6) | 12.1 (1.7) |
| BUCKET | 5 | 5 | 1.8 (0.1) | 7.2 (0.6) | 4.0 (0.0) | 4.0 (0.0) | 8.0 (0.5) | 10.2 (0.2) | 2.2 (0.2) | 2.8 (0.4) | 10.2 (0.5) | 13.4 (1.9) | 6.8 (1.7) |
| COMPASS | 5 | 5 | 13.5 (1.8) | 22.0 (1.7) | 6.0 (0.0) | 9.2 (0.2) | 10.4 (1.4) | 11.0 (0.6) | 6.8 (1.0) | 10.2 (1.0) | 17.2 (1.6) | 20.9 (1.9) | 14.3 (0.8) |
| LEATHER | 5 | 5 | 1.8 (0.1) | 7.0 (0.5) | 4.0 (0.0) | 4.0 (0.0) | 6.9 (0.5) | 10.0 (0.0) | 2.4 (0.2) | 2.6 (0.4) | 11.1 (0.9) | 16.9 (5.6) | 8.9 (3.3) |
| PAPER | 5 | 5 | 2.0 (0.2) | 7.6 (0.6) | 4.0 (0.0) | 4.0 (0.0) | 7.7 (1.1) | 10.0 (0.0) | 3.0 (0.3) | 2.6 (0.4) | 10.1 (0.9) | 18.9 (3.3) | 5.6 (0.8) |
| QUILL | 5 | 5 | 11.8 (1.3) | 22.0 (1.2) | 6.0 (0.0) | 9.2 (0.2) | 12.8 (1.5) | 10.6 (0.2) | 6.4 (0.7) | 11.0 (0.9) | 15.3 (1.3) | 13.5 (1.0) | 12.1 (1.4) |
| SUGAR | 5 | 5 | 1.6 (0.1) | 6.4 (0.4) | 4.0 (0.0) | 4.0 (0.0) | 6.5 (0.7) | 10.0 (0.0) | 2.4 (0.2) | 2.0 (0.3) | 9.6 (0.6) | 15.3 (3.6) | 16.6 (9.2) |
| BOOK | 5 | 5 | 224.8 (71.6) | 27.0 (1.9) | 6.0 (0.0) | 6.4 (0.2) | 139.7 (21.8) | 11.6 (0.4) | 3.2 (0.4) | 18.2 (1.4) | 22.0 (1.6) | 24.7 (6.5) | 23.5 (1.2) |
| MAP | 5 | 5 | 339.9 (33.6) | 33.0 (2.8) | 6.0 (0.0) | 6.4 (0.2) | 204.8 (27.1) | 10.6 (0.2) | 2.8 (0.5) | 25.6 (2.5) | 25.4 (0.8) | 21.8 (3.1) | 25.2 (1.1) |
| MILKBUCKET | 5 | 5 | 3.5 (0.3) | 8.2 (0.6) | 4.0 (0.0) | 3.0 (0.0) | 47.6 (3.7) | 10.2 (0.2) | 2.6 (0.4) | 3.4 (0.4) | 10.3 (0.7) | 16.2 (1.7) | 14.2 (1.8) |
| BOOKQUILL | 5 | 5 | 19.0 (2.2) | 15.4 (1.5) | 4.0 (0.0) | 4.0 (0.0) | 383.4 (83.7) | 10.0 (0.0) | 1.0 (0.3) | 11.4 (1.5) | 38.2 (1.6) | 14.1 (4.9) | 23.8 (1.0) |
| MILKB.SUGAR | 5 | 5 | 11.4 (2.1) | 14.4 (1.7) | 4.0 (0.0) | 4.0 (0.0) | 87.4 (8.9) | 10.4 (0.2) | 1.0 (0.4) | 10.0 (1.3) | 19.7 (1.2) | 8.7 (4.9) | 17.6 (1.2) |
| CAKE | 4 | 1 | 277.4 (0.0) | 33.0 (0.0) | 5.0 (0.0) | 4.0 (0.0) | 264.1 (0.0) | 10.0 (0.0) | 2.0 (0.0) | 28.0 (0.0) | 46.7 (0.0) | 36.0 (0.0) | 22.9 (0.0) |

| | |
|---|---|
| Completed Runs | 5 |
| Total Time (s.) | 701.0 (111.2) |
| Total Calls | 199.8 (6.9) |

(c) FRL

| Task | # G | # L | Time (s.) | Calls | States | Edges | Ep. First HRM ($\times 10^2$) | # Examples G | # Examples D | # Examples I | Example Length G | Example Length D | Example Length I |
|---|---|---|---|---|---|---|---|---|---|---|---|---|---|
| BATTER | 5 | 5 | 11.1 (1.4) | 23.4 (2.5) | 6.0 (0.0) | 9.2 (0.2) | 468.4 (121.9) | 10.4 (0.2) | 7.6 (0.9) | 11.4 (1.9) | 11.9 (0.6) | 10.1 (1.3) | 9.9 (0.4) |
| BUCKET | 5 | 5 | 2.2 (0.1) | 7.0 (0.3) | 4.0 (0.0) | 4.0 (0.0) | 129.5 (69.4) | 10.2 (0.2) | 2.8 (0.2) | 2.0 (0.3) | 7.8 (0.5) | 9.9 (1.7) | 6.4 (2.1) |
| COMPASS | 5 | 5 | 12.9 (1.9) | 24.6 (2.2) | 6.0 (0.0) | 9.4 (0.2) | 550.8 (156.4) | 10.4 (0.2) | 7.8 (1.0) | 12.4 (1.2) | 12.5 (1.6) | 9.4 (1.0) | 8.4 (0.5) |
| LEATHER | 5 | 5 | 2.8 (0.4) | 7.8 (0.7) | 4.0 (0.0) | 4.0 (0.0) | 89.0 (18.0) | 10.0 (0.0) | 3.2 (0.4) | 2.6 (0.4) | 7.3 (0.4) | 9.3 (1.7) | 3.7 (0.4) |
| PAPER | 5 | 5 | 2.1 (0.1) | 7.0 (0.3) | 4.0 (0.0) | 4.0 (0.0) | 82.7 (18.8) | 10.0 (0.0) | 3.0 (0.0) | 2.0 (0.3) | 6.9 (0.7) | 10.2 (1.8) | 4.7 (2.7) |
| QUILL | 5 | 5 | 11.6 (1.1) | 23.8 (1.5) | 6.0 (0.0) | 9.6 (0.2) | 458.9 (61.0) | 10.6 (0.2) | 8.0 (0.9) | 11.2 (1.2) | 11.9 (0.6) | 13.1 (2.7) | 9.2 (0.8) |
| SUGAR | 5 | 5 | 2.6 (0.3) | 8.4 (0.7) | 4.0 (0.0) | 4.0 (0.0) | 103.5 (39.5) | 10.0 (0.0) | 3.6 (0.4) | 2.8 (0.5) | 8.2 (0.7) | 10.1 (1.9) | 5.0 (1.1) |
| BOOK | 5 | 0 | 0.0 (0.0) | 0.0 (0.0) | 0.0 (0.0) | 0.0 (0.0) | 0.0 (0.0) | 0.0 (0.0) | 0.0 (0.0) | 0.0 (0.0) | 0.0 (0.0) | 0.0 (0.0) | 0.0 (0.0) |
| MAP | 3 | 0 | 0.0 (0.0) | 0.0 (0.0) | 0.0 (0.0) | 0.0 (0.0) | 0.0 (0.0) | 0.0 (0.0) | 0.0 (0.0) | 0.0 (0.0) | 0.0 (0.0) | 0.0 (0.0) | 0.0 (0.0) |
| MILKBUCKET | 5 | 2 | 4.7 (0.5) | 11.0 (1.0) | 4.0 (0.0) | 3.0 (0.0) | 885.6 (142.3) | 10.0 (0.0) | 2.0 (0.0) | 7.0 (1.0) | 8.2 (0.4) | 10.2 (1.7) | 8.8 (0.2) |
| BOOKQUILL | 0 | 0 | 0.0 (0.0) | 0.0 (0.0) | 0.0 (0.0) | 0.0 (0.0) | 0.0 (0.0) | 0.0 (0.0) | 0.0 (0.0) | 0.0 (0.0) | 0.0 (0.0) | 0.0 (0.0) | 0.0 (0.0) |
| MILKB.SUGAR | 0 | 0 | 0.0 (0.0) | 0.0 (0.0) | 0.0 (0.0) | 0.0 (0.0) | 0.0 (0.0) | 0.0 (0.0) | 0.0 (0.0) | 0.0 (0.0) | 0.0 (0.0) | 0.0 (0.0) | 0.0 (0.0) |
| CAKE | 0 | 0 | 0.0 (0.0) | 0.0 (0.0) | 0.0 (0.0) | 0.0 (0.0) | 0.0 (0.0) | 0.0 (0.0) | 0.0 (0.0) | 0.0 (0.0) | 0.0 (0.0) | 0.0 (0.0) | 0.0 (0.0) |

| | |
|---|---|
| Completed Runs | 5 |
| Total Time (s.) | 47.1 (0.9) |
| Total Calls | 106.4 (2.9) |

Table 6: Results of LHRM in WATERWORLD for the default case.

(a) WOD

| Task | # G | # L | Time (s.) | Calls | States | Edges | Ep. First HRM $(\times 10^2)$ | # Examples | | Example Length | |
|---|---|---|---|---|---|---|---|---|---|---|---|
| | | | | | | | | G | I | G | I |
| RG | 5 | 5 | 0.9 (0.0) | 4.0 (0.0) | 3.0 (0.0) | 2.0 (0.0) | 0.9 (0.1) | 10.0 (0.0) | 2.0 (0.0) | 11.2 (1.0) | 5.8 (1.1) |
| BC | 5 | 5 | 0.9 (0.1) | 3.8 (0.2) | 3.0 (0.0) | 2.0 (0.0) | 0.8 (0.1) | 10.0 (0.0) | 1.8 (0.2) | 10.8 (0.8) | 11.9 (3.4) |
| MY | 5 | 5 | 0.9 (0.0) | 3.6 (0.2) | 3.0 (0.0) | 2.0 (0.0) | 0.7 (0.0) | 10.0 (0.0) | 1.6 (0.2) | 8.7 (0.8) | 6.6 (1.9) |
| RG&BC | 5 | 5 | 4.5 (0.3) | 13.4 (0.4) | 4.0 (0.0) | 4.0 (0.0) | 8.8 (0.3) | 11.8 (0.6) | 8.6 (0.7) | 12.2 (0.9) | 14.8 (1.2) |
| BC&MY | 5 | 5 | 5.8 (1.0) | 15.6 (2.1) | 4.0 (0.0) | 4.0 (0.0) | 8.1 (0.2) | 12.8 (1.3) | 9.8 (1.5) | 13.2 (1.7) | 17.1 (1.6) |
| RG&MY | 5 | 5 | 4.7 (0.5) | 13.2 (1.0) | 4.0 (0.0) | 4.0 (0.0) | 8.5 (0.2) | 10.8 (0.2) | 9.4 (0.9) | 12.2 (0.7) | 18.6 (1.2) |
| RGB | 5 | 5 | 1.2 (0.1) | 4.8 (0.5) | 3.0 (0.0) | 2.0 (0.0) | 8.6 (0.2) | 10.0 (0.0) | 2.8 (0.5) | 7.8 (0.2) | 7.0 (1.4) |
| CMY | 5 | 5 | 1.4 (0.2) | 5.4 (0.7) | 3.0 (0.0) | 2.0 (0.0) | 8.8 (0.5) | 10.0 (0.0) | 3.4 (0.7) | 8.0 (0.3) | 10.2 (1.3) |
| RGB&CMY | 5 | 5 | 15.1 (1.7) | 21.6 (1.7) | 4.0 (0.0) | 4.0 (0.0) | 2.3 (0.0) | 11.0 (0.4) | 17.6 (1.7) | 17.3 (0.4) | 22.6 (1.6) |

Completed Runs 5
Total Time (s.) 35.4 (2.0)
Total Calls 85.4 (3.1)

(b) WD

| Task | # G | # L | Time (s.) | Calls | States | Edges | Ep. First HRM $(\times 10^2)$ | # Examples | | | Example Length | | |
|---|---|---|---|---|---|---|---|---|---|---|---|---|---|
| | | | | | | | | G | D | I | G | D | I |
| RG | 5 | 5 | 1.9 (0.2) | 7.8 (1.1) | 4.0 (0.0) | 4.0 (0.0) | 3.0 (0.3) | 10.0 (0.0) | 3.0 (0.3) | 2.8 (0.9) | 7.0 (0.7) | 10.3 (1.6) | 4.2 (0.8) |
| BC | 5 | 5 | 1.8 (0.3) | 7.2 (1.0) | 4.0 (0.0) | 4.0 (0.0) | 2.7 (0.3) | 10.2 (0.2) | 2.4 (0.2) | 2.6 (0.7) | 8.4 (0.5) | 6.9 (1.5) | 6.3 (1.7) |
| MY | 5 | 5 | 1.4 (0.1) | 5.8 (0.4) | 4.0 (0.0) | 4.0 (0.0) | 2.9 (0.3) | 10.0 (0.0) | 2.4 (0.2) | 1.4 (0.2) | 6.9 (0.4) | 5.8 (1.5) | 4.6 (1.6) |
| RG&BC | 5 | 5 | 11.7 (2.5) | 24.0 (3.8) | 4.8 (0.2) | 4.8 (0.2) | 12.0 (0.4) | 13.0 (0.7) | 6.2 (1.6) | 11.8 (1.9) | 11.7 (0.8) | 6.9 (1.5) | 11.8 (0.8) |
| BC&MY | 5 | 5 | 9.5 (1.5) | 20.8 (2.4) | 4.8 (0.2) | 4.8 (0.2) | 11.5 (0.4) | 11.2 (0.6) | 5.2 (0.7) | 11.4 (1.6) | 10.6 (0.9) | 8.8 (1.4) | 13.2 (0.9) |
| RG&MY | 5 | 5 | 5.4 (0.5) | 14.0 (1.3) | 4.2 (0.2) | 4.2 (0.2) | 11.8 (0.6) | 10.6 (0.2) | 3.2 (0.7) | 7.2 (1.1) | 9.8 (0.3) | 6.2 (1.9) | 11.6 (0.7) |
| RGB | 5 | 5 | 2.5 (0.3) | 8.2 (0.7) | 4.0 (0.0) | 3.0 (0.0) | 11.9 (0.6) | 10.2 (0.2) | 3.0 (0.3) | 3.0 (0.5) | 7.9 (0.4) | 14.0 (1.8) | 10.6 (2.0) |
| CMY | 5 | 5 | 3.6 (0.4) | 11.4 (1.2) | 4.0 (0.0) | 3.0 (0.0) | 10.8 (0.3) | 10.0 (0.0) | 4.2 (0.7) | 5.2 (0.6) | 7.9 (0.2) | 8.8 (1.6) | 10.5 (1.1) |
| RGB&CMY | 5 | 5 | 29.0 (4.1) | 31.4 (2.8) | 4.4 (0.2) | 4.4 (0.2) | 4.9 (0.3) | 11.2 (0.4) | 5.4 (1.6) | 21.8 (1.3) | 16.6 (0.8) | 7.4 (0.9) | 17.2 (0.6) |

Completed Runs 5
Total Time (s.) 67.0 (6.2)
Total Calls 130.6 (6.0)

Table 7: Results of LHRM in WATERWORLD with a restricted set of callable RMs.

(a) WOD

| Task | # G | # L | Time (s.) | Calls | States | Edges | Ep. First HRM $(\times 10^2)$ | # Examples | | Example Length | |
|---|---|---|---|---|---|---|---|---|---|---|---|
| | | | | | | | | G | I | G | I |
| RG | 5 | 5 | 0.9 (0.0) | 4.0 (0.0) | 3.0 (0.0) | 2.0 (0.0) | 0.9 (0.1) | 10.0 (0.0) | 2.0 (0.0) | 11.2 (1.0) | 5.8 (1.1) |
| BC | 5 | 5 | 0.9 (0.1) | 3.8 (0.2) | 3.0 (0.0) | 2.0 (0.0) | 0.8 (0.1) | 10.0 (0.0) | 1.8 (0.2) | 10.8 (0.8) | 11.9 (3.4) |
| MY | 5 | 5 | 0.9 (0.0) | 3.6 (0.2) | 3.0 (0.0) | 2.0 (0.0) | 0.7 (0.0) | 10.0 (0.0) | 1.6 (0.2) | 8.7 (0.8) | 6.6 (1.9) |
| RG&BC | 5 | 5 | 5.3 (0.4) | 15.2 (0.9) | 4.0 (0.0) | 4.0 (0.0) | 8.6 (0.3) | 12.4 (0.2) | 9.8 (0.8) | 14.7 (1.3) | 16.0 (0.8) |
| BC&MY | 5 | 5 | 3.9 (0.1) | 12.4 (0.2) | 4.0 (0.0) | 4.0 (0.0) | 8.3 (0.4) | 11.8 (0.7) | 7.6 (0.7) | 11.2 (0.8) | 13.2 (1.0) |
| RG&MY | 5 | 5 | 4.6 (0.3) | 13.8 (0.9) | 4.0 (0.0) | 4.0 (0.0) | 8.5 (0.2) | 10.2 (0.2) | 10.6 (0.9) | 10.7 (0.5) | 15.8 (1.6) |
| RGB | 5 | 5 | 1.2 (0.1) | 4.8 (0.7) | 3.0 (0.0) | 2.0 (0.0) | 8.7 (0.2) | 10.2 (0.2) | 2.6 (0.7) | 8.3 (0.5) | 16.2 (3.8) |
| CMY | 5 | 5 | 1.6 (0.2) | 6.2 (0.7) | 3.0 (0.0) | 2.0 (0.0) | 8.6 (0.5) | 10.0 (0.0) | 4.2 (0.7) | 8.0 (0.3) | 10.8 (1.2) |
| RGB&CMY | 5 | 5 | 5.7 (0.8) | 15.0 (1.6) | 4.0 (0.0) | 4.0 (0.0) | 2.6 (0.1) | 10.4 (0.4) | 11.6 (1.6) | 17.0 (1.1) | 15.9 (1.3) |

Completed Runs 5
Total Time (s.) 24.9 (0.9)
Total Calls 78.8 (2.7)

(b) WD

| Task | # G | # L | Time (s.) | Calls | States | Edges | Ep. First HRM $(\times 10^2)$ | # Examples | | | Example Length | | |
|---|---|---|---|---|---|---|---|---|---|---|---|---|---|
| | | | | | | | | G | D | I | G | D | I |
| RG | 5 | 5 | 1.9 (0.3) | 7.8 (1.1) | 4.0 (0.0) | 4.0 (0.0) | 3.0 (0.3) | 10.0 (0.0) | 3.0 (0.3) | 2.8 (0.9) | 7.0 (0.7) | 10.3 (1.6) | 4.2 (0.8) |
| BC | 5 | 5 | 1.8 (0.3) | 7.2 (1.0) | 4.0 (0.0) | 4.0 (0.0) | 2.7 (0.3) | 10.2 (0.2) | 2.4 (0.2) | 2.6 (0.7) | 8.4 (0.5) | 6.9 (1.5) | 6.3 (1.7) |
| MY | 5 | 5 | 1.4 (0.1) | 5.8 (0.4) | 4.0 (0.0) | 4.0 (0.0) | 2.9 (0.3) | 10.0 (0.0) | 2.4 (0.2) | 1.4 (0.2) | 6.9 (0.4) | 5.8 (1.5) | 4.6 (1.6) |
| RG&BC | 5 | 5 | 6.9 (0.7) | 17.6 (1.5) | 4.6 (0.2) | 4.6 (0.2) | 12.0 (0.4) | 10.8 (0.2) | 4.6 (0.5) | 9.2 (1.0) | 10.4 (0.5) | 9.4 (1.8) | 12.6 (0.7) |
| BC&MY | 5 | 5 | 9.3 (1.8) | 21.4 (2.9) | 4.8 (0.2) | 4.8 (0.2) | 11.7 (0.5) | 12.2 (1.0) | 5.8 (1.1) | 10.4 (1.8) | 11.4 (0.7) | 6.9 (1.3) | 12.1 (0.7) |
| RG&MY | 5 | 5 | 7.8 (1.1) | 18.8 (1.9) | 4.8 (0.2) | 4.8 (0.2) | 11.8 (0.5) | 11.0 (0.3) | 4.8 (0.4) | 10.0 (2.0) | 9.8 (0.2) | 8.6 (0.8) | 13.0 (0.8) |
| RGB | 5 | 5 | 2.1 (0.1) | 7.6 (0.2) | 4.0 (0.0) | 3.0 (0.0) | 11.9 (0.5) | 10.0 (0.0) | 2.6 (0.2) | 3.0 (0.0) | 7.6 (0.5) | 11.8 (1.7) | 10.7 (1.7) |
| CMY | 5 | 5 | 2.3 (0.2) | 8.2 (0.8) | 4.0 (0.0) | 3.0 (0.0) | 10.7 (0.2) | 10.0 (0.0) | 2.2 (0.4) | 4.0 (0.5) | 7.8 (0.2) | 9.9 (1.6) | 8.9 (0.5) |
| RGB&CMY | 5 | 5 | 9.6 (1.5) | 20.6 (2.6) | 5.0 (0.0) | 5.0 (0.0) | 5.0 (0.6) | 10.2 (0.2) | 6.4 (0.9) | 11.0 (1.8) | 15.0 (0.5) | 12.3 (0.8) | 14.2 (1.2) |

Completed Runs 5
Total Time (s.) 42.9 (3.7)
Total Calls 115.0 (7.5)

Table 8: Results of LHRM in WATERWORLD without exploration using options.

(a) WOD

| Task | # G | # L | Time (s.) | Calls | States | Edges | Ep. First HRM (×10²) | # Examples G | # Examples I | Example Length G | Example Length I |
|---|---|---|---|---|---|---|---|---|---|---|---|
| RG | 5 | 5 | 0.9 (0.0) | 4.0 (0.0) | 3.0 (0.0) | 2.0 (0.0) | 0.9 (0.1) | 10.0 (0.0) | 2.0 (0.0) | 11.2 (1.0) | 5.8 (1.1) |
| BC | 5 | 5 | 0.9 (0.1) | 3.8 (0.2) | 3.0 (0.0) | 2.0 (0.0) | 0.8 (0.1) | 10.0 (0.0) | 1.8 (0.2) | 10.8 (0.8) | 11.9 (3.4) |
| MY | 5 | 5 | 0.9 (0.0) | 3.6 (0.2) | 3.0 (0.0) | 2.0 (0.0) | 0.7 (0.0) | 10.0 (0.0) | 1.6 (0.2) | 8.7 (0.8) | 6.6 (1.9) |
| RG&BC | 5 | 5 | 4.2 (0.4) | 12.2 (0.9) | 4.0 (0.0) | 4.0 (0.0) | 9.5 (0.3) | 10.6 (0.2) | 8.6 (1.1) | 13.8 (0.2) | 15.3 (1.6) |
| BC&MY | 5 | 5 | 4.3 (0.3) | 11.8 (0.7) | 4.0 (0.0) | 4.0 (0.0) | 9.8 (0.1) | 11.6 (0.2) | 7.2 (0.6) | 15.3 (0.9) | 16.7 (1.7) |
| RG&MY | 5 | 5 | 4.6 (0.3) | 12.6 (0.7) | 4.0 (0.0) | 4.0 (0.0) | 9.5 (0.1) | 11.2 (0.4) | 8.4 (0.7) | 14.2 (0.9) | 14.8 (0.8) |
| RGB | 5 | 5 | 1.2 (0.2) | 4.6 (0.7) | 3.0 (0.0) | 2.0 (0.0) | 9.0 (0.1) | 10.0 (0.0) | 2.6 (0.7) | 9.4 (0.4) | 8.7 (1.7) |
| CMY | 5 | 5 | 1.4 (0.1) | 5.0 (0.5) | 3.0 (0.0) | 2.0 (0.0) | 8.8 (0.2) | 10.0 (0.0) | 3.0 (0.5) | 8.8 (0.2) | 10.6 (1.5) |
| RGB&CMY | 5 | 5 | 16.1 (1.1) | 19.8 (1.1) | 4.0 (0.0) | 4.0 (0.0) | 4.1 (0.1) | 11.2 (0.6) | 15.6 (1.3) | 26.0 (1.2) | 21.6 (0.9) |

| | |
|---|---|
| Completed Runs | 5 |
| Total Time (s.) | 34.4 (1.4) |
| Total Calls | 77.4 (2.0) |

(b) WD

| Task | # G | # L | Time (s.) | Calls | States | Edges | Ep. First HRM (×10²) | # Examples G | # Examples D | # Examples I | Example Length G | Example Length D | Example Length I |
|---|---|---|---|---|---|---|---|---|---|---|---|---|---|
| RG | 5 | 5 | 1.9 (0.3) | 7.8 (1.1) | 4.0 (0.0) | 4.0 (0.0) | 3.0 (0.3) | 10.0 (0.0) | 3.0 (0.3) | 2.8 (0.9) | 7.0 (0.7) | 10.3 (1.6) | 4.2 (0.8) |
| BC | 5 | 5 | 1.8 (0.2) | 7.2 (1.0) | 4.0 (0.0) | 4.0 (0.0) | 2.7 (0.3) | 10.2 (0.2) | 2.4 (0.2) | 2.6 (0.7) | 8.4 (0.5) | 6.9 (1.5) | 6.3 (1.7) |
| MY | 5 | 5 | 1.4 (0.1) | 5.8 (0.4) | 4.0 (0.0) | 4.0 (0.0) | 2.9 (0.3) | 10.0 (0.0) | 2.4 (0.2) | 1.4 (0.2) | 6.9 (0.4) | 5.8 (1.5) | 4.6 (1.6) |
| RG&BC | 5 | 5 | 8.1 (1.4) | 18.2 (2.2) | 4.6 (0.2) | 4.6 (0.2) | 97.4 (4.2) | 10.8 (0.4) | 5.2 (0.6) | 9.2 (1.4) | 10.5 (0.5) | 10.3 (1.7) | 13.7 (1.5) |
| BC&MY | 5 | 5 | 6.2 (0.5) | 15.6 (0.7) | 4.6 (0.2) | 4.6 (0.2) | 91.5 (5.8) | 10.6 (0.2) | 4.6 (0.6) | 7.4 (0.7) | 9.7 (0.2) | 7.0 (1.2) | 11.3 (1.0) |
| RG&MY | 5 | 5 | 8.6 (1.8) | 19.2 (2.7) | 4.4 (0.2) | 4.4 (0.2) | 90.3 (5.3) | 11.2 (0.8) | 5.6 (0.9) | 9.4 (1.3) | 10.4 (0.7) | 7.7 (0.7) | 13.3 (0.9) |
| RGB | 5 | 5 | 2.3 (0.1) | 7.6 (0.2) | 4.0 (0.0) | 3.0 (0.0) | 65.3 (1.6) | 10.2 (0.2) | 2.6 (0.4) | 2.8 (0.4) | 7.6 (0.3) | 11.1 (3.0) | 8.8 (1.7) |
| CMY | 5 | 5 | 4.4 (0.6) | 13.2 (1.5) | 3.8 (0.2) | 3.0 (0.0) | 59.2 (2.9) | 10.2 (0.2) | 3.8 (0.6) | 7.2 (1.0) | 6.9 (0.5) | 8.2 (1.0) | 7.6 (0.8) |
| RGB&CMY | 5 | 5 | 32.1 (5.0) | 31.4 (3.3) | 4.4 (0.2) | 4.4 (0.2) | 125.7 (9.9) | 11.4 (0.5) | 5.8 (1.2) | 21.2 (2.2) | 17.1 (0.6) | 8.4 (1.6) | 17.8 (1.0) |

| | |
|---|---|
| Completed Runs | 5 |
| Total Time (s.) | 66.7 (6.6) |
| Total Calls | 126.0 (6.3) |

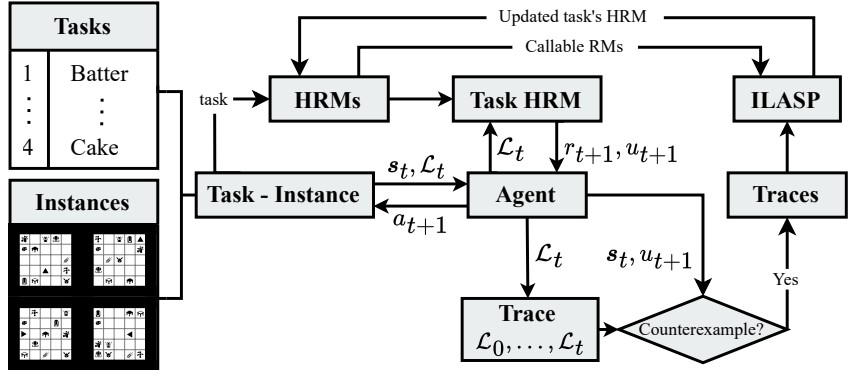

Figure 13: Overview of LHRM.

# H  ILLUSTRATION OF LHRM

Figure 13 illustrates the main components of our algorithm for learning HRMs, LHRM, described in Section 5. Given a set of tasks with known levels and a set of instances, we select a (task, instance) pair at the beginning of an episode. The HRM corresponding to the selected task is taken from the bank of HRMs. At each step, the agent performs an action $a_{t+1}$ in the (task, instance) environment, and observes a tuple $s_t$ and a label $\mathcal{L}_t$. The label is used to (i) know the next hierarchy state $u_{t+1}$ in the HRM and the reward $r_{t+1}$, and (ii) update the label trace $\langle \mathcal{L}_0, \ldots, \mathcal{L}_t \rangle$. If the trace is a counterexample (i.e., the tuple $s_t$ and the new hierarchy state $u_{t+1}$ are inconsistent with each other),[8] we add it to our set of counterexample traces for the current task and learn a new HRM using the ILASP system. The learned HRM is then updated in the bank of HRMs so that future tasks can reuse it. Note that the description omits some aspects for simplicity, such as how are the (task, instance) pairs selected, the exploration using options from lower level HRMs, or the accumulation of traces before learning the first HRM for a given task.

---

[8]For example if $(s_t^T, s_t^G) = (\top, \top)$ and the state in $u_{t+1}$ is not the accepting state of the root RM.

Table 9: Results of learning non-flat and flat HRMs using different methods. The columns are the following: the number of completed runs without timing out, the amount of time needed to learn the HRMs or RMs, and the number of states and edges of the RM.

| Task | LHRM (Non-Flat) | | | | LHRM (Flat) | | | | DeepSynth | | | | LRM | | | | JIRP | | | |
|---|---|---|---|---|---|---|---|---|---|---|---|---|---|---|---|---|---|---|---|---|
| | C | Time (s.) | States | Edges | C | Time (s.) | States | Edges | C | Time (s.) | States | Edges | C | Time (s.) | States | Edges | C | Time (s.) | States | Edges |
| MilkBucket | 5 | 1.5 (0.2) | 3.0 (0.0) | 2.0 (0.0) | 5 | 3.2 (0.6) | 4.0 (0.0) | 3.6 (0.2) | 5 | 325.6 (29.7) | 13.4 (0.4) | 93.2 (1.7) | 5 | 347.5 (64.5) | 4.0 (0.0) | 14.0 (1.0) | 5 | 17.1 (5.5) | 4.0 (0.0) | 3.0 (0.0) |
| Book | 5 | 191.2 (36.4) | 5.0 (0.0) | 5.8 (0.2) | 0 | - | - | - | 5 | 288.9 (31.7) | 16.6 (3.1) | 119.0 (19.4) | 5 | 2261.0 (552.2) | 8.0 (0.0) | 31.2 (2.0) | 0 | - | - | - |
| BookQuill | 5 | 17.9 (1.4) | 4.0 (0.0) | 4.0 (0.0) | 0 | - | - | - | 5 | 308.6 (52.6) | 12.8 (0.5) | 92.8 (2.3) | 0 | - | - | - | 0 | - | - | - |
| Cake | 5 | 74.5 (25.7) | 4.0 (0.0) | 3.2 (0.2) | 0 | - | - | - | 4 | 290.6 (36.4) | 17.2 (2.5) | 110.2 (11.6) | 0 | - | - | - | 0 | - | - | - |
| RG | 5 | 0.9 (0.0) | 3.0 (0.0) | 2.0 (0.0) | 5 | 0.9 (0.0) | 3.0 (0.0) | 2.0 (0.0) | 0 | - | - | - | 0 | - | - | - | 5 | 32.3 (7.9) | 3.8 (0.2) | 82.4 (9.1) |
| RG&BC | 5 | 4.5 (0.3) | 4.0 (0.0) | 4.0 (0.0) | 0 | - | - | - | 0 | - | - | - | 0 | - | - | - | 0 | - | - | - |
| RGB&CMY | 5 | 15.1 (1.7) | 4.0 (0.0) | 4.0 (0.0) | 0 | - | - | - | 0 | - | - | - | 0 | - | - | - | 0 | - | - | - |

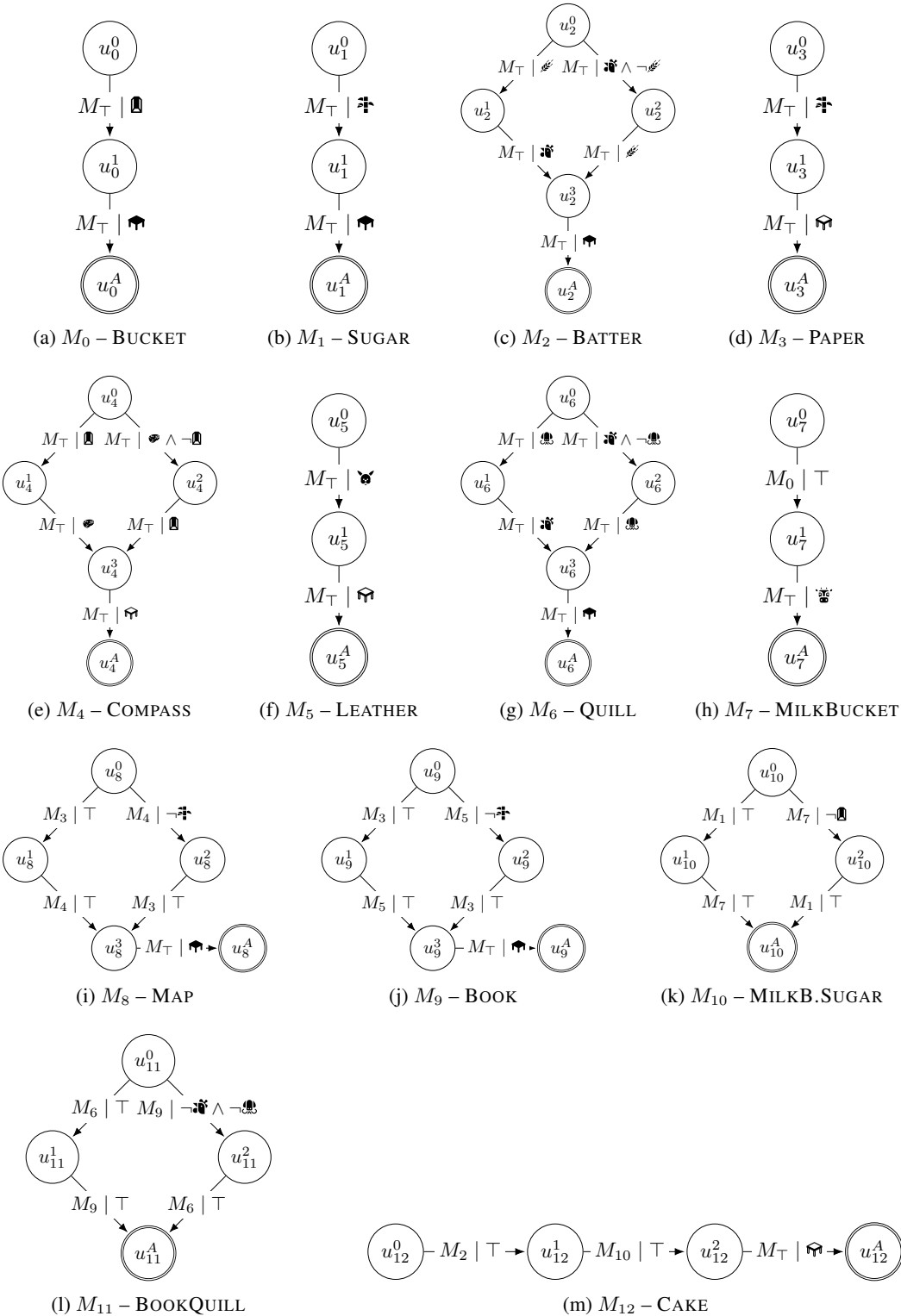

Figure 14: Root reward machines for each of the CRAFTWORLD tasks.

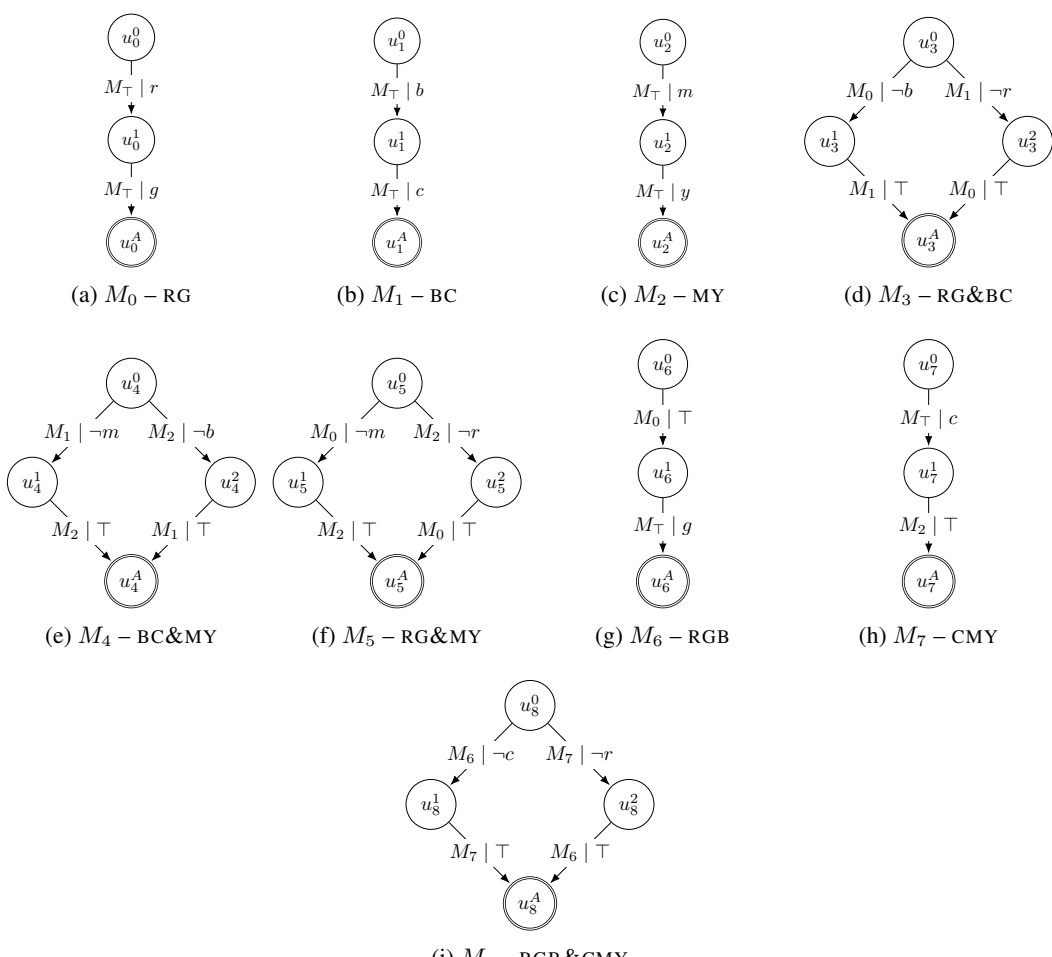

Figure 15: Root reward machines for each of the WATERWORLD tasks.

