# OpenReview forum: "Hierarchies of Reward Machines"
_ICLR.cc/2023/Conference — Submitted to ICLR 2023_

### Official Review · Reviewer_M97T · 2022-10-20

**Confidence:** 4
**Correctness:** 4
**Technical Novelty And Significance:** 3
**Empirical Novelty And Significance:** 3
**Recommendation:** 8

**Clarity, Quality, Novelty And Reproducibility:**

For both clarity and quality the whole paper passes my personal threshold for publication. Having said that, I think the first half of the paper is very well written and of high quality (many important theor. questions and relations answered), whereas 4, 5, and 6 could be slightly improved for an excellent paper. Readers that are completely unfamiliar with the Reward Machines framework might have a harder time with the first part of the paper, but I do not see an easy way to fix that (the paper does not have space for a general RM intro).

The method and algorithms in the paper are to the best of my knowledge novel and original. The extensive and detailed appendix helps greatly with reproducibility.


**Strength And Weaknesses:**

**Main contributions, Impact**

1) Hierarchical extension of the Reward Machines framework. Though not entirely unexpected, the construction presented is solid and well executed. I would give this a medium to high impact since it puts a lot of potential follow-up work on solid footing and addresses important shortcomings of basic reward machines.
2) Illustrative algorithm / proof of concept for learning policies with hierarchical reward machines. This certainly strengthens the paper since it allows for empirical results. I do not feel confident judging the sophistication of the algorithm, and suspect that there might be improvements in the near future inspired by this work. Impact: medium.
3) Proof of concept algorithm for learning hierarchical reward machines. Another great point to strengthen the paper since it addresses the criticism of having to manually construct reward machines. Quite a number of choices went into the algorithm, and it currently also comes with quite a number of restrictions / assumptions. While it would be nice to see some alternatives explored and compared to, I think this is beyond the scope of this paper (and certainly beyond the page limit), and similarly to 2) I would expect more follow-up work along these lines to appear in the near future. Impact: medium to potentially high (e.g. for learning complex reward functions from human data).

**Strengths**

* Interesting and natural open problem in the Reward Machines framework addressed in a theoretically sound and solid fashion
* Very well written intro and theory (up to Sec. 4)
* Adding important bits for empirical evaluation: (i) adapting/suggesting a hierarchical RL algorithm that can learn with HRMs, (ii) suggesting an algorithm to learn HRMs from data.

**Weaknesses**

* Paper is very dense, this comes at the expense of clarity and sufficient detail in sections 4, 5 and 6. Both 4 and 5 (including some empirical results form 6) could be turned into separate full publications. While I appreciate having 4, 5, and 6 in the paper, and would not recommend removing them, the strict page limitations of a conference make some trade-offs necessary. Additionally, the paper comes already with an excessive appendix (which contains interesting parts).
* Sections 4 and 5 propose two algorithms - both algorithms could have been designed differently, and there are some (more or less) obvious alternatives and ablations that would be nice to see; but the current paper simply does not have space for them (and I do not recommend filling up the appendix even more).

**Improvements**
1. There is not too much that can be done about a dense paper with an already long appendix, other than aiming for a journal paper with less severe page restrictions. While I think this would improve the paper, I also think that the current version is clearly above the threshold for publication at the conference, and I also think that publishing these results should not be delayed any longer. So I don’t expect to see an improvement here; maybe as a suggestion, the related work section could potentially be moved to the appendix to give 4 and 5 a bit more space.

2. I think it is worth giving Sections 4 and 5 another pass for clarity - it currently seems that all the information is in place, but not presented in the most easily understandable way. Some overview/summary/illustration of the actual scheme proposed and the algorithms implemented for the empirical evaluation would be helpful. Having said that, the two sections do currently pass my threshold for acceptance, but I think could be improved for an excellent paper.

3. Discussion of limitations of (L)HRMs is very short (determinism, having to define labeling functions). It would be nice to expand on this a little bit more, particularly compared to RMs.

**Minor comments**

A) Theorem 1: It would be nice to be more precise what “equivalent” means in the main paper (e.g. equivalent: an automaton that accepts the same language (language = set of all possible label traces \lambda that respect the MDP transition dynamics)).

B) Learning HRMs - why select tasks and instances with lower returns first? Why not choose highest returns first and switch once the increase in returns starts to saturate (i.e. learn the easiest tasks first, then switch to the next hardest ones once there is no progress on the easy ones anymore).

C) Learning HRMs - how exactly are learned options re-used? “An option is selected from a
pool of options appearing in lower height RMs”; how is this selection done, probabilistically or by exhaustive evaluation and some deterministic selection criterion?

D) Not sure if the experiments are optimal to show the advantages of non-flat HRMs. I would imagine non-flat HRMs to perform even better in cases where a simple task (such as gathering paper) must be repeated multiple times (e.g. to create a composite object of ‘book’ and ‘map’, ‘paper’ needs to be gathered twice). The current tasks (at first glance) seem to be chosen to collect a series of objects once. I might have missed that the same advantage would play out in flat HRMs (which are different from plain RMs?), so please correct me if this is wrong.


**Summary Of The Paper:**

**Update after rebuttal** I think the other reviewers have raised some valid criticism, and have definitely helped to make the manuscript more accessible and improve readability. Having read the authors' response I think that all main issues have been addressed in a satisfactory manner (certainly true for the issues raised by me), and therefore remain in favor of accepting the paper.


This paper extends the Reward Machine framework by allowing for hierarchical compositions of reward machines. The basic idea for the construction is that transitions in one reward machine are captured by whole separate reward machines - a higher-level reward machine can call a lower-level reward machine, which allows for compositionality and re-use of reward machines. The paper also shows how to use hierarchical RL to learn policies from hierarchical reward machines, and how to synthesize hierarchical reward machines from data. Evaluations on simple environments prove that the concept works, and show some favorable results compared to flat reward machines.

**Summary Of The Review:**

The paper constructs and analyzes an interesting and sensible hierarchical extension to the Reward Machines framework, which addresses important shortcomings and puts follow-up work on a solid theoretical footing. It also introduces two important and nontrivial algorithms for learning from HRMs and learning HRMs themselves from data. Overall I think this is a well executed and interesting extension to the Reward Machines framework, which has seen good adoption lately; the paper will be relevant to researchers in hierarchical RL and modeling of complex reward functions / specifications (which itself is a growing area of interest). The main idea of the extension is sensible and straightforward; the execution in the paper is very good and the precise theoretical construction presented is sound and interesting - this is a nontrivial contribution. My main criticism is that the paper is very dense and packs a lot of material into the page limit - while the first sections are very well written and can be well understood (with a background in basic Reward Machines), Section 4, 5, and 6 could be improved in terms of clarity and detail (but adding detail is hard given the already crammed paper). I think this is a high-quality, well-written and well executed paper that is ready for publication, and has good potential to stir follow-up theoretical work and almost certainly more investigations into learning with HRMs and synthesizing HRMs from data, and therefore currently suggest acceptance at the conference.

---

> ### Author Response · Authors · 2022-11-10
> **Response to Reviewer M97T**
>
> We are thankful for your comments and for appreciating the strengths of our work. We address your concerns and questions below.
>
> > Q1 The related work section could potentially be moved to the appendix to give 4 and 5 a bit more space.
>
> Thanks for the suggestion. We have decided not to apply it in the revision since it would be a major change in the paper. We note that Reviewer 8bbu emphasized comparing against related work, so moving it fully might be undesirable. We could potentially consider moving part of this section following a consensus among reviewers, leaving room for figures or extended explanations as suggested in the following question.
>
> > Q2 [On Sec. 4 and 5] Some overview/summary/illustration of the actual scheme proposed and the algorithms implemented for the empirical evaluation would be helpful.
>
> We have reworked the introduction of these sections such that the first paragraph gives an overview of the method (i.e. what parts are going to be presented, the multi-task setup in the case of Sec. 5) and assumptions are clearly stated (following a comment from Reviewer 8bbu).
>
> We have attempted to introduce illustrations but since they would have incurred major changes in the main paper (i.e. removing substantial portions of text) we have not done so. However, you can find the figure we have made for Sec. 5 in App. H along with a description of it, which we could potentially introduce in the main paper.
>
> > Q3 It would be nice to be more precise what “equivalent” means in the main paper
>
> We have slightly modified the paragraph to match closely the definition given in the appendix.
>
> > Q4 Why select tasks and instances with lower returns first? Why not choose highest returns first and switch once the increase in returns starts to saturate?
>
> We decided to assign higher probabilities to tasks and instances with lower returns based on the successes of previous methods in the literature (Andreas et al., ICML 2017; Pierrot et al., NeurIPS 2019). We experimented with the suggested alternative *but* without checking for saturation, which naturally did not work since already mastered tasks and instances will be given high probabilities.
>
> We believe your suggestion would fall into “computing the probabilities based on learning progress”. We believe it would be definitely interesting to evaluate the performance of these alternative methods (e.g., some of those by Matiisen et al. (2020)) in our setting.
>
> > Q5 How exactly are learned options re-used?
>
> Given all the options we have learned so far (formula and call options), we sequentially pick one uniformly at random and run its greedy policy until termination (and repeat until the end of the episode). We considered more complex methods, e.g. iteratively constraining the set of options depending on the propositions in the goal traces; however, we stuck to the finally implemented strategy for its simplicity. We have rephrased this part a bit (end of Sec. 5) to make this point clearer.
>
> > Q6 I would imagine non-flat HRMs to perform even better in cases where a simple task (such as gathering paper) must be repeated multiple times. [...] I might have missed that the same advantage would play out in flat HRMs (which are different from plain RMs?).
>
> The intuition is correct. In general, the more times RMs are called in the hierarchy, the better since it implies we are reusing knowledge very often and not learning it from scratch. Thus, we expect to observe non-flat HRMs to perform better. In the simplest case, we can observe benefits for a task like “gather paper $t$ times”: in the hierarchical case we just need to learn a single module for the “paper” task, whereas in the flat one we need to learn it $t$ times. We have run an experiment for $t=10$ and added preliminary results confirming the intuition in App. F.3.
>
> Regarding the second part of the question, the same advantage would not play out in flat HRMs. Flat HRMs are essentially plain RMs since their root is an RM that does not call any other RM (apart from the special leaf RM); thus, just like plain RMs, they suffer from a lack of modularity.
>
> ---
>
> We hope our answers resolve your questions and that our modifications make the paper more accessible. Please, let us know if you have further concerns or suggestions.

---

> > ### Comment · Reviewer_M97T · 2022-11-28
> > **Thank you for the response and additional material (illustration, control experiment, discussion)**
> >
> > Thank you for answering my questions - I agree that it is hard to add additional content to the main paper, since it would mean that other content needs to be moved to the appendix; from the reviews I think it became clear that clarity and accessibility to readers less familiar with the RM framework should be prioritized. I have no further comments, except thanking you for the additional experiment with repetitions - from a "marketing" viewpoint I think the tasks shown in the main paper make it challenging for HRMs to play out their full strengths (yet they do perform very well, justifying their use even in this setting); whereas you could have easily "made them look much better" by focusing on repeated sub-tasks. Don't get me wrong, I appreciate that the main comparison is actually on the challenging setting (a fair comparison), but I also like the addition in the appendix of showing the reader that in settings that are favorable for HRMs they outperform flat RMs even more significantly.

---

> > > ### Author Response · Authors · 2022-11-28
> > > **Response to Reviewer M97T**
> > >
> > > We are grateful for the comments and the fact that the modifications we have made to our submission are appreciated. Indeed, we believe that the suggested experiment shows the advantages of HRMs with respect to flat RMs in a more extreme case (the more tasks are repeated, the better). We thank you again for your suggestion to include this kind of experiment.
> > >
> > > If any additional questions/suggestions arise before the end of Stage 2, do not hesitate to ask/make them. We strongly believe that we have made progress in this direction in our revised version based on all reviewers' comments. We are committed to making the paper more accessible and we are happy to implement any further suggestions that help achieve this purpose.
> > >
> > > Thanks.

---

### Official Review · Reviewer_2jNi · 2022-10-24

**Confidence:** 3
**Correctness:** 2
**Technical Novelty And Significance:** 4
**Empirical Novelty And Significance:** 2
**Recommendation:** 5

**Clarity, Quality, Novelty And Reproducibility:**

Clarity:
I found the paper rather difficult to follow, which is partly due to the fact that I am not familiar with reward machine and the notation used in the paper. However, I also think that the formalism is not well presented:

- Section 2 states that $r(u, u')$ returns the reward for a transition of RM states. However, if I understand correctly, this reward should depend on the actual low-level transition (even with the assumed sparse reward structure, due to discounting). The RM paper (Toro Icarte et al. 2022) instead defines $\delta_r(u, u')$ which returns a reward *function*, rather than a scalar reward. This deviation is not addressed in current submission.

- I found it difficult to comprehend Figure 1b, in particular the "not rabbit" condition in the edge from $u_0^0$ to $u_0^1$, since a corresponding condition does not appear in the edge from $u_0^0$ to $u_0^2$. Only later, I realized that such condition needs to be added to preserve determinism, and that the choice for using the particular edge was arbitrary.

- The paper does not seem to mention whether a tuple is removed from the call stack after a reward machine "accepts". If no tuples are removed from the call stack, the statement that "each RM appears in the stack only once" would indicate a major limitation, as it would not only imply that RM can not call themself recursively, but also that the same RM can not be used at several stages of a rollout.

- The algorithms for learning policy and HRM are not sufficiently well discussed.  For learning the policies, I found the definition of the reward transition function confusing as it seems to ignore the effects of discounting. Furthermore, I do not see how the algorithm is capable of intra-option learning, given that it uses Semi-MDP Q-Learning and assumes options to terminate for updating the Q-function. I also found it confusing that the descriptions mentions DQN, although the experiments use grid worlds, which I assume used tabular Q functions.
For the algorithm for learning HRMs, I find it strange the paper only considers a multi-task setting, where *several* HRMs are learned. Wouldn't it be more sensible to consider learning of a single HRM for one task?

Quality:
The presented formalism of HRMs seems rather limited due to several strong assumptions:
- The paper only considers sparse reward function, where the reward is 1 for goal states and 0 otherwise.
- When learning HRMs, dead-end traces must be common across tasks and the depth of each hierarchy must be known a priori.
- It is not clear to me, whether HRMs can correctly handle discounting, and whether the same RM can be used multiple times during a task (like the "Navigate" action in the Tax environment (Dietterich, 2000).

Several aspects of the formalism seem problematic:
- The requirement that a trace uniquely specifies the option history ("determinism") can only be satisfied by introducing arbitrary preferences over subtasks, as in Fig. 1.b.
- By assuming that every RM contains a single accepting state and a single rejecting state some tasks may require an exponential number of RMs.

Novelty:
The formalism of HRMs is novel.

Reproducibility:
The paper does not provide source code nor sufficient details to reproduce the experiments.

**Strength And Weaknesses:**

Strength:
- Relevance: the paper tackles an important problem of learning and exploiting hierarchical task specifications.
- Novelty: The presented formalism (HRM) and the methods for learning policy and HRMs are novel.

Weaknesses:
- Clarity: The paper is hard to follow (see "Clarity" below)
- Significance: HRMs seem to suffer from too strong assumptions (see "Quality" below)

**Summary Of The Paper:**

The paper extends the concept of reward machines to hierarchical reward machines (HRM), and presents a method for learning policies for HRMs and for learning HRMs from data (learning is interleaved with policy optimization).

**Summary Of The Review:**

The paper makes relevant contributions, which, however, are currently not sufficiently well presented. There are several aspects that need to be clarified, for example, regarding the treatment of discounting and the possibility of executing the same RM several times during a single trajectory.

---

> ### Author Response · Authors · 2022-11-10
> **Response to Reviewer 2jNi**
>
> We thank you for the detailed review. We address all your concerns below.
>
> > Q1 [the reward transition function] should depend on the low-level transition
>
> HRMs are constituted by “*simple* reward machines”, a type of RM introduced by Toro Icarte et al. (2018, Def. 3.2) that does not depend on the low-level transition. Most of the work on RMs (see Sec. 7) has adopted this formalism. We have clarified this in the revised version.
>
> > Q2 In Fig 1b [...] "not rabbit" needs to be added to preserve determinism
>
> We have clarified this when introducing Fig. 1b (i.e. mention that “not rabbit” is used for determinism and that we explain why later).
>
> > Q3 The paper does not seem to mention whether a tuple is removed from the call stack after an RM "accepts" [then] the same RM cannot be used at several stages
>
> This is mentioned in p. 4’s enumeration: “If $u$ is an accepting state of $M_i$ and the stack $\Gamma$ is not empty, pop the top element of $\Gamma$ and return control to the previous RM”, so we can use the same RM at several stages (though not recursively, see p. 3).
>
> > Q4 the reward transition function [...] seems to ignore the effects of discounting
>
> The discounting is not taken into account in the reward transition function. *However*, it is used in the learning of call option policies: the reward $r$ in the second equation of Sec. 4 is the sum of *discounted* rewards (coming from the reward-transition function) during the $k$ steps the call option is executed. Thus, our algorithm considers discounting.
>
> > Q5 How the algorithm is capable of intra-option learning given that it uses SMDP Q-Learning
>
> Intra-option learning only occurs for *formula options*, whose policies are not learned using SMDP Q-Learning but Q-Learning, e.g. in Figure 1b the transitions experienced by “table” are used to train “rabbit” since all these policies share the same buffer. In contrast, SMDP Q-Learning is used to train *call options* and intra-option learning is not used in this case.
>
> > Q6 The descriptions mention DQN, [...] experiments use grid worlds, which I assume used tabular Q functions.
>
> Although the CraftWorld is a gridworld, we build on MiniGrid (Chevalier-Boisvert et al., 2018), whose observations prevent us from using a tabular approach (see App. F.1). We emphasize the WaterWorld domain is not tabular.
>
> > Q7 [Why not] learning a single HRM for one task?
>
> We justify our choice for learning several HRMs by considering the question: “How can we get the RMs upon which the HRM is learned?”. In the simplest case, we could have handcrafted these RMs, but we considered it is more interesting to learn them too. Besides, we could check if it is more feasible to learn a task’s parts rather than the full task at once.
>
> > Q8 The paper only considers sparse reward function, where the reward is 1 for goal states and 0 otherwise
>
> We make the assumption to make a direct mapping to the reward functions used in Sec. 4. The reward transition functions in Sec. 4 could differ, e.g. by applying the reward shaping method by Camacho et al. (IJCAI 2019) over the current functions. Note that the HRM learning algorithm is reward-independent.
>
> > Q9 [Sec. 5] dead-end traces must be common across tasks and the depth of each hierarchy must be known
>
> We foresee ways of addressing the former by conditioning the policies on the edges to the rejecting states, and the latter by associating heights to each HRM dynamically depending on their estimated difficulty. However, we emphasize ours is a first step towards learning HRMs and believe the assumptions are fair at this stage (see also Q1, Rev. 8bbu).
>
> > Q10 The requirement that a trace uniquely specifies the option history ("determinism") can only be satisfied by introducing arbitrary preferences over subtasks
>
> We do not introduce arbitrary preferences over subtasks: these would be effectively broken once a counterexample trace is observed. However, the edge where a negation is placed can indeed be arbitrary if the conditions labeling such edges are implicitly mutually exclusive, i.e. they can never happen together. For example, in CraftWorld two propositions cannot happen together, so the learner would enforce determinism arbitrarily in the RMs of Fig. 1.
>
> > Q11 By assuming that every RM contains single accepting and rejecting states some tasks may require an exponential number of RMs
>
> This is a non-problematic assumption: since these states do not have outgoing transitions, it does not make a difference to have one or more of them in an RM. Thus, the number of RMs does not grow exponentially due to this.
>
> > Q12 The paper does not provide code nor sufficient details to reproduce the experiments
>
> We will provide the code if the paper is accepted. The appendix contains detailed descriptions to ensure reproducibility.
>
> ---
>
> We hope our answers shed light on your concerns and they are sufficient to reconsider your assessment. Please, let us know if any aspect requires clarification.

---

### Official Review · Reviewer_8bbu · 2022-10-25

**Confidence:** 3
**Correctness:** 4
**Technical Novelty And Significance:** 3
**Empirical Novelty And Significance:** 2
**Recommendation:** 6

**Clarity, Quality, Novelty And Reproducibility:**

The paper is rigorous but hard to read. It provides a clear extension of the prior RM work, but significance with respect to other option learning papers is not clearly established.

**Strength And Weaknesses:**

# Strengths

- incorporating hierarchy into RMs seems like an important step towards making them more applicable to long-horizon tasks

- the submission rigorously defines all aspects of the problem setup

- the toy experiments demonstrate that the introduced hierarchical RMs learn faster than regular (flat) RMs if a RM hierarchy is pre-defined, they also demonstrate learning of long-horizon tasks with HRMs (not possible with RMs)


# Weaknesses

(A) **hard to follow writing**: The writing of the paper is very hard to follow. I attribute this in large part to the very heavy usage of formalism and introduction of terms, which make smooth reading challenging. It seems that the paper could benefit a lot from replacing / augmenting some of the formal definitions with their plain-english equivalents. On other occasions crucial terms are not properly defined. Eg in the background section 2, the terms “propositions” and “labels” are not properly introduced, no intuition / example is given for what they could refer to. This makes it very hard to understand the paper for readers (like myself) that aren’t already familiar with the RM literature.

(B) **assumptions not clearly mentioned**: compared to other papers that use (deep) RL to learn options & policies the submission seemingly makes a number of assumptions (given subtask hierarchies or at least a priori knowledge about depth of the hierarchy, symbolic observations, access to training tasks for *non-hierarchical* RMs, …). However, these assumptions are only step-by-step introduced throughout the text and can be easily missed. It would be good to add a “problem formulation” section or similar that explicitly lists the required assumptions for learning hierarchies of HRMs and how they compare to prior works, so that readers can better understand in what use cases they can consider HRMs.

(C) **no comparisons to other option learning approaches**: the only comparison in the experimental section is to flat RMs — this is a meaningful comparison since the proposed approach is a hierarchical extension of flat RMs. However, since the paper proposes an approach for learning hierarchical options, it should compare to prior works on option learning with deep RL too, if applicable. E.g. the work of Sungryull Sohn could be applicable, since it also addresses hierarchical RL with discrete observational symbols (eg Sohn et al., NeurIPS 2018).

(D) **only tested in toy environments**: the submission tests the proposed approach only in simple 2D grid-world like environments with low-dimensional observation and action spaces. This makes it hard to judge how well the method would scale to more realistic settings, eg. learning robot control in 3D environments.


**Summary Of The Paper:**

The paper extends the Reward Machines (RM) formalism to hierarchical RMs (HRMs) by allowing RMs to call other RMs as sub-programs. The paper demonstrates that this allows for more efficient learning of RMs on long-horizon tasks in a number of toy grid-world environments.

**Summary Of The Review:**

Overall, I found the submission quite hard to read. I was not previously familiar with RMs and I found the heavy use of formalism and special terms throughout the submission hampered the reading flow. It also makes it hard for me to properly judge the novelty of the work: my perception is that the submission adds novelty over the previous RM work, but it also seems to make multiple assumptions that can reduce its impact outside the currently rather niche RM topic. The experimental evaluation does not help judge this either, since it is limited to toy environments and does not compare to non-RM works.

I am leaning towards not accepting the submission, but am willing to change my mind if the authors can clearly state their assumptions and how they relate to other option learning papers (ie is there good use cases for their approach that others cannot do, can you compare to any non-RM approach?). Additionally, I am curious to know what other reviewers opinions are, especially for reviewers with a better overview of the RM literature.

=============
Post-Rebuttal: I updated my score to a weak accept based on the discussion with the other reviewers -- see my reply below.

---

> ### Author Response · Authors · 2022-11-10
> **Response to Reviewer 8bbu**
>
> We are grateful for the thorough and thoughtful review. We address your concerns below.
>
> > Q1 assumptions not clearly mentioned [...] can reduce its impact outside the currently rather niche RM topic
>
> We believe the first part of the comment concerns Sec. 5 (learning of the HRMs). We have separated the first paragraph of this section into two: the first one keeps giving an overview of the method, while the second one mentions all assumptions of this method, as suggested.
>
> Regarding the impact of the assumptions made throughout the paper, we distinguish between two groups of them:
> 1. Those on the formalism (Sec. 3), which are made to preserve equivalence between HRMs and RMs so that the former are as applicable as the latter (if not more, due to their modularity).
> 2. Those on the methods (Sec. 4, 5), which are not essential. We made them to propose a first stepping stone towards exploiting modularity in RMs, and clearly display the benefits of HRMs with respect to RMs experimentally. Hence, we believe these assumptions are fair at this point and, crucially, addressable in future work as outlined in Sec. 8. Note that removing some of these assumptions implies answering non-trivial questions such as: “How can we incrementally derive a set of propositions/high-level events and tasks from raw data?”, which do not only concern RMs but any work that aims to minimize human input.
>
> > Q2 no comparisons to other option learning approaches
>
> In this work, we have focused on showing the benefits of hierarchically composing RMs with respect to flat RMs, both when the (H)RMs are given or learned from experience. While there are similarities between RM learning and option discovery (see Sec. 7), we found them difficult to compare quantitatively.  Learning an RM, unlike option discovery, is not about learning a set of options, but learning the structure of a task in terms of a set of propositions. Crucially, combinations between propositions result in a relevant set of options (e.g. “go to the table”), and the RM sequences represent the task’s structure. Thus, we believed that using an option discovery method when such potential options are already present would not give as valuable insights as performing a detailed comparison against other RM learning methods.
>
> Qualitative comparisons with respect to option discovery methods (and HRL methods in general) are made in Sec. 7, where we mention a few usual assumptions such as bounding the number of options, or the need to solve each task at least once. Because of space reasons we omitted a brief mention of *bottleneck* option discovery methods, where a bottleneck normally is a set of states always on the path to a  solution. The difference to other methods is that our bottlenecks are abstracted into propositional logic formulas, whereas the aforementioned methods define them directly over the state space.
>
> We thank you for pointing us to Sohn et al.’s work. We believe there are a couple of crucial differences. First, Sohn et al. assume that the subtask graphs are given, while we learn the RMs from experience. Second, the aim of Sohn et al. is to generalize to previously unseen subtask graphs, while we assume that the tasks remain fixed throughout (though the learned RMs may be refined over time).
>
> > Q3 hard to follow writing
>
> We are committed to making the formalism understandable by any reader without previous experience with RMs. We have modified the submission to indicate that propositions correspond to high-level events, as described by Toro Icarte et al. (2018). We have introduced other suggestions to improve clarity (see general message on the revised version). We emphasize we have done our best to provide plain English descriptions in intricate parts (e.g., the hierarchical transition function) and added step-by-step examples in the appendices.
>
> > Q4 only tested in toy environments
>
> Due to limited resources (we used CPUs, see the first paragraph of App. F), it is hard to evaluate our approach in more intricate environments that require training more complex models. We emphasize, nonetheless, that 1) the domains we have used *are not* tabular, 2) the WaterWorld domain is not a grid-world, and 3) we strongly believe our results transfer to more complex domains as long as tasks can be decomposed into simpler ones. For instance, RMs have been applied in vision-based robotic manipulation (Camacho et al., ICRA 2021); thus, HRMs and methods that exploit them are also applicable in that setting. Finally, one could use DDPG (Lillicrap et al., ICLR 2016) to learn the formula option policies akin to our way of using DQNs in order to deal with continuous state spaces.
>
> ---
>
> We hope our answers address your concerns and they are sufficient to reconsider your assessment. We are happy to work further to make the paper clearer and welcome any suggestions you may have. Please, let us know if any aspect requires clarification.

---

> > ### Comment · Reviewer_8bbu · 2022-11-29
> > **Rebuttal Response**
> >
> > Thank you for your reply!
> >
> > I have looked over the updated paper and also had a discussion with the other reviewers and the AC. I do not think the rebuttal fundamentally changed my perception of the paper. I think it still is not very easy for someone who is not familiar with the RM framework to grasp the key takeaways from this paper and how it relates to their research. I think a section of the type "this is how (H)RMs differ from common_technique_XYZ and here are the additional assumptions we make over those in common_technique_XYZ" would really help and I don't think the rebuttal added that. That might also make it easier to understand why there's currently no easy comparison to techniques outside the RM framework.
> >
> > That being said, I gave my review for this paper the lowest confidence score out of all papers in my batch this year, since I am not familiar with RMs and thus have a hard time judging the impact _within_ the RM community. I trust reviewer M97T's expertise that this paper is a well-executed and relevant extension within the field of RMs. And since there seems to exist a small RM community (i.e. this is not the first paper) I am okay accepting the paper based on that ground. While I am unsure of the impact of this work outside the RM community, I tend to err on the accepting side in such cases instead of relying on my (non-expert) guess as to what future impact is possible.

---

> > > ### Author Response · Authors · 2022-11-29
> > > **Response to Reviewer 8bbu**
> > >
> > > We are really thankful for your comments. We agree that having a section such as the one suggested could be helpful for readers unfamiliar with RMs. We are determined to make our work as accessible as we can; therefore, we plan to include this section in future versions of the paper, e.g. putting all assumptions we have made for each of the parts together and compare them against those made in related work so that the applicability of (H)RMs is made clearer.
> > >
> > > We are once again grateful for your thorough review and strongly believe your comments have helped (and will help, following the above) improve the work. If you have any additional questions/suggestions before the deadline, do not hesitate to ask/make them.

---

> > > > ### Comment · Reviewer_M97T · 2022-11-30
> > > > **+1**
> > > >
> > > > The reviewer discussion was quite fruitful for me, and given the discussion I would like to second 8bbu's (and 2jNi's) suggestions to make sure the manuscript is as accessible as possible, particularly to readers unfamiliar with the basic RM formalism. One aspect that might be harder to recognize as a significant contribution is that the (H)RM framework provides powerful methodology for constructing specifications (reward functions for non-Markovian tasks) with nice (theoretical) properties. On top of that, there are algorithms for learning with an (H)RM that can exploit the reward-function structure, and algorithms for learning an (H)RM itself from data; though the latter two might still be improved in the future (what's currently proposed is clearly more than a first stab, but also makes some simplifying assumptions that might be reduced in future work).

---

> > > > > ### Author Response · Authors · 2022-12-04
> > > > > **Added Summary of Planned Changes for Future Versions.**
> > > > >
> > > > > Thanks for your comment. We have decided to put together the main suggestions related to accessibility that we plan to implement in future versions of the paper [here](https://openreview.net/forum?id=wV09GfqYC-n&noteId=TxcBASXu1tD). We strongly believe that applying these changes would definitely make the paper more accessible to a reader unfamiliar with RMs.

---

### Author Response · Authors · 2022-11-10
**List of Changes in Revision**

We thank the reviewers for their comments and suggestions. We have uploaded a revised version implementing them. We compile a list of the changes here, including the page in the paper, the reviewer, and the associated question number in our response. All changes in the paper are colored in red.

* Page 2 - Reviewer 8bbu [Q3] - State that *propositions* represent high-level events, following the terminology used by Toro Icarte et al. (2018). *Labels* are sets of propositions.
* Page 2 - Reviewer 2jNi [Q1] - Clarify that the reward machines we consider are *simple* reward machines.
* Page 3 - Reviewer 2jNi [Q2] - State that the condition “not rabbit” is used for determinism and point out it will be explained later.
* Page 4 - Reviewer M97T [Q3] - We have rephrased the definition of equivalence to match that mentioned in the appendix.
* Page 5 - General - We have slightly rewritten the introduction dividing it into two paragraphs, moving the description of the option types to a new paragraph for clarity. The first provides a general description of the method, as suggested by Reviewer M97T.
* Page 6 - General - We have divided the first paragraph into two, thus distinguishing between (a) a general overview and setup of the method, and (b) the assumptions of the method. We have had to remove some content from the “Curriculum learning” section (the low-level criteria for choosing tasks and instances), which was covered in Appendix D in detail.
* Page 6 - Reviewer 8bbu [Q1] - We have put all assumptions together in the second paragraph for clarity.
* Page 6 - Reviewer M97T [Q5] - We have clarified how options in lower level machines are chosen for exploration when learning the higher level ones.
* Pages 38 (description) and 42 (plot) - Reviewer M97T [Q6] - We have added an experiment in response to the reviewer for an HRM that calls the Paper RM 10 times in a row. We have shown that exploiting the non-flat HRM leads to faster convergence.
* Page 44 - Reviewer M97T [Q2] - We have added a figure illustrating LHRM along with a description of it. We have decided not to place it in the main paper for now since doing so would incur major changes (i.e. removing some of the content or placing it in the appendix) at this point.

---

> ### Author Response · Authors · 2022-11-16
> **Follow Up before End of Discussion Stage 1**
>
> Dear reviewers,
>
> we hope that the revised version we uploaded and our responses address all your questions and concerns. Please let us know if there are any aspects of the paper that you would like us to clarify in a revised version of the paper since further updates will not be enabled during Stage 2. We are also happy to answer any questions you may have.
>
> Kind regards.

---

> ### Author Response · Authors · 2022-11-24
> **Follow Up Discussion Stage 2**
>
> Dear reviewers,
>
> we thank you again for your comments and suggestions, which we strongly believe have improved the quality of the paper. Please, let us know if our answers have addressed the concerns expressed in your reviews and, if so, whether you would consider updating your assessments. We are committed to discussing any remaining questions you may have.
>
> Thanks again.

---

### Author Response · Authors · 2022-12-04
**List of Planned Changes in Future Versions**

We are very grateful to all the reviewers for their comments and strong participation in the discussions. A common concern is that the paper can be less accessible to readers unfamiliar with reward machines. Given that the period for making changes to the manuscript expired, we proceed to make a list of the modifications we aim to do following the reviewers’ suggestions in future versions of the paper:

1. Modify Section 2 (Background) to include the description of the CraftWorld domain originally introduced in Section 3 (Formalization of Hierarchies of Reward Machines). By doing so, we can exemplify many of the crucial elements that characterize RMs before introducing our hierarchies (e.g., a proposition, an RM, a trace, …). We strongly believe that this would make the transition to our formalism easier to understand.

2. Highlight the importance of the “hierarchies of RMs” formalism itself, as suggested by Reviewer M97T [(link)](https://openreview.net/forum?id=wV09GfqYC-n&noteId=77L6KKqByD): “the (H)RM framework provides a powerful methodology for constructing specifications (reward functions for non-Markovian tasks) with nice (theoretical) properties”. We will also emphasize upfront that the proposed algorithms for policy and HRM learning are the first steps and that their corresponding assumptions can be relaxed in the future.

3. Add a section in the appendix where we compare the assumptions we make (or that RMs in general make) with respect to other works in deeper detail, as suggested by Reviewer 8bbu [(link)](https://openreview.net/forum?id=wV09GfqYC-n&noteId=FMh3V40YG_R). We believe this could make the applicability of (H)RMs clearer to any reader. If space allows, we will try to keep as much information as possible in the main paper.

We strongly believe that applying these changes would substantially make the paper more accessible and also drive future work in exciting directions. If you have any further suggestions, please do not hesitate to let us know.

---

### Decision · Program_Chairs · 2023-01-20

**Decision:**

Reject

**Justification For Why Not Higher Score:**

My own sense is that this paper is asking a lot in terms of reader investment for primary demonstration only on gridworlds. I have a lot of respect for reviewer M97T's expert judgement, and I do trust their assessment that the work is a technically valid contribution within the RM framework.  But, in my opinion, in order to accept the paper, I need to be a bit more convinced that the investment in this framework is going to be useful.  So I would especially impact weaknesses (C) and (D) in reviewer 8bby's review.  Evaluation limited to gridworlds with comparisons only to standard RMs is not an externally grounded form of support.  Essentially, this paper is only targeted to aficionados of the RM framework.

**Justification For Why Not Lower Score:**

N/A

**Metareview: Summary, Strengths And Weaknesses:**

This paper proposes an extension of reward machines to hierarchical reward machines (HRMs).  Since manually designing HRMs is not generally practical, the authors introduce a scheme whereby the HRM can be learned along with policies that solve the various tasks.

One reviewer was quite positive about this work and the other two reviewers were borderline.  Reviewer M97T, the most positive reviewer, saw this work as a natural extension of the RM literature and believed it is positive that learning approaches are being introduced.  However, this reviewer did find the paper a bit dense at the expense of clarity and believed some arbitrary choices were being made.  Reviewer 2jNi believes the work addresses a meaningful problem and is novel, but the approach is hard to follow and may have too many assumptions built in.  Reviewer 8bbu identifies strengths of the paper as allowing RMs to extend to longer horizon tasks, doing so rigorously, and demonstrating the approach relative to flat RMs (albeit only on grid-world environments).  However, 8bbu also articulates 4 core weakness (plainer writing, clearer assumptions, compare outside the RM literature, test outside of toy grid worlds).  My own reading of the paper lead me to strongly second these 4 identified weaknesses.

**Summary Of Ac-Reviewer Meeting:**

Given the mixed scores, we held a video chat discussion of the paper and all of us (AC + 3 reviewers) were able to participate.  Reviewer M97T noted that they are fairly familiar with the RM framework and that this made the paper much easier to follow and see how the contribution fits in.  Within the RM framework, the hierarchical extension makes sense and is coherent.  The biggest weaknesses of this work is that the paper is likely not accessible standalone and this framework does not presently have wide adoption, effectively rendering this contribution somewhat niche (i.e. to a limited audience).  Reviewer 2jNi finds RMs and HRMs interesting, but does not see a clear demonstration of (broader) utility in this paper.  Reviewer 8bbu was unfamiliar with the RM framework and willing to defer judgement to M97T about how this work makes sense in that context.  However, this reviewer highlighted that the comparisons are only made within the framework (i.e., HRMs vs flat RMs).  And at the same time, this framework isn't here shown to be applicable to problems that are of wide interest.  Reviewer 8bby also noted (similar to their later post when updating their score to a 6) that the author feedback didn't address their 4 core concerns, but that they are open-minded, given reviewer M97T's expertise.  We continued with more nuanced discussion of the specific merits, but my main take-away was uncertainty about whether we should be publishing incremental development of this framework out of technical/theoretical interest, despite lack of evidence that it is going to be useful on more challenging problems.